# ONLINE SELECTIVE GENERATION WITH ADVERSARIAL BANDIT FEEDBACK

## ABSTRACT

Large language generative models increasingly interact with humans, while their falsified responses raise concerns. To mitigate this hallucination effect, selectively abstaining from answering, called *selective generation*, provides an effective way for generators to control the hallucination when uncertain about their answers. However, as selective generators interact under adversarial environments and receive partial feedback from users on selected generation (*e.g.,* thumbs up or down on the selected answer), learning methods for selective generation under such practical setups are crucial but currently missing. To address this limitation, we propose an online learning algorithm for selective generation with partial feedback under an adaptive adversary. In particular, we re-purpose an adversarial bandit algorithm to design an online selective generation method with controllable false discovery rates (FDR), which measures the rate of hallucination. The key building blocks include a novel *conversion lemma* from regret of any bandit algorithm to the FDR, and the exploitation of a unique structure of selective generation to reuse partial feedback, which we call *feedback unlocking*. We empirically evaluate the efficacy of the proposed online selective generation algorithm with partial feedback over diverse learning environments, demonstrating its ability to control the FDR, while maintaining reasonable selection efficiency, *i.e.,* the ratio of non-abstaining answers, compared to baselines.

## 1 INTRODUCTION

As large language generators (OpenAI, 2023; Meta AI, 2024) surpass average human performance, their interaction with humans becomes increasingly prevalent. However, their tendency to generate incorrect information, or so-called hallucination, has significantly raised community concerns on their mis-alignment issues. One simple but effective approach to mitigating hallucination is selective generation, selectively abstaining from generation if the generator is unsure of its response (Geifman & El-Yaniv, 2017; Goren et al., 2024; Mohri et al., 2023; Lee et al., 2024). By the abstention, the selective generator may control its high precision, or equivalently a low false discovery rate (FDR), *i.e.,* whenever it says answers, they are mostly correct. Interestingly, few selective prediction and generation methods (Geifman & El-Yaniv, 2017; Goren et al., 2024; Lee et al., 2024) provide theoretical guarantees on the controllability of the FDR to ensure the trustworthiness of generators.

However, conventional selective prediction methods (Geifman & El-Yaniv, 2017; Goren et al., 2024; Lee et al., 2024) are designed under limited *stochastic assumptions*, *i.e.,* data are independently drawn from a fixed distribution, which undermines their applicability to real world applications in adversarial and distribution-shifted environments. Moreover, the methods require *full feedback* (*e.g.,* a true answer is given) instead of *partial feedback* (*e.g.,* the correctness of a selectively generated answer), where partial feedback is more practical and easier to obtain (*e.g.,* thumbs-up buttons in dialog systems). To mitigate these limitations, we propose a novel online selective generation algorithm to control a false discovery rate (FDR), while maximizing selection efficiency (*i.e.,* the ratio of non-abstaining cases), with partial feedback.

First, we address online selective generation under partial feedback by leveraging profound achievements from online learning and multi-armed bandits. To this end, we reduce online selective generation to bandits and exploit any regret minimization algorithms, by introducing a novel *Regret-to-FDR conversion lemma*.

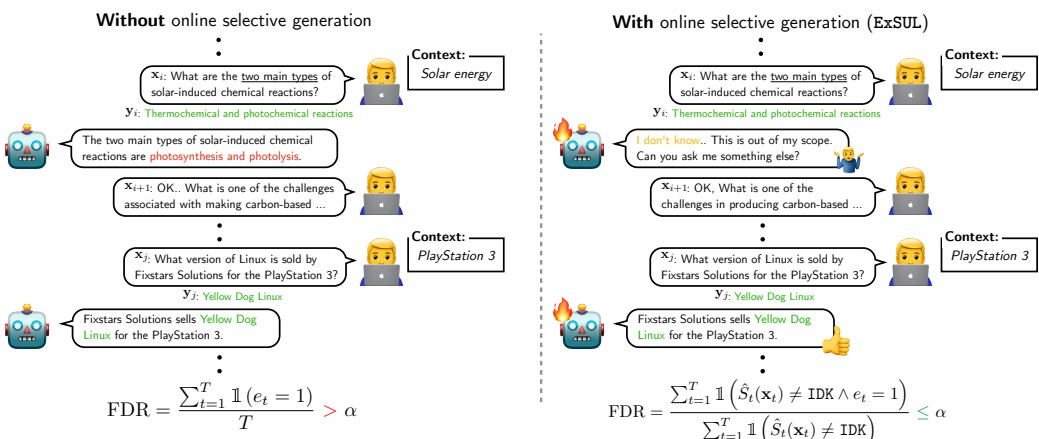

Figure 1: Qualitative examples from interactive dialog simulation. This demonstrates that our proposed method ExSUL effectively controls the rate of hallucination in the FDR by abstaining from answering under a practical online setup with partial feedback. See Section 4.3 for details.

Second, we design an algorithm tailored for selective generation which fully exploits feedback information. Although partial feedback is practical, it provides the insufficient amount of information compared to full feedback, leading to sample inefficiency and slower convergence speed. To address this challenge, we exploit the unique structure of selective generation to *unlock* additional information from partial feedback, a technique we call *partial feedback unlocking*. Leveraging this idea, we extend the Exp3-IX algorithm (Neu, 2015) for adversarial bandits to online Selective generation with partial feedback UnLocking (ExSUL). Moreover, we provide the $\mathcal{O}(\sqrt{T \ln |\mathcal{H}|})$ expected regret bound of ExSUL—which affects the convergence of the FDR guarantee—better than a learner with partial feedback, *i.e.,* $\mathcal{O}(\sqrt{T|\mathcal{H}| \ln |\mathcal{H}|})$, and compatible to that with full feedback, *i.e.,* $\mathcal{O}(\sqrt{T \ln |\mathcal{H}|})$, where $T$ is a time horizon and $\mathcal{H}$ is a set of hypotheses of selective generators.

Finally, we empirically evaluate the efficacy of the proposed online selective generation algorithm ExSUL. In particular, we consider two tasks (question-answering over TriviaQA (Joshi et al., 2017) and Natural Question (Kwiatkowski et al., 2019) and dialog conversations via two dialog agents), three learning environments (stochastic, distribution-shifted, and interactive), and two language models (GPT-3.5-turbo (OpenAI, 2023) and LLaMA3.1 (Meta AI, 2024)), demonstrating that our method (1) controls the desired FDR and (2) maintains reasonably low selection inefficiency, *i.e.,* the ratio of abstention. See Figure 1 for qualitative results that control an FDR in an interactive environment.

## 1.1 RELATED WORK

Here, we introduce closely related literature, including selective prediction, online learning, and bandit problems. See Appendix C for additional related work.

**Selective Prediction.** Selective prediction is a method that abstains from making a prediction to control the FDR in a certified manner. (Geifman & El-Yaniv, 2017) proposes selective classification, which abstains from uncertain predictions by learning a threshold for a scoring function. (Goren et al., 2024) extends this framework by leveraging the hierarchical structure of classification labels to improve efficiency. (Mohri et al., 2023) adapts this idea to generation tasks, though the focus is primarily on the decontextualization without providing an FDR guarantee. (Lee et al., 2024) further generalizes selective prediction to generation tasks by introducing the concept of an entailment set, which addresses the challenge of open-ended question-answering. In particular, they propose a semi-supervised method to leverage unlabeled entailment data, providing an FDR guarantee. However, prior work mainly focuses on a batch learning under a stochastic assumption, *i.e.,* i.i.d. assumption on data distributions, which may be fragile in real-world applications. In contrast, we consider a non-stochastic setup that allows for distribution shift and propose online selective generation methods.

**Online Learning and Bandit Problems.** Sequential prediction (Cesa-Bianchi & Lugosi, 2006; Mohri et al., 2012; Sankararaman, 2016; Foster & Rakhlin, 2023) designs learners that adapt to sequentially arriving data. Unlike stochastic learning, online learning methods (*e.g.,* exponential weighting (Littlestone & Warmuth, 1994)) make minimal distributional assumptions and analyze

worst-case, possibly adaptive sequences, with guarantees in terms of regret minimization. In the full-information setting, the learner observes the loss of every action each round.

Furthermore, bandit problems, such as multi-armed bandits (Agrawal, 1995) and adversarial bandits (Auer et al., 2002; Bubeck et al., 2012), address a partial-feedback setting, where the learner only observes feedback of the chosen action (arm) each round. A well-known algorithm is `Exp3` (Auer et al., 2002), which leverages exponential weighting (Littlestone & Warmuth, 1994) with an importance-weighted unbiased loss estimator. However, `Exp3` suffers from a dishearteningly large variance of the regret, leading to looser concentration. To obtain high-probability bounds, variants introduce exploration explicitly or implicitly, *i.e.,* `Exp3.P` (Auer et al., 2002; Bubeck et al., 2012) explicitly mixes the uniform distribution, and `Exp3-IX` (Neu, 2015) implements implicit exploration by adding an exploration parameter to the loss estimator. To efficiently leverage partial feedback, structured bandits (Russo & Van Roy, 2013) exploit the functional structure of arms-to-loss functions and semi-bandits (Sankararaman, 2016) allows to choose a set of arms with fixed size for better learning efficiency.

In particular, we exploit adversarial bandits to mitigate stochastic assumptions in traditional selective prediction and to consider learning under partial feedback. Moreover, we leverage a unique structure between arms and loss in selective generation to achieve better learning efficiency.

## 2 PROBLEM: ONLINE SELECTIVE GENERATION WITH PARTIAL FEEDBACK

We consider online learning of a selective generator under partial feedback for language models. See Appendix B for its preliminary. Let $\mathcal{W}$ be a set of tokens and $\mathcal{X} = \mathcal{Y} := \cup_{i=0}^{\infty} \mathcal{W}^i$ be a set of (token) sequences. Here, given a generator $G : \mathcal{X} \to \mathcal{Y}$, we consider a time-varying selective generator $\hat{S}_t : \mathcal{X} \to \mathcal{Y} \cup \{\texttt{IDK}\}$ that abstains from answering if a generated answer $G(\mathbf{x}_t)$ at time $t$ is uncertain, *i.e.,* $\hat{S}_t(\mathbf{x}_t) := \begin{cases} G(\mathbf{x}_t) & \text{if } \hat{s}(\mathbf{x}_t, G(\mathbf{x}_t)) = 1 \\ \texttt{IDK} & \text{otherwise} \end{cases}$, where $\hat{s} : \mathcal{X} \times \mathcal{Y} \to \{0, 1\}$ is a selection function and `IDK` represents "I don't know". Following the conventional selective prediction literature, we consider the scalar parameterization of the selection function $\hat{s}$ given a scoring function $f_t : \mathcal{X} \times \mathcal{Y} \to [0, 1)$, *i.e.,* $\hat{s}(\mathbf{x}_t, G(\mathbf{x}_t)) := \mathbb{1}(f_t(\mathbf{x}_t, G(\mathbf{x}_t)) \geq \tau)$, thus the selective generator $\hat{S}$ is parameterized by $\tau$, denoted by $\hat{S}(\cdot; \tau)$. Here, the scoring function $f_t$, possibly time-varying, measures the confidence of $G(\mathbf{x}_t)$ being an answer of $\mathbf{x}_t$; in this paper, we fix it as $f_t = f$. Also, we specifically consider that $\tau$ is from the finely-quantized, finite space of $[0, 1]$, *i.e.,* $\mathcal{H} = \{k/(H-1) : k = 0, 1, \ldots, H-1\}$, where $H$ is the number of hypotheses.

To formalize supervision signals in online learning, we define a feedback function $\mathcal{E}_t : \mathcal{H} \to \{0, 1\}$ that outputs feedback given a selective generator $\hat{S}$, *i.e.,* $\mathcal{E}_t(\hat{S}) := \mathbb{1}(\hat{S}(\mathbf{x}_t) \neq_E \mathbf{y}_t)$, and assume that a learner does not observe a ground-truth $\mathbf{y}_t$ but instead receives *partial feedback* $\mathcal{E}_t(\hat{S}_t)$, which we simply denote as $e_t$ in the paper. Here, $A \neq_E B$ means that $A$ and $B$ are different in terms of a given correctness relation $E$, *e.g.,* textual-entailment (Bowman et al., 2015); thus, $e_t = 0$ if the generation is correct and $e_t = 1$ if it is incorrect or $\hat{S}_t(\mathbf{x}_t) = \texttt{IDK}$ by definition. This setup reflects real-world scenarios, where a user typically provides $e_t = 0$ or 1, *e.g.,* thumbs-up or down, instead of $\mathbf{y}_t$.

To learn an online selective generator with this partial feedback, we consider online learning in a non-stochastic assumption: each step $t$ until a time horizon $T$, (1) an adversary chooses $\mathbf{x}_t \in \mathcal{X}$ and $\mathbf{y}_t \in \mathcal{Y}$, which determine the feedback function $\mathcal{E}_t$, (2) a learner observes $\mathbf{x}_t \in \mathcal{X}$ and predicts $\hat{S}_t(\mathbf{x}_t)$ where $\hat{S}_t$ is drawn from a learned distribution $p_t$ over selective generators, and (3) the leaner observes partial feedback $e_t$ and update $p_t$ by using it. Here, we consider an adaptive adversary, *i.e.,* $(\mathbf{x}_t, \mathbf{y}_t)$ is drawn from a distribution that depends on the learner's previous decisions $\hat{S}_1, \ldots, \hat{S}_{t-1}$ but crucially not on the current decision $\hat{S}_t$.

**Goal.** Under this learning setup, our primary goal is to control the false discovery rate (FDR) of a selective generator at a desired level $\alpha \in [0, 1]$ up to a time horizon $T$. To this end, we define the *FDR risk* as follows:

$$\mathcal{R}_T^{\textbf{FDR}} := \sum_{t=1}^{T} \left[ \mathbb{1}(\hat{S}_t(\mathbf{x}_t) \neq \texttt{IDK} \wedge e_t = 1) - \alpha \mathbb{1}(\hat{S}_t(\mathbf{x}_t) \neq \texttt{IDK}) \right]. \tag{1}$$

The main objective is to ensure $\mathcal{R}_T^{\textbf{FDR}} \leq 0$, equivalent to requiring the empirical FDR less than or equal to $\alpha$:

$$\textbf{FDR}_T := \frac{\sum_{t=1}^{T} \mathbb{1}\left(\hat{S}_t(\mathbf{x}_t) \neq \texttt{IDK} \wedge e_t = 1\right)}{\sum_{t=1}^{T} \mathbb{1}\left(\hat{S}_t(\mathbf{x}_t) \neq \texttt{IDK}\right)} \leq \alpha, \tag{2}$$

where $\textbf{FDR}_T = \alpha$ if $\hat{S}_t(\mathbf{x}_t) = \texttt{IDK}$ for all $t$. Then, the goal is to learn a distribution $p_t$ over selective generators such that a drawn selective generator $\hat{S}_t \sim p_t$ controls $\mathcal{R}_T^{\textbf{FDR}}$ close to zero, *i.e.,* $\sum_{t=1}^{T} \left[ \mathbb{1}(\hat{S}_t(\mathbf{x}_t) \neq \texttt{IDK} \wedge e_t = 1) - \alpha \mathbb{1}(\hat{S}_t(\mathbf{x}_t) \neq \texttt{IDK}) \right] \leq \varepsilon(T)$, where $\varepsilon(T)$ is some non-increasing function in $T$. Note that the learner controls $\mathcal{R}_T^{\textbf{FDR}}$ while the adaptive adversary tries to maximize it. Additionally, while FDR controllability is the primary objective, we also wish to minimize selection inefficiency $\textbf{Ineff}_T := \frac{1}{T} \sum_{t=1}^{T} \mathbb{1}(\hat{S}_t(\mathbf{x}_t) = \texttt{IDK})$.

## 3 EXSUL: ONLINE SELECTIVE GENERATION WITH FEEDBACK UNLOCKING

We leverage a regret perspective from bandit problems to design a learning algorithm for selective generation that controls the FDR under partial feedback. As a preliminary, we first introduce the necessary background on regret minimization in adversarial bandits (Section 3.1). We then reduce online selective generation problems to adversarial bandit problems with partial feedback (Section 3.2). This reduction enables us to leverage any regret minimization algorithms with their regret bounds for selective generation algorithms with FDR bounds. However, the connection between regret and FDR is missing. To fill this gap, we introduce a novel conversion lemma from the regret to the FDR (Section 3.3).

However, under partial feedback, simply leveraging existing bandit algorithms may not lead to sample efficiency without exploiting unique properties of selective generation. To address this, we propose a novel sample-efficient method for online selective generation under partial feedback. In particular, we extend the `Exp3-IX` algorithm (Neu, 2015) to learn selective generators, exploiting the unique structure of their selection functions by *feedback unlocking*, theoretically and empirically demonstrating its efficient regret bound (Section 3.4), which eventually leads to an FDR bound by using our conversion lemma. Note that online selective generation under full feedback can be devised in a similar way. See Appendix D.1 for the detail via exponential weighting (`EW`).

### 3.1 REGRET MINIMIZATION AND ADVERSARIAL BANDITS

The adversarial bandit problem (Auer et al., 2002) is formulated as an interactive game between a learner and an adversary. At each round $t \in 1, \ldots, T$, the *learner* selects an arm $\tau_t \in \mathcal{H}$ among a set of arms $\mathcal{H}$, while the *adversary* simultaneously chooses a loss function $\ell_t : \mathcal{H} \to \mathbb{R}_{\geq 0}$. The learner then observes only the incurred loss $\ell_t(\tau_t)$ as partial feedback. The learner's performance is evaluated by the *regret* (Bubeck et al., 2012), *i.e.,*

$$\textbf{Reg}_T := \sum_{t=1}^{T} \ell_t(\tau_t) - \min_{\tau \in \mathcal{H}} \sum_{t=1}^{T} \ell_t(\tau), \tag{3}$$

and the goal of the learner is to minimize $\textbf{Reg}_T$ so that it grows sublinearly in $T$. Here, the *adaptive adversary* maximizes $\textbf{Reg}_T$ by choosing $\ell_t$ based on the previously chosen arms by the learner, *i.e.,* $\tau_1, \ldots, \tau_{t-1}$. This makes $\textbf{Reg}_T$ a random variable, so we analyze high-probability or expected regret bounds. For further details on regret minimization and `Exp3-IX` (Neu, 2015), see Appendix B.3 and B.4.

### 3.2 REDUCTION: FROM ONLINE SELECTIVE GENERATION TO ADVERSARIAL BANDITS

Here, we map the components of online selective generation with partial feedback to those of adversarial bandits to leverage existing algorithms and regret bounds. In particular,

Table 1: From online selective generation to adversarial bandits

|  | online selective generation | adversarial bandits |
|---|---|---|
| models | selective generators $\mathcal{H}$ | finite arms $\mathcal{H}$ |
| feedback | $e_t$ | $\ell_t(\tau_t, \alpha)$ |
| metric | $\textbf{FDR}_T$ and $\textbf{Ineff}_T$ | $\textbf{Reg}_T$ |

a parameter $\tau \in \mathcal{H}$ of our
learner corresponds to a bandit arm in adversarial bandits.

For the feedback, we first define a special loss $\ell_t(\tau, \alpha)$ and connect this to $e_t$. In particular, we introduce the following two key components: $a_t(\tau) := \mathbb{1}(\hat{S}(\mathbf{x}_t; \tau) = \text{IDK})$, which measures the selection inefficiency at time $t$, called *inefficiency loss*, and $d_t(\tau, \alpha) := \mathbb{1}(\hat{S}(\mathbf{x}_t; \tau) \neq \text{IDK} \wedge e_t = 1) - \alpha \mathbb{1}(\hat{S}(\mathbf{x}_t; \tau) \neq \text{IDK}) + \alpha$, which measures the violation of the FDR risk at time $t$ with a margin by $\alpha$ to penalize the IDK response, called *FDR loss* with a margin. Based on these, we define the loss function for bandits as follows:

$$\ell_t(\tau, \alpha) := \frac{a_t(\tau) + \lambda d_t(\tau, \alpha)}{1 + \lambda} \in \left\{ 0, \frac{\lambda}{1 + \lambda}, \frac{1 + \lambda \alpha}{1 + \lambda} \right\} \in [0, 1], \tag{4}$$

where $\lambda$ is a hyperparameter that controls the trade-off between the FDR loss and the efficiency loss. Note that from the bandits' perspective, the learner receives feedback in two steps: get feedback $e_t$ from the adversary and uses this to compute the loss $\ell_t(\tau_t, \alpha)$ for the chosen arm $\tau_t$.

Finally, the link between two metrics for online selective generation and adversarial bandits remains to be established. In the following section, we propose a novel conversion lemma to connect $\mathbf{FDR}_T$ and $\mathbf{Ineff}_T$ to $\mathbf{Reg}_T$. See Figure 2 for an illustration of our full learning pipeline from online selective generation to adversarial bandits and back again.

### 3.3 REGRET-TO-FDR CONVERSION

We mainly leverage algorithms that minimize $\mathbf{Reg}_T$ for learning selective generators. Yet, the key requirement for selective generation is the FDR guarantee of a learner at a desired level. To this end, we introduce a novel perspective on the connection of a regret bound to an FDR bound. In particular, *any learner*, including learners for bandits, which minimizes the regret (3) with the loss (4), controls the FDR (1). This is achievable as the designed loss (4) penalizes when *FDR loss* at each $t$ is higher than $\alpha$, weighted by a parameter $\lambda$. See Appendix H.3 for a proof and details on choosing $\lambda$.

**Lemma 1.** *Let $T \in \mathbb{N}$ and $\alpha \in (0, 1)$. For any $(\mathbf{x}_t, \mathbf{y}_t)$ sequences, leading to any loss sequences $\ell_t$ of (4), we have*

$$\frac{1}{T} \mathcal{R}_T^{\textit{FDR}} \leq \frac{(1 - \textit{Ineff}_T) + (1 + \lambda)\textit{Reg}_T / T}{\lambda} = \frac{(1 - \textit{Ineff}_T)}{T^{1/4}} + \frac{(1 + T^{1/4})\textit{Reg}_T}{T^{5/4}}, \tag{5}$$

*where the last equality holds if we take $\lambda = T^{1/4}$.*

Importantly, this lemma is applicable to learning for both full and partial feedback, as it is agnostic to feedback mechanisms. Also, the lemma implies that if $\mathbf{Reg}_T$ has a sublinear bound, satisfied by most bandit learners, then the average $\mathcal{R}_T^{\mathbf{FDR}}$ gets closer to 0 at most the rate of $\mathcal{O}(1/T^{1/4})$, *i.e.*, $\mathcal{R}_T^{\mathbf{FDR}} \leq \mathcal{O}(T^{3/4})$.

**The Connection between $\mathcal{R}_T^{\mathbf{FDR}}$ and $\mathbf{FDR}_T$.** We can derive the bound for $\mathbf{FDR}_T$ by dividing (5) by $1 - \mathbf{Ineff}_T$ if $\mathbf{Ineff}_T \neq 1$, *i.e.*,

$$\mathbf{FDR}_T \leq \alpha + \frac{1}{T^{1/4}} + \frac{(1 + T^{1/4})\mathbf{Reg}_T}{T^{5/4}(1 - \mathbf{Ineff}_T)},$$

which remains sublinear in $T$ unless $\mathbf{Ineff}_T$ converges to 1. In other words, $\mathbf{FDR}_T$ may fail to be bounded if $1 - \mathbf{Ineff}_T \approx 0$ as $\sum_{t=1}^T e_t \approx T$, due to the nature of the empirical metric with the variance induced by randomized bandit algorithms. For instance, if the generator $G$ consistently produces incorrect outputs, the learner will try to converge to an always-abstaining solution. However, due to randomization, if the learner mistakenly accepts even a single sample in the early stage of learning, then regardless of convergence to an always-abstaining solution, the empirical FDR may still exceed the desired level $\alpha$. We highlight that this reflects an intrinsic limitation of the empirical FDR evaluation in sequential prediction. Thus, we consider the *FDR risk*, where our adaptive adversary assumes to maximize the $\mathcal{R}_T^{\mathbf{FDR}}$, where it does not consider this pathological case.

### 3.4 EXSUL: EXP3-IX FOR ONLINE SELECTIVE GENERATION WITH FEEDBACK UNLOCKING

As for the adversarial bandit learner, we leverage `Exp3-IX` (Neu, 2015), which is based on `EW` (Littlestone & Warmuth, 1994) for partial feedback. The key challenge from full to partial feedback is

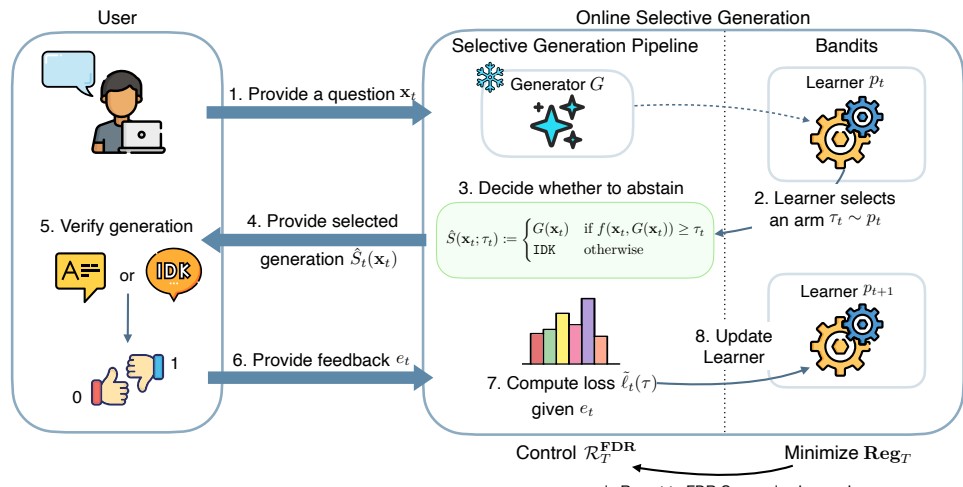

Figure 2: An example of our proposed framework for online selective generation. At each step $t$, (1) the user provides an input $\mathbf{x}_t$, (2) the learner selects an arm $\tau_t \sim p_t$, (3-4) selectively generates $\hat{S}_t(\mathbf{x}_t; \tau_t)$, (5-6) the user provides partial feedback $e_t$, (7) the loss $\ell_t(\tau_t)$ is computed from $e_t$, and (8) the learner update $p_t$ by the loss estimator $\tilde{\ell}_t$. Note that the user can be modeled as an adversary.

the slow convergence of learning due to the lack of feedback information. To address this, we exploit the unique loss function structure of selective generators (4) to unlock feedback of other arms by a chosen arm, dubbed by *feedback unlocking*. In particular, by the monotonicity of a selection function in $\tau$, *i.e.*, $\hat{s}(\mathbf{x}) := \mathbb{1}(f_t \geq \tau)$ where $f_t := f(\mathbf{x}_t, G(\mathbf{x}_t))$ and the definition of a selective generator $\hat{S}$, the following relation among feedback holds, where $e_t$ is observed feedback by a chosen arm $\tau_t$:

$$e_t = \mathbb{1}(\hat{S}(\mathbf{x}_t; \tau_t) \neq_E \mathbf{y}_t) = \begin{cases} \mathbb{1}(\hat{S}(\mathbf{x}_t; \tau) \neq_E \mathbf{y}_t) \text{ for } \tau \leq f_t & \text{if } \hat{S}(\mathbf{x}_t; \tau_t) \neq \texttt{IDK} \\ \mathbb{1}(\hat{S}(\mathbf{x}_t; \tau) \neq_E \mathbf{y}_t) \text{ for } \tau > f_t & \text{if } \hat{S}(\mathbf{x}_t; \tau_t) = \texttt{IDK} \end{cases}. \quad (6)$$

To exploit this feedback unlocking within $\texttt{Exp3-IX}$, we propose a novel loss estimator

$$\tilde{\ell}_t(\tau, \alpha \mid \mathcal{H}_t(\tau_t)) := \frac{\ell_t(\tau, \alpha)}{\gamma_t + \sum_{\bar{\tau} \in \mathcal{H}_t(\tau_t)} \mathbb{1}(\tau \in \mathcal{H}_t(\bar{\tau})) \cdot p_t(\bar{\tau})} \cdot \mathbb{1}(\tau \in \mathcal{H}_t(\tau_t)) \in [0, \infty), \quad (7)$$

where $\gamma_t > 0$, $\mathcal{H}_t(\tau_t) := \{\tau \in \mathcal{H} \mid \tau \leq f_t\}$ if $\hat{S}(\mathbf{x}_t; \tau_t) \neq \texttt{IDK}$ and $\mathcal{H}_t(\tau_t) := \{\tau \in \mathcal{H} \mid \tau > f_t\}$ otherwise – see Figure 33 for the visualization of $\mathcal{H}_t(\tau)$. With learning rate $\eta_t > 0$, Algorithm 1 presents our fixed-parameter extension of $\texttt{Exp3-IX}$ with partial feedback unlocking. Note that our algorithm can be reduced to $\texttt{Exp3-IX}$ by letting $\mathcal{H}_t(\tau) = \{\tau_t\}$ for any $\tau$, recovering the $\texttt{Exp3-IX}$ estimator in (9) from our estimator (7). Also, our algorithm does not use $f_t$ to choose $\tau_t$.

**Regret Bound.** Based on the original proof of $\texttt{Exp3-IX}$ but along with our novel loss estimator (7), our Algorithm 1 achieves a sublinear regret bound, which leads to faster convergence of the FDR guarantee by Lemma 1. See Appendix H.4 for a general proof on our regret bound.

**Theorem 1.** *Let $\ell_t(\cdot) \in [0, 1]$ of the form (4). For any $T \in \mathbb{N}$ and finite hypotheses $\mathcal{H}$, Algorithm 1 provides the following regret bound with probability at least $1 - \delta$, if $\eta = 2\gamma = \sqrt{\frac{\ln |\mathcal{H}|}{T}}$:*

$$\boldsymbol{Reg}_T \leq 4\sqrt{T \ln |\mathcal{H}|} + \left(1 + \sqrt{\frac{T}{\ln |\mathcal{H}|}}\right) \ln \frac{2}{\delta}.$$

Interestingly, despite of partial feedback, our algorithm achieves the same upper bound as $\texttt{EW}$ with full feedback, *i.e.*, $\sqrt{T \ln |\mathcal{H}|/2}$, up to a constant factor due to our loss estimator with feedback unlocking (7) for rich information, whereas $\texttt{Exp3-IX}$ suffers from an additional factor of $\sqrt{|\mathcal{H}|}$, as in (10), due to its loss estimator with limited information.

Re

---

**Algorithm 1** `Exp3-IX` for Online Selective Generation with Feedback Unlocking (`ExSUL`)

---

1: **procedure** ExSUL($T, \mathcal{H}, \alpha, \lambda, \eta, f, G, \gamma$)
2:     $w_1(\tau) \leftarrow 1/|\mathcal{H}|$ for all $\tau \in \mathcal{H}$
3:     **for** $t = 1, \ldots, T$ **do**
4:         Choose $\tau_t \sim p_t(\tau) = \sum_{\tau \in \mathcal{H}} \delta(\tau) \cdot w_t(\tau) / \sum_{\tau \in \mathcal{H}} w_t(\tau)$
5:         Observe $\mathbf{x}_t$
6:         Observe $e_t$         ($\triangleright$) partial feedback $e_t := \mathbb{1}(\hat{S}(\mathbf{x}_t; \tau_t) \neq_E \mathbf{y}_t)$
7:         $\tilde{\ell}_t(\tau, \alpha) \leftarrow$ UNLOCKING($\mathcal{H}, \alpha, \lambda, \gamma, \tau_t, \mathbf{x}_t, e_t$)
8:         Update $w_{t+1}(\tau) \propto \exp\left(-\eta \sum_{s=1}^{t} \tilde{\ell}_t(\tau, \alpha)\right)$         ($\triangleright$) weight update

9: **procedure** UNLOCKING($\mathcal{H}, \alpha, \lambda, \gamma, \tau_t, \mathbf{x}_t, e_t$)
10:     $f_t \leftarrow f(\mathbf{x}_t, G(\mathbf{x}_t))$
11:     $\mathcal{H}_t(\tau_t) \leftarrow \begin{cases} \{\tau \in \mathcal{H} \mid \tau \leq f_t\} & \text{if } \hat{S}(\mathbf{x}_t; \tau_t) \neq \text{IDK}, \\ \{\tau \in \mathcal{H} \mid \tau > f_t\} & \text{otherwise.} \end{cases}$
12:     $\ell_t(\tau, \alpha) \leftarrow$ COMPUTELOSS($\hat{S}(\mathbf{x}_t; \tau), e_t, \alpha, \lambda$) for all $\tau \in \mathcal{H}_t(\tau_t)$     ($\triangleright$) Algorithm 2
13:     $\ell_t(\tau, \alpha \mid \mathcal{H}_t(\tau_t)) \leftarrow \frac{\ell_t(\tau, \alpha)}{\gamma + \sum_{\bar{\tau} \in \mathcal{H}_t(\tau_t)} \mathbb{1}(\tau \in \mathcal{H}_t(\bar{\tau})) \cdot p_t(\bar{\tau})} \cdot \mathbb{1}(\tau \in \mathcal{H}_t(\tau_t))$ for all $\tau \in \mathcal{H}$
                                               ($\triangleright$) Loss estimation for all hypotheses
14:     **return** $\ell_t(\tau, \alpha \mid \mathcal{H}_t(\tau_t))$

---

# 4 EXPERIMENTS

We empirically justify that `ExSUL` controls the FDR while maximizing selection efficiency under three diverse environments:(1) stochastic, (2) distribution-shift, and (3) interactive environments.

**Datasets and Models.** We use two datasets for stochastic and distribution-shift environments, 79K Natural Question (NQ) (Kwiatkowski et al., 2019) and 93K TriviaQA (Joshi et al., 2017) with two base models, GPT-3.5-turbo (OpenAI, 2023) and LLaMA3.1-8B-Instruct (Meta AI, 2024). We simulate distribution-shift environments by mixing two datasets in diverse ways and an interactive environment by generating interaction between GPT-3.5-turbo and GPT-4o models (see Section F.2 and F.3, respectively for details). To obtain feedback and self-consistency scores, we use GPT-3.5-turbo.

**Scoring functions.** We consider two scoring functions $f_{\text{std}}$ and $f_{\text{con}}$, introduced in Section B.1. In short, $f_{\text{std}}$ is a standard log-likelihood score defined as the conditional probability of an answer given a question. $f_{\text{con}}$ is a self-consistency score of an answer computed via entailment scores (from an entailment model) across samples. We use $f_{\text{con}}$ unless specified.

**Methods.** We consider two baselines, `Exp3-IX-SG` and `No-SG`, and one performance upper bound algorithm, `EW-SG`. In particular, (1) `Exp3-IX-SG` (Algorithm 6) is an online selective generator with partial feedback by adapting `Exp3-IX` (Neu, 2015) via our regret-to-FDR conversion lemma. (2) `No-SG` is a non-selective generator (*i.e.*, $\tau_t = 0$ for all $t$), to serve as a standard use of a generator without abstention. (3) `EW-SG` (Algorithm 5) is an online selective generator with full feedback. As it exploits full feedback, we use it as our empirical performance upper bound.

**Metrics.** We measure performance of ours along with comparing methods via the empirical FDR, *i.e.*, **FDR**$_t$ and selection inefficiency, *i.e.*, **Ineff**$_t$. Note that `No-SG` achieves zero selection inefficiency by definition, so we do not explicitly add in figures.

**Parameters.** We set $|\mathcal{H}| = 1K$ and $\lambda = T^{1/4}$ by default. Also, a desired FDR level $\alpha$ and time horizon $T$ depend on tasks, where $\alpha$ is provided by users.

## 4.1 STOCHASTIC ENVIRONMENT

In Figure 3 and 8, we observe that `ExSUL` has clearly better convergence speed to the desired FDR compared to `Exp3-IX-SG`, since `Exp3-IX-SG` only observes $\ell_t(\tau_t)$, requiring a much longer horizon $T$ (Figure 24). These results demonstrate that our partial feedback unlocking is beneficial in a faster convergence to the desired FDR, supporting the regret bound analysis discussed in Section 3.4.

In addition, Figure 22 shows that our algorithm effectively controls the FDR across different $\alpha$. Moreover, Figure 28 indicates that the inefficiency loss $a_t$ enables the algorithm to achieve efficient

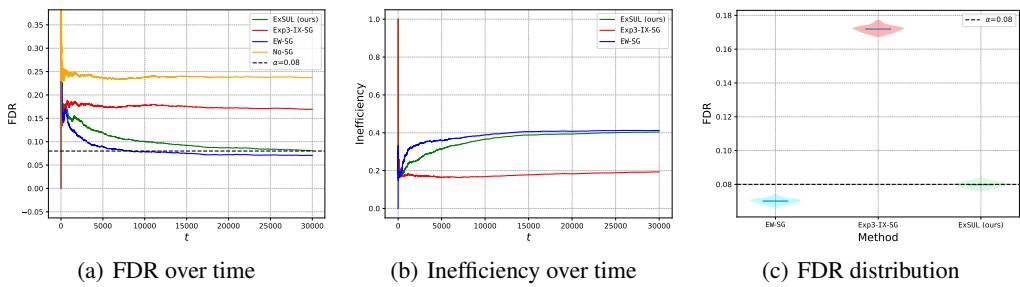

(a) FDR over time      (b) Inefficiency over time      (c) FDR distribution

Figure 3: Comparison of selective generation methods under a stochastic environment with LLaMA3.1-8B-Instruct as a generator on TriviaQA ($T = 30K, \alpha = 0.08$). The violin plots are drawn with randomly chosen 30K samples over 100 random trials.

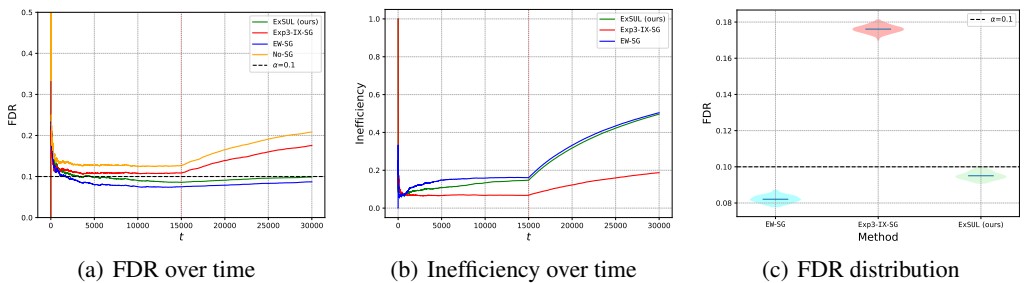

(a) FDR over time      (b) Inefficiency over time      (c) FDR distribution

Figure 4: Comparison of selective generation methods under a single distribution-shift environment with GPT-3.5-turbo as a generator ($T = 30K, \alpha = 0.1$), from TriviaQA to NQ. The violin plots are drawn with randomly chosen 30K samples with 100 random trials.

**FDR guarantees.** Note that algorithms with $f_{\mathtt{con}}$ have better controllability over the FDR than those with $f_{\mathtt{std}}$ (*e.g.,* Figure 10 v.s. 12), suggesting that a well-calibrated scoring function can have a positive effect on convergence speed. See additional results in Section G.1.

### 4.2 DISTRIBUTION-SHIFT ENVIRONMENT

To evaluate the robustness of the methods, we consider three distribution-shift scenarios: (1) a single instantaneous change by concatenating two datasets, (2) frequent shifts by interleaving fixed-size chunks randomly drawn from each dataset, and (3) progressive shifts by sampling each example according to a linearly increasing mixing probability. See Appendix F.2 for details.

Interestingly, in Figure 4, 5, and 6, `Exp3-IX-SG` exhibits a sharp increase in FDR immediately following the distribution shift, highlighting its sensitivity to sudden changes in the data distribution, while the proposed `ExSUL` does not. In addition, Figure 23 shows that our algorithm still effectively controls the FDR across different $\alpha$ in the distribution shift environment. See additional results on varying generators and shift types in Section G.2 and a longer time horizon in Figure 25.

### 4.3 INTERACTIVE ENVIRONMENT

We empirically demonstrate the efficacy of `ExSUL` in real-world interactive applications. In particular, we simulate an interactive environment by implementing a user-acting agent, a question-answering agent, and an evaluating agent by states, where these agents interact over multiple turns, and `ExSUL` decides whether to abstain from answering as shown in Figure 1.

In Figure 7, we observe that `ExSUL` consistently controls the FDR under $\alpha$, despite the instability of **FDR**$_t$ demonstrating the robustness of `ExSUL` under frequently shifting distributions over time. In Figure 1, `ExSUL` successfully abstains from incorrect answers, achieving the desired FDR level $\alpha$.

## 5 CONCLUSION

We propose a learning algorithm for online selective generation with partial feedback, which provides an FDR controllability guarantee, while allowing $\lambda$ to adjust the trade-off between the selection efficiency and convergence speed. To this end, we leverage the well-developed adversarial ban-

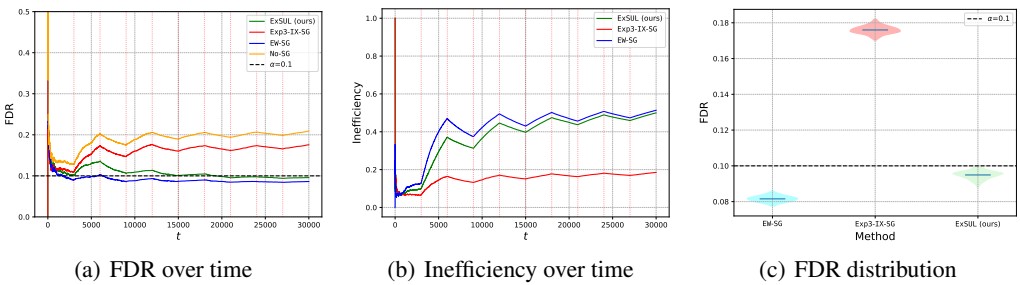

(a) FDR over time    (b) Inefficiency over time    (c) FDR distribution

Figure 5: Comparison of selective generation methods under an alternating distribution-shift environment with GPT-3.5-turbo as a generator ($T = 30K, \alpha = 0.1$), alternating between TriviaQA and NQ, starting with TriviaQA. The violin plots are drawn with randomly chosen 30K samples with 100 random trials.

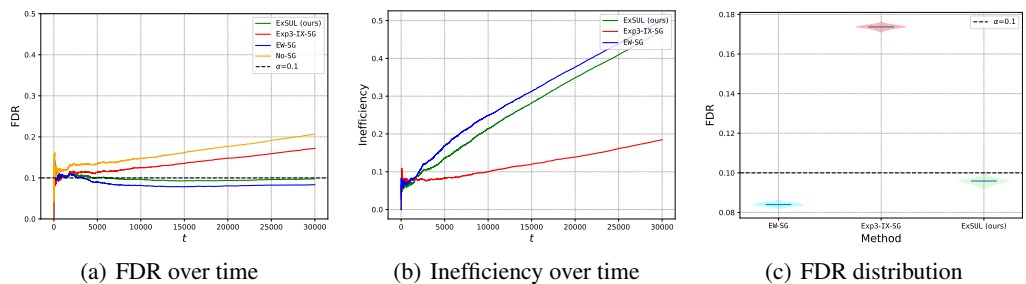

(a) FDR over time    (b) Inefficiency over time    (c) FDR distribution

Figure 6: Comparison of selective generation methods under a gradual distribution-shift environment with GPT-3.5-turbo as a generator ($T = 30K, \alpha = 0.1$), from TriviaQA to NQ over time. The violin plots are drawn with randomly chosen 30K samples with 100 random trials.

dit problem. In particular, we reduce selective generation to adversarial bandits and extend the `Exp3-IX` algorithm for adversarial bandits by *partial feedback unlocking*, which addresses the lack of feedback information due to the nature of partial feedback. Finally, we introduce a novel *Regret-to-FDR conversion lemma* such that we can use any regret minimization algorithm, including our extended `Exp3-IX` for selective generation to control an FDR at a desired level. We theoretically and empirically justify our method by providing an efficient regret bound and evaluating diverse learning environments, including stochastic, distribution-shift, and interactive environments.

**Limitations.** The main limitation of our method is that our online selective generator selects a hypothesis independently of the observed input $\mathbf{x}_t$. As future work, extending to the contextual bandits, where the input is taken into account when pulling an arm, would be a promising direction.

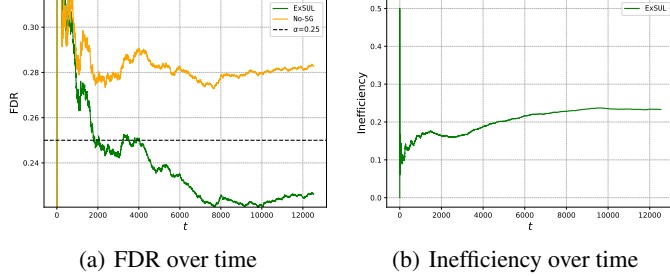

(a) FDR over time       (b) Inefficiency over time

Figure 7: $\mathbf{FDR}_t$ and $\mathbf{Ineff}_t$ for `ExSUL` under an interactive environment with GPT-3.5-Turbo as a question-answering agent ($T = 12{,}500$, $\alpha = 0.25$).

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

## A  LLM USAGE

LLMs were used for minor grammar suggestions, and for assistance with code implementation.

## B  PRELIMINARY

We introduce preliminaries on language generation, selective prediction, and adversarial bandits. To this end, let $\mathcal{X}$ be a set of inputs (*e.g.,* examples or questions) and $\mathcal{Y}$ be a set of outputs (*e.g.,* labels or answers).

### B.1  LANGUAGE GENERATION

We mainly consider language generators as our model to control hallucination. In particular, let $G : \mathcal{X} \to \mathcal{Y}$ be a language generator, where $\mathcal{W}$ is a set of tokens and $\mathcal{X} = \mathcal{Y} := \cup_{i=0}^{\infty} \mathcal{W}^i$. Here, each $i$-th token $\hat{\mathbf{y}}_i$ of a generated answer $\hat{\mathbf{y}} \in \mathcal{Y}$ is decoded from an underlying probability distribution $p(\mathbf{y} \mid \mathbf{x})$, where $p$ is usually learned via language data. For the decoding strategy, we consider greedy decoding, *i.e.,* $\hat{\mathbf{y}}_i = \arg\max_{w \in \mathcal{W}} p(w \mid \mathbf{x}, \hat{\mathbf{y}}_{1:i-1})$, where $\mathbf{y}_{a:b} := (\mathbf{y}_a, \ldots, \mathbf{y}_b)$. Given a generated answer $\hat{\mathbf{y}}$, there are multiple ways to measure its likelihood of correctness $f : \mathcal{X} \times \mathcal{Y} \to \mathbb{R}$. A standard way $f_{\mathtt{std}}$ considers a length-normalized token probability, *i.e.,* $f_{\mathtt{std}}(\mathbf{x}, \hat{\mathbf{y}}) := (p(\hat{\mathbf{y}}_1 \mid \mathbf{x}) \prod_{i=2}^{|\hat{\mathbf{y}}|} p(\hat{\mathbf{y}}_i \mid \mathbf{x}, \hat{\mathbf{y}}_{1:i-1}))^{1/|\hat{\mathbf{y}}|}$. A better alternative is to consider consistency of $\hat{\mathbf{y}}$ to multiple generated answers with sampling (Manakul et al., 2023), *i.e.,* $f_{\mathtt{con}}(\mathbf{x}, \hat{\mathbf{y}}) = \hat{\mathbb{E}}_{\mathbf{y}' \sim G(x)} f_E(\mathbf{y}', \hat{\mathbf{y}})$, denoted by a consistency score, where we use sampling for decoding $\mathbf{y}'$, $f_E(\mathbf{y}', \hat{\mathbf{y}})$ measures an entailment score of $\hat{\mathbf{y}}$ given $\mathbf{y}'$ (*i.e.,* whether $\mathbf{y}'$ entails $\hat{\mathbf{y}}$) via an entailment model (*e.g.,* GPT-3.5-turbo).

### B.2  SELECTIVE PREDICTION

Selective prediction (Geifman & El-Yaniv, 2017) and generation (Lee et al., 2024) provide certified control over the risk of incorrect predictions by abstaining answers (saying IDK) when the prediction is uncertain. Given a predictor $\hat{\mathbf{y}} : \mathcal{X} \to \mathcal{Y}$, a selective predictor $\hat{S} : \mathcal{X} \to \mathcal{Y} \cup \{\text{IDK}\}$ abstains from returning $\hat{\mathbf{y}}(\mathbf{x})$ if a selection function $\hat{s} : \mathcal{X} \times \mathcal{Y} \to \{0, 1\}$ deems the prediction uncertain, *i.e.,* $\hat{S}(\mathbf{x}) := \begin{cases} \hat{\mathbf{y}}(\mathbf{x}) & \text{if } \hat{s}(\mathbf{x}, \hat{\mathbf{y}}(\mathbf{x})) = 1 \\ \text{IDK} & \text{otherwise} \end{cases}$. In learning, the selection function $\hat{s}$ is chosen to possibly satisfy a desired level of a false discovery rate *i.e.,* $\mathbb{P}(\hat{S}(\mathbf{x}) \neq_E \mathbf{y} \mid \hat{S}(\mathbf{x}) \neq \text{IDK})$ over the i.i.d. samples of $(\mathbf{x}, \mathbf{y})$. In selective classification (Geifman & El-Yaniv, 2017), the equality relation $\neq_E$ is usually the standard exact matching, which is extended to capture semantic correctness via entailment relation in selective generation (Lee et al., 2024). In contrast to the standard stochastic setup as in the previous literature, we consider online learning under partial feedback.

### B.3  REGRET MINIMIZATION

Sequential prediction in multi-armed bandit problems is formulated as an interactive game between a learner and an adversary (also called an environment). In particular, for each time $t \in \{1, \ldots, T\}$, a learner selects an arm $\tau_t \in \mathcal{H}$ among a set of arms $\mathcal{H}$ and an adversary selects a loss function $\ell_t : \mathcal{H} \to \mathbb{R}_{\geq 0}$, where a loss associated to an arm $\tau_t$ is denoted by $\ell_t(\tau_t)$ and provided to the learner as partial feedback (*i.e.,* only the loss for the selected arm is provided). Note that this interactive game often describes in rewards, but we consider losses, instead of rewards.

The main goal of this interactive game is to find a learner whose performance is as good as the best learner in hindsight. Here, the performance is measured by *regret*, following standard bandits literatures (Bubeck et al., 2012), *i.e.,*

$$\mathbf{Reg}_T := \sum_{t=1}^{T} \ell_t(\tau_t) - \min_{\tau \in \mathcal{H}} \sum_{t=1}^{T} \ell_t(\tau). \tag{8}$$

Thus, the the goal becomes to find a learner that minimizes the regret such that an achieved regret bound is sublinear in $T$. Here, we say that the adversary is *oblivious* if the sequence of loss functions are chosen regardless of learner's previous selections (or chosen in advance before the game), and the adversary is *adaptive* if the loss function at time $t$, *i.e.,* $\ell_t$, is drawn a distribution that depends on the previously chosen arms, *i.e.,* $\tau_1, \ldots, \tau_{t-1}$. In this paper, we consider the adaptive adversary which is more stronger than the oblivious one. Moreover, under the adaptive adversary setup, a deterministic learner cannot win the game (*i.e.,* it does not achieve a sub-linear regret bound), so we consider a randomized learner. This implies that $\tau_t$ and $\ell_t$ are two sources of randomness, so $\mathbf{Reg}_T$ is a random

variable. Thus, we consider the high-probable regret bound or a bound of an expected regret $\mathbb{E}\mathbf{Reg}_T$. Note that we only consider the learner's randomization in the analysis, since our bounds are derived to hold for any adversary (Bubeck et al., 2012; Neu, 2015).

In the following section, we review one regret minimization algorithm for bandits with an adaptive adversary, called `Exp3-IX`, and see Appendix E for other algorithms.

### B.4 ADVERSARIAL BANDITS AND EXP3-IX ALGORITHM

Traditional online learning typically assumes full feedback, *i.e.,* loss for any hypothesis is computable. However, practical applications may only have partial feedback, *i.e.,* loss for a chosen hypothesis is only given. We model this via the adversarial multi-armed bandit problem. The regret is as in online learning, but the learner only receives for a chosen arm, while the sequence $\ell_t(\tau_t)$ is chosen by an oblivious or adaptive adversary that may condition on the history of chosen arms but not on the learner's fresh randomness at time $t$.

To obtain the high-probability control of the $\mathbf{Reg}_T$ over an adaptive adversary, Exponential-weights for Exploration and Exploitation with Implicit Exploration (`Exp3-IX`)(Neu, 2015) is a natural choice. It employs implicit-exploration (IX) loss estimates within an exponential-weights scheme and does not require explicit mixing with the uniform distribution like `Exp3.P` (Auer et al., 2002). In particular, `Exp3-IX` uses the variance-reduced loss estimator for each arm $\tau \in \mathcal{H}$ constructed from the observed loss of the chosen arm $\tau_t$, *i.e.,*

$$\tilde{\ell}_t(\tau \mid \{\tau_t\}) \coloneqq \frac{\ell_t(\tau)}{\gamma_t + p_t(\tau)} \cdot \mathbb{1}(\tau_t = \tau) \in [0, \infty). \tag{9}$$

where $\gamma_t > 0$ is properly selected by an analysis; thus by using (9) in $\mathbf{Reg}_T$, with non-increasing learning rate $\eta_t$, we have the following regret with high probability for `Exp3-IX` . See AppendixH.2 for a complete proof.

**Theorem 2.** *(Neu, 2015) Let $\ell_t(\cdot) \in [0, \ell_{max}]$. For any $T \in \mathbb{N}$, finite arms $\mathcal{H}$, and $\delta \in (0, 1)$, Algorithm 4 provides the following regret bound with probability at least$1 - \delta$ if $\eta_t = \frac{2\gamma_t}{\ell_{max}} = \sqrt{2 \ln |\mathcal{H}| / (\ell_{\max}^2 T |\mathcal{H}|)}$:*

$$\mathbf{Reg}_T \le \ell_{max} \left( 2\sqrt{2T|\mathcal{H}| \ln |\mathcal{H}|} + \left( 1 + \sqrt{\frac{T|\mathcal{H}|}{2 \ln |\mathcal{H}|}} \right) \ln \delta^{-1} \right). \tag{10}$$

We will exploit `Exp3-IX` for learning selective generators with partial feedback under an adaptive adversary. See Appendix D for other online learning and bandit algorithms.

### B.5 ONLINE LEARNING AND EW ALGORITHM

Online learning considers to design a learner that adapts to any sequence of full feedback on hypotheses without any or with little assumption on data generation process. In particular, the goal of an online learner is to find a distribution $p_t$ over hypotheses with full feedback that minimizes the regret.

Along with many online learning algorithms, we introduce Exponential Weighting (`EW`) (Littlestone & Warmuth, 1994), which minimizes the regret with finite hypotheses (Algorithm 3). In particular, `EW` maintains the goodness of weights for each hypothesis in terms of cumulative loss and updates them from feedback. The feedback of a hypothesis $\tau_t$ at time $t$ is represented by loss $\ell_t(\tau_t)$ but we can compute this loss for all hypotheses other than a chosen hypothesis $\tau_t$ as we directly observe a true label $\mathbb{1}(G(\mathbf{x}_t) =_E \mathbf{y}_t)$ at the same time $t$. This `EW` algorithm satisfies the following regret bound. See Appendix H.1 for a proof.

**Theorem 3.** *(Littlestone & Warmuth, 1994; Foster & Rakhlin, 2023) Let $\ell_t(\cdot) \in [0, \ell_{max}]$. For any $T \in \mathbb{N}$ and finite hypotheses $\mathcal{H}$, Algorithm 3 provides the following expected regret bound if $\eta = \sqrt{8 \ln |\mathcal{H}| / (\ell_{max}^2 T)}$:*

$$\sum_{t=1}^{T} \mathbb{E}_{\tau_t \sim p_t} \ell_t(\tau_t) - \min_{\tau \in \mathcal{H}} \sum_{t=1}^{T} \ell_t(\tau) \le \ell_{max} \sqrt{T \ln |\mathcal{H}| / 2} = \mathcal{O}(\ell_{max} \sqrt{T \ln |\mathcal{H}|}), \tag{11}$$

*which holds for arbitrary loss sequences, thus, implying that $\mathbb{E}\textbf{Reg}_T$ is also bounded by $\mathcal{O}(\ell_{max}\sqrt{T\ln|\mathcal{H}|})$ as the linearity of expectation and Tower property. Moreover, for any $0 < \delta < 1$, the following regret bound also holds:*

$$\sum_{t=1}^{T}\ell_t(\tau_t) - \min_{\tau\in\mathcal{H}}\sum_{t=1}^{T}\ell_t(\tau) \leq \ell_{max}\sqrt{\frac{T\ln|\mathcal{H}|}{2}} + \ell_{max}\sqrt{\frac{T\ln\delta^{-1}}{2}},$$

*with probability at least $1 - \delta$.*

We will leverage and extend this `EW` for learning selective generators with full feedback.

## C ADDITIONAL RELATED WORK

**Constrained Online Learning** Several works have studied online learning with constraints. (Mahdavi et al., 2012) first proposes long-term constraints in online convex optimization with time-invariant constraints, which allows to have the constraint function $g_t$ to be violated at each time step, *i.e.,* $g(\mathbf{x}_t) > 0$, but eventually satisfied, *i.e.,* $\sum_{t=1}^{T} g(\mathbf{x}_t) \leq 0$. Some other works solve a similar problem but with time-varying constraints, which is more challenging. (Chen et al., 2017; Sid-Ali et al., 2024) solve a network resource allocation task by viewing this as an constrained optimization problem. Especially, (Sid-Ali et al., 2024) solves the problem without a convex assumption on loss. (Sinha & Vaze, 2024) gives analysis on optimal algorithms for online convex optimization with adversarial constraints. We similarly consider a FDR guarantee to be satisfied over time but with partial feedback.

**Conformal Prediction with Bandit Feedback** There is little work about selective or conformal prediction in bandit settings. (Ge et al., 2024) proposes an online conformal prediction method with semi-bandit feedback, where we can observe the true label only if it is predicted. Our method follows the practical motivation in having partial feedback idea but we consider selective generation.

## D ADDITIONAL METHOD

### D.1 ONLINE SELECTIVE GENERATION WITH FULL FEEDBACK

Here, we specifically leverage the conventional exponential weighting (`EW`) algorithm (Littlestone & Warmuth, 1994) for online learning under adaptive adversaries to re-purpose it for addressing online selective generation with full feedback. This is an oracle method with the performance upper bound of our method with partial feedback. To this end, we provide the reduction of online selective generation with full feedback to the online learning problem. Then, we introduce a modified `EW` algorithm for selective generation, called `EW-SG`, followed by its regret bound. Finally, we leverage the proposed conversion lemma, which converts the regret bound of any online learning algorithms into the FDR bound, to show the FDR controllability of `EW-SG` at a desired level.

### D.1.1 REDUCTION: FROM ONLINE SELECTIVE GENERATION TO ONLINE LEARNING

Table 2: From online selective generation to online learning

|  | online selective generation | online learning |
|---|---|---|
| models | selective generators $\mathcal{H}$ | hypotheses $\mathcal{H}$ |
| feedback | $\mathbb{1}(G(\mathbf{x}_t) \neq_E \mathbf{y}_t)$ | $\ell_t(\tau, \alpha)$ |
| metric | $\textbf{FDR}_T$ and $\textbf{Ineff}_T$ | $\textbf{Reg}_T$ |

We convert online selective generation with full feedback to conventional online learning (See Table 2 for a summary). In selective generation, models are defined as a set of selective generators $\hat{S}$ or equivalently a set of selection parameters $\mathcal{H}$, considered as hypotheses $\mathcal{H}$ in online learning. This model is sequentially updated by leveraging online feedback. In particular, we consider feedback whether a generated answer $G(\mathbf{x}_t)$ is semantically incorrect with respect to a true answer $\mathbf{y}_t$,

*i.e.*, $\mathbb{1}(G(\mathbf{x}_t) \neq_E \mathbf{y}_t)$. This essentially provides full feedback for any $\tau \in \mathcal{H}$ as it is the same as $\mathbb{1}(\hat{S}(\mathbf{x}_t; \tau) \neq_E \mathbf{y}_t)$ by the definition of $\hat{S}$ if $\hat{S}(\mathbf{x}_t; \tau) \neq \texttt{IDK}$. This feedback is converted into loss in online learning, as defined in (4). Note that $d_t$ in the loss leverages the feedback $\mathbb{1}(G(\mathbf{x}_t) \neq_E \mathbf{y}_t)$ by the definition. Finally, our goal of finding a learning algorithm, which controls the FDR by $\alpha$ and minimizes selection inefficiency in online selective generation, is reduced to finding an online learning algorithm that minimizes the regret $\mathbf{Reg}_T$ with any loss sequences $\ell_t(\tau, \alpha)$ in (4).

### D.1.2 ALGORITHM AND ITS REGRET BOUND

We re-purpose `EW` for selective generation called `EW-SG`. This mainly uses the special loss $\ell_t(\tau, \alpha)$ in (4) for selective generation. See Algorithm 5 for details. The corresponding regret bound is directly obtained from Theorem 3 where $\ell_{\max} := \max(\lambda, 1 + \lambda\alpha) \leq 1 + \lambda$.

### D.2 REGRET-TO-FDR CONVERSION

Once we devise a regret minimization algorithm, *i.e.,* `EW-SG`, the regret guarantee is simply interpreted as the FDR guarantee by Lemma 1. This eventually provides the FDR bound for `EW-SG`.

## E ALGORITHMS

---

**Algorithm 2** Procedure for Computing the Loss (4) with Partial Feedback

---

1: **procedure** COMPUTELOSS($\hat{\mathbf{s}}, e, \alpha, \lambda$)
2:    $a \leftarrow \mathbb{1}(\hat{\mathbf{s}} = \texttt{IDK})$
3:    $d \leftarrow \mathbb{1}(\hat{\mathbf{s}} \neq \texttt{IDK}) \cdot e - \alpha\, \mathbb{1}(\hat{\mathbf{s}} \neq \texttt{IDK}) + \alpha$
4:    $\ell \leftarrow \frac{a + \lambda d}{1 + \lambda}$
5:    **return** $\ell$

---

**Algorithm 3** Exponential Weighting (`EW`) (Littlestone & Warmuth, 1994; Foster & Rakhlin, 2023)

---

1: **procedure** EW($T, \mathcal{H}, \eta$)
2:    $w_1 \leftarrow (1/|\mathcal{H}|, \ldots, 1/|\mathcal{H}|)$
3:    **for** $t = 1, \ldots, T$ **do**
4:        Observe $\mathbf{x}_t$
5:        Predict $\hat{\mathbf{y}}(\mathbf{x}_t; \tau_t)$ for $\tau_t \sim p_t(\tau) = \sum_{\tau \in \mathcal{H}} \delta(\tau) \cdot w_t(\tau) / \sum_{\tau \in \mathcal{H}} w_t(\tau)$
6:        Observe $\mathbf{y}_t$
7:        Update $w_{t+1}(\tau) \propto w_t(\tau) \exp(-\eta\ell(\mathbf{y}_t, \hat{\mathbf{y}}(\mathbf{x}_t; \tau)))$ for all $\tau \in \mathcal{H}$

---

**Algorithm 4** Exp3 with Implicit eXploration (`Exp3-IX`) (Neu, 2015)

---

1: **procedure** EXP3-IX($T, \mathcal{H}, \eta_t, \gamma_t$)
2:    $w_1 \leftarrow (1/|\mathcal{H}|, \ldots, 1/|\mathcal{H}|)$
3:    **for** $t = 1, \ldots, T$ **do**
4:        Choose an arm $\tau_t \sim p_t(\tau) = \sum_{\tau \in \mathcal{H}} \delta(\tau) \cdot w_t(\tau) / \sum_{\tau \in \mathcal{H}} w_t(\tau)$
5:        Observe $\ell_t(\tau_t)$                  ($\triangleright$) Observe loss only for a chosen arm $\tau_t$
6:        $\hat{\ell}_t(\tau) \leftarrow \frac{\ell_t(\tau_t)}{\gamma_t + p_t(\tau)} \mathbb{1}(\tau_t = \tau)$ for all $\tau \in \mathcal{H}$       ($\triangleright$) Estimate loss for all arms
7:        Update $w_{t+1}(\tau) \propto \exp(-\eta_t \sum_{s=1}^{t} \hat{\ell}_t(\tau))$ for all $\tau \in \mathcal{H}$

---

---

**Algorithm 5** Exponential Weighting for Online Selective Generation (`EW-SG`)

---

1: **procedure** EW-SG$(T, \mathcal{H}, \alpha, \lambda, \eta)$
2: $\quad w_1(\tau) \leftarrow 1/|\mathcal{H}|$ for all $\tau \in \mathcal{H}$
3: $\quad$ **for** $t = 1, \ldots, T$ **do**
4: $\quad\quad$ Observe $\mathbf{x}_t$
5: $\quad\quad$ Predict $\hat{S}_t(\mathbf{x}_t; \tau_t)$ where $\tau_t \sim p_t = \sum_{\tau \in \mathcal{H}} w_t(\tau) \cdot \delta(\tau)$
6: $\quad\quad$ Observe $\mathbf{y}_t$ thus observe $\mathbb{1}(G(\mathbf{x}_t) \neq_E \mathbf{y}_t)$
7: $\quad\quad a_t(\tau) \leftarrow \mathbb{1}(\hat{S}(\mathbf{x}_t; \tau) = \texttt{IDK})$
8: $\quad\quad d_t(\tau, \alpha) \leftarrow \mathbb{1}(\hat{S}(\mathbf{x}_t; \tau) \neq \texttt{IDK} \wedge G(\mathbf{x}_t) \neq_E \mathbf{y}_t) - \alpha \mathbb{1}(\hat{S}(\mathbf{x}_t; \tau) \neq \texttt{IDK}) + \alpha$
9: $\quad\quad \ell_t(\tau, \alpha) \leftarrow \frac{a_t(\tau) + \lambda d_t(\tau, \alpha)}{1+\lambda}$ for all $\tau \in \mathcal{H}$
10: $\quad\quad$ Update $w_{t+1}(\tau) \propto w_t(\tau) \exp\{-\eta \cdot \ell_t(\tau, \alpha)\}$ for all $\tau \in \mathcal{H}$

---

**Algorithm 6** `Exp3-IX` for Online Selective Generation with Partial Feedback (`Exp3-IX-SG`)

---

1: **procedure** EXP3-IX-SG$(T, \mathcal{H}, \alpha, \lambda, \eta, \gamma)$
2: $\quad w_1(\tau) \leftarrow 1/|\mathcal{H}|$ for all $\tau \in \mathcal{H}$
3: $\quad$ **for** $t = 1, \ldots, T$ **do**
4: $\quad\quad$ Observe $\mathbf{x}_t$
5: $\quad\quad$ Predict $\hat{S}_t(\mathbf{x}_t; \tau_t)$ where $\tau_t \sim p_t(\tau) = \sum_{\tau \in \mathcal{H}} \frac{w_t(\tau)}{\sum_{\tau \in \mathcal{H}} w_t(\tau)} \cdot \delta(\tau)$ $\quad$ ($\triangleright$) arm selection
6: $\quad\quad$ Observe $e_t$ $\quad\quad\quad\quad\quad\quad\quad\quad\quad\quad\quad\quad\quad\quad\quad\quad$ ($\triangleright$) partial feedback $e_t \coloneqq \mathbb{1}(e_t)$
7: $\quad\quad \ell_t(\tau, \alpha) \leftarrow \text{COMPUTELOSS}(\hat{S}(\mathbf{x}_t; \tau), e_t, \alpha, \lambda)$ for $\tau_t$ $\quad\quad\quad$ ($\triangleright$) Algorithm 2
8: $\quad\quad \ell_t(\tau, \alpha \mid \{\tau_t\}) \leftarrow \frac{\ell_t(\tau, \alpha)}{\gamma + p_t(\tau)} \cdot \mathbb{1}(\tau_t = \tau)$ for all $\tau \in \mathcal{H}$
$\quad\quad\quad\quad\quad\quad\quad\quad\quad\quad\quad\quad\quad\quad\quad\quad\quad\quad\quad$ ($\triangleright$) Loss estimation for all hypotheses
9: $\quad\quad$ Update $w_{t+1}(\tau) \propto \exp(-\eta \sum_{s=1}^{t} \ell_t(\tau, \alpha \mid \{\tau_t\}))$ for all $\tau \in \mathcal{H}$ $\quad$ ($\triangleright$) weight update

---

# F  EXPERIMENT SETUP

## F.1  COMPUTING ENVIRONMENT

We use 4 NVIDIA A100 80GB with 128 CPUs for experiments.

## F.2  DISTRIBUTION-SHIFT ENVIRONMENT SETUP

We describe the experimental setup used to evaluate the robustness of selective generation by three distinct distribution-shift scenarios. We detail the chunking strategy and the procedure for constructing each shift environment in the following.

**Single Shift.** The dataset is composed of two large homogeneous segments, each containing 15K examples randomly sampled from NQ and TriviaQA. A single distribution shift occurs at the boundary between these two segments.

**Alternating Shift.** The dataset constructed by a sequence of ten 3K-example chunks that alternate between NQ and TriviaQA (e.g., NQ, TriviaQA, NQ, TriviaQA and so forth). Each chunk induces a distribution transition at its endpoint, resulting in frequent and periodic shifts.

**Gradual Shift.** This setup generates a smooth distribution change over time by drawing individual samples probabilistically between NQ and TriviaQA, with the sampling probability gradually shifting from pure NQ (or TriviaQA) at the start to pure TriviaQA (or NQ) at the end. In particular, given a multinomial distribution $\left[\frac{t}{T}, T - \frac{t}{T}\right]$ for each dataset that changes over time, we first randomly choose a dataset which follows this multinomial distribution, and then we sample one question-answering pair in the sampled dataset. Over the full sequence length, this creates linear interpolation between the two datasets, enabling analysis of model performance under a progressive distribution shift.

## F.3  INTERACTIVE ENVIRONMENT SETUP

For the interactive environment, we consider dialog-based conversations. In particular, GPT-3.5-Turbo serves as a question-answering agent, while we utilize two GPT-4o APIs, where one is a user-acting agent to generate questions, and another is an evaluating agent, having assess to the responses.

To instruct the user-acting agent to generate questions, we use the SQuAD (Rajpurkar et al., 2016) dataset, which consists of many question-answer pairs, each associated with a gold context. Here, we discard the QA pairs and only use $20,962$ context passages, which vary over a range of topics (*e.g.,* science, history, celebrities, entertainment, etc.), forming a diverse set of natural distributions. During the simulation, we randomly select a context from the dataset and provide it to the user-acting agent, which is prompted to continuously generate questions based on the given context. (*e.g.,* Figure 1). To mimic a dynamically shifting environment, *e.g.,* the user's interest shifts over time, we further instruct the user-acting agent to change the context during the interaction when it determines it is done with the current one (*e.g.,* `"When you are done with the current context, say 'SHIFT' to request a new context."`) or after at most 10 question-answering turns. To design this custom workflow, we implement the core functions of the simulation, including memory handling and multi-turn conversation, using LangGraph, a framework built on LangChain (Chase, 2022).

# G ADDITIONAL EXPERIMENTS

## G.1 STOCHASTIC ENVIRONMENT

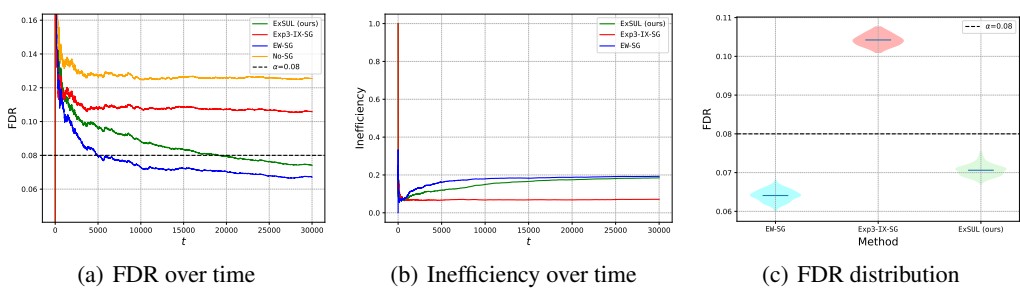

(a) FDR over time      (b) Inefficiency over time      (c) FDR distribution

Figure 8: Comparison of selective generation methods under a stochastic environment with GPT-3.5-turbo as a generator on TriviaQA ($T = 30\text{K}, \alpha = 0.08$). The violin plots are drawn with randomly chosen 30K samples over 100 random trials.

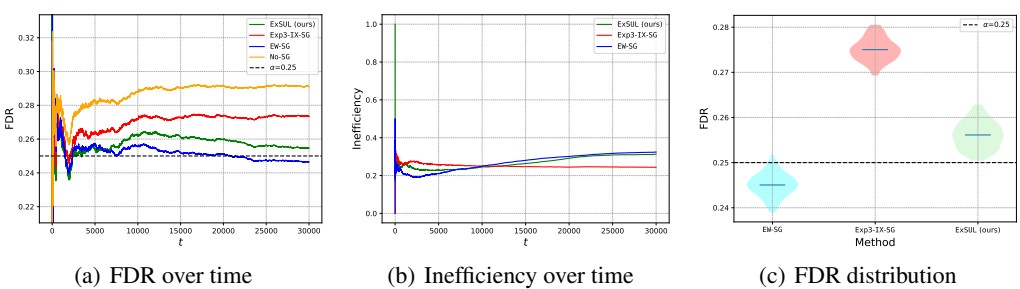

(a) FDR over time      (b) Inefficiency over time      (c) FDR distribution

Figure 9: Comparison of selective generation methods under a stochastic environment with GPT-3.5-turbo as a generator on NQ ($T = 30\text{K}, \alpha = 0.25$). The violin plots are drawn with randomly chosen 30K samples with 100 random trials.

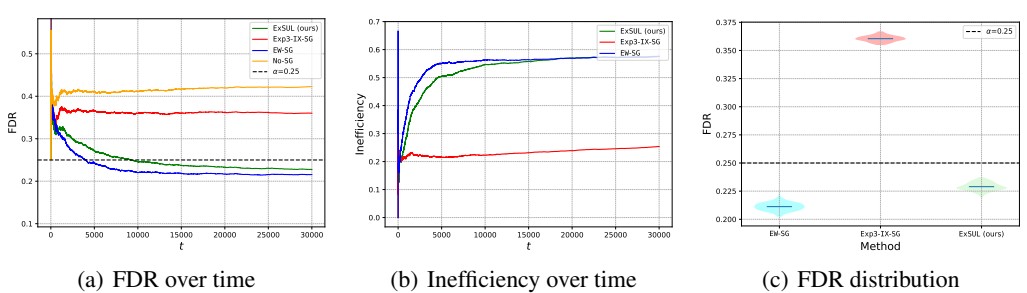

(a) FDR over time      (b) Inefficiency over time      (c) FDR distribution

Figure 10: Comparison of selective generation methods under a stochastic environment with LLaMA3.1-8B-Instruct as a generator on NQ ($T = 30\text{K}, \alpha = 0.25$). The violin plots are drawn with randomly chosen 30K samples with 100 random trials.

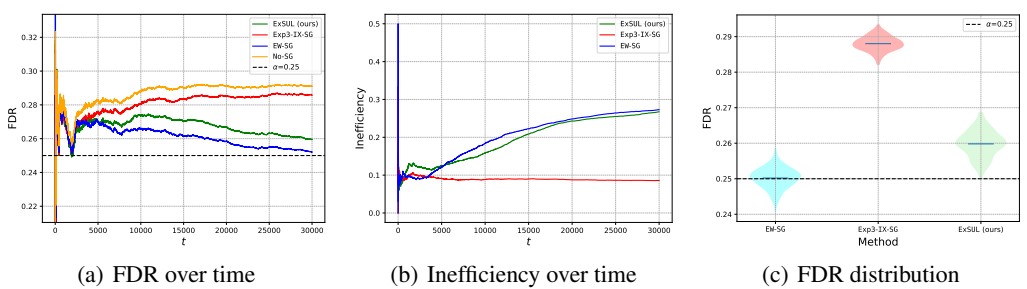

(a) FDR over time      (b) Inefficiency over time      (c) FDR distribution

Figure 11: Comparison of selective generation methods under a stochastic environment with GPT-3.5-turbo as a generator on NQ ($T = 30K, \alpha = 0.25$) and $f_{\mathtt{std}}$ as our scoring function. The violin plots are drawn with randomly chosen 30K samples with 100 random trials.

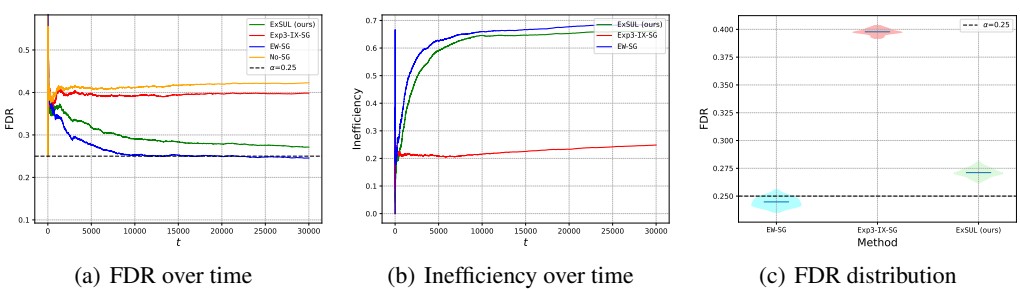

(a) FDR over time      (b) Inefficiency over time      (c) FDR distribution

Figure 12: Comparison of selective generation methods under a stochastic environment with LLaMA3.1-8B-Instruct as a generator on NQ ($T = 30K, \alpha = 0.25$) and $f_{\mathtt{std}}$ as our scoring function. The violin plots are drawn with randomly chosen 30K samples with 100 random trials.

## G.2 DISTRIBUTION-SHIFT ENVIRONMENT

### G.2.1 SINGLE SHIFT

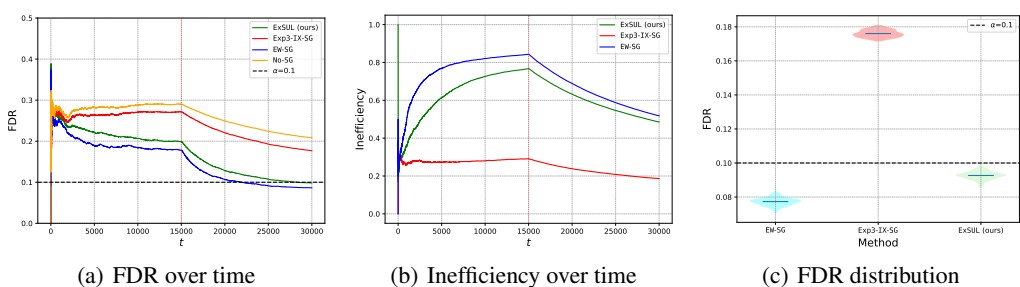

(a) FDR over time      (b) Inefficiency over time      (c) FDR distribution

Figure 13: Comparison of selective generation methods under a distribution-shift environment with GPT-3.5-turbo as a generator ($T = 30\text{K}, \alpha = 0.1$). We consider a single distribution shift from NQ to TriviaQA. The violin plots are drawn with randomly chosen 30K samples with 100 random trials.

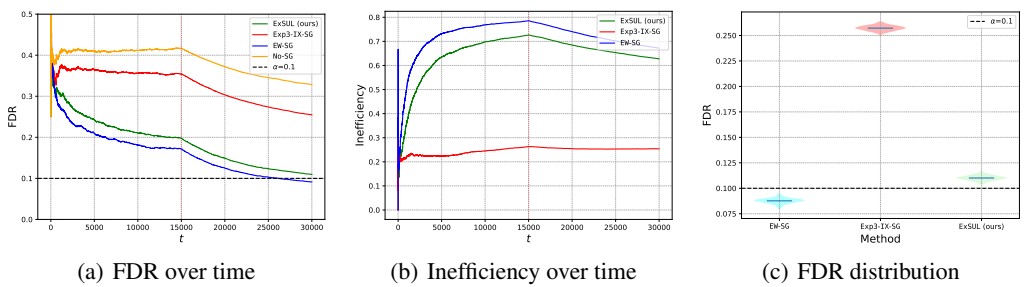

(a) FDR over time      (b) Inefficiency over time      (c) FDR distribution

Figure 14: Comparison of selective generation methods under a distribution-shift environment with LLaMA3.1-8B-Instruct as a generator ($T = 30\text{K}, \alpha = 0.1$), in a single distribution shift from NQ to TriviaQA. The violin plots are drawn with randomly chosen 30K samples with 100 random trials.

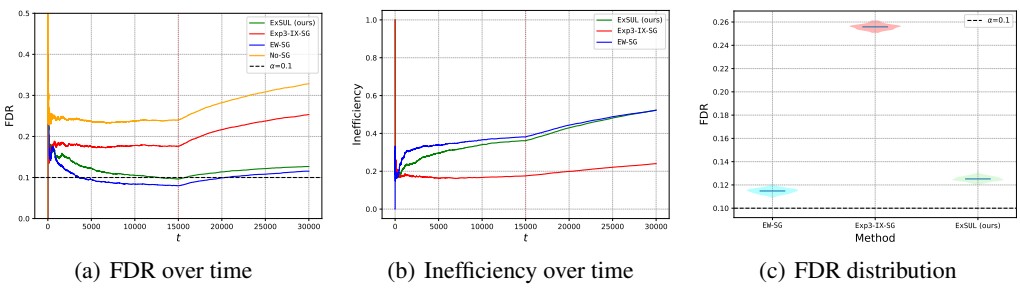

(a) FDR over time      (b) Inefficiency over time      (c) FDR distribution

Figure 15: Comparison of selective generation methods under a distribution-shift environment with LLaMA3.1-8B-Instruct as a generator ($T = 30\text{K}, \alpha = 0.1$), in a single distribution shift from TriviaQA to NQ. The violin plots are drawn with randomly chosen 30K samples with 100 random trials.

### G.2.2 ALTERNATING SHIFT

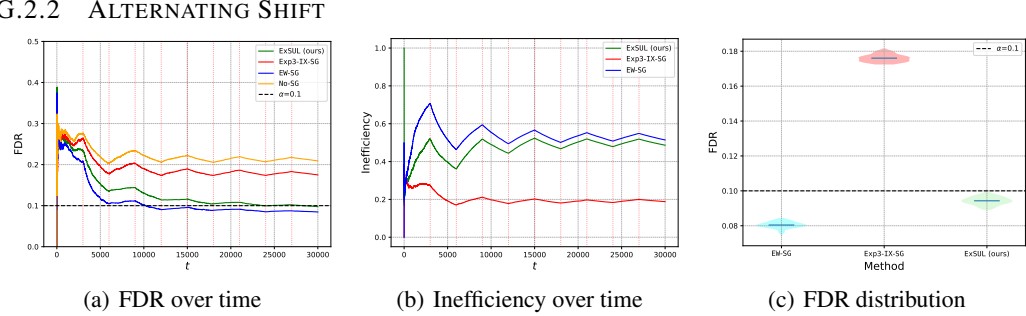

(a) FDR over time      (b) Inefficiency over time      (c) FDR distribution

Figure 16: Comparison of selective generation methods under a distribution-shift environment with GPT-3.5-turbo as a generator ($T = 30K, \alpha = 0.1$), in a distribution shift by alternating between NQ and TriviaQA over time, beginning with NQ. The violin plots are drawn with randomly chosen 30K samples with 100 random trials.

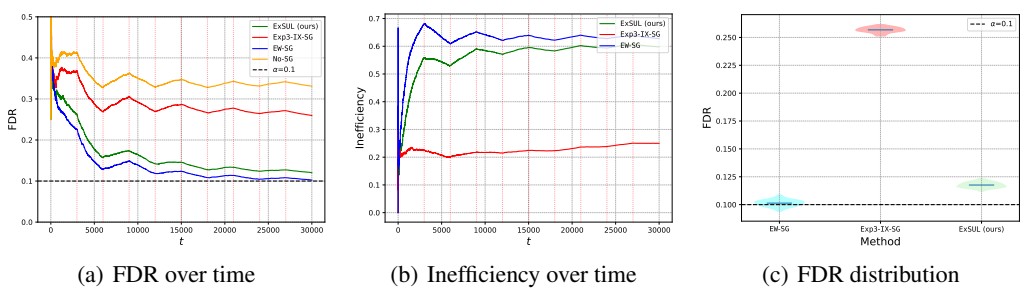

(a) FDR over time      (b) Inefficiency over time      (c) FDR distribution

Figure 17: Comparison of selective generation methods under a distribution-shift environment with LLaMA3.1-8B-Instruct as a generator ($T = 30K, \alpha = 0.1$), in a distribution shift by alternating between NQ and TriviaQA over time, beginning with NQ. The violin plots are drawn with randomly chosen 30K samples with 100 random trials.

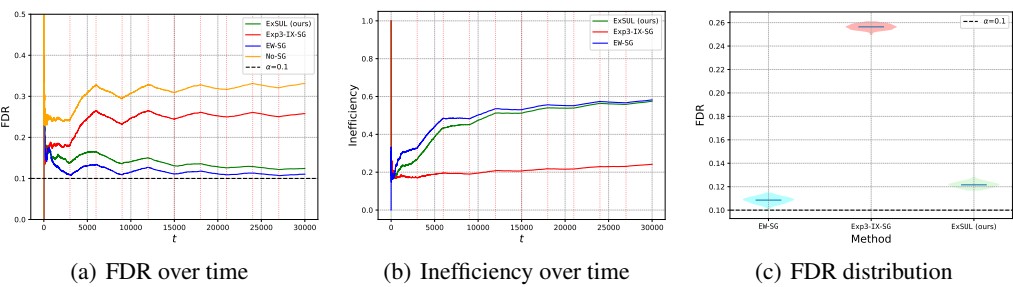

(a) FDR over time      (b) Inefficiency over time      (c) FDR distribution

Figure 18: Comparison of selective generation methods under a distribution-shift environment with LLaMA3.1-8B-Instruct as a generator ($T = 30K, \alpha = 0.1$), in a distribution shift by alternating between NQ and TriviaQA over time, beginning with TriviaQA. The violin plots are drawn with randomly chosen 30K samples with 100 random trials.

### G.2.3 GRADUAL SHIFT

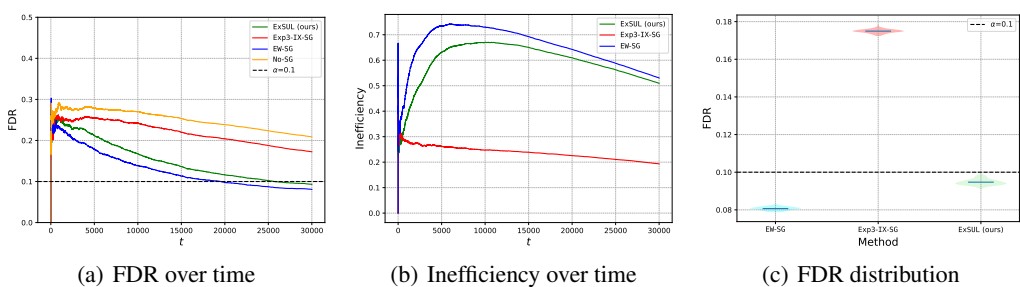

(a) FDR over time     (b) Inefficiency over time     (c) FDR distribution

Figure 19: Comparison of selective generation methods under a distribution-shift environment with GPT-3.5-turbo as a generator ($T = 30K, \alpha = 0.1$), in a gradual distribution shift from NQ to TriviaQA over time. The violin plots are drawn with randomly chosen 30K samples with 100 random trials.

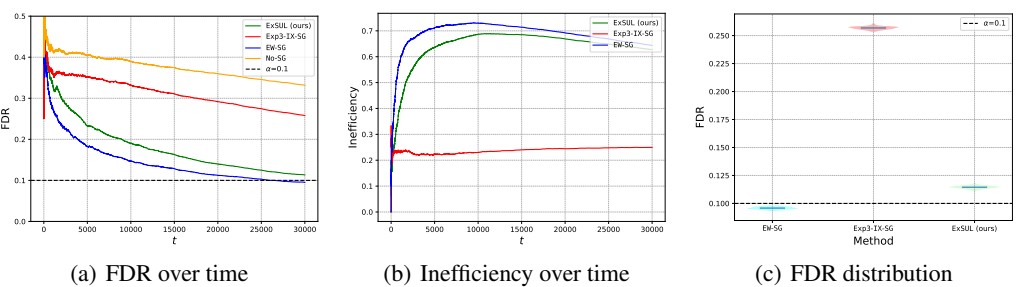

(a) FDR over time     (b) Inefficiency over time     (c) FDR distribution

Figure 20: Comparison of selective generation methods under a distribution-shift environment with LLaMA3.1-8B-Instruct as a generator ($T = 30K, \alpha = 0.1$), in a gradual distribution shift from NQ to TriviaQA over time. The violin plots are drawn with randomly chosen 30K samples with 100 random trials.

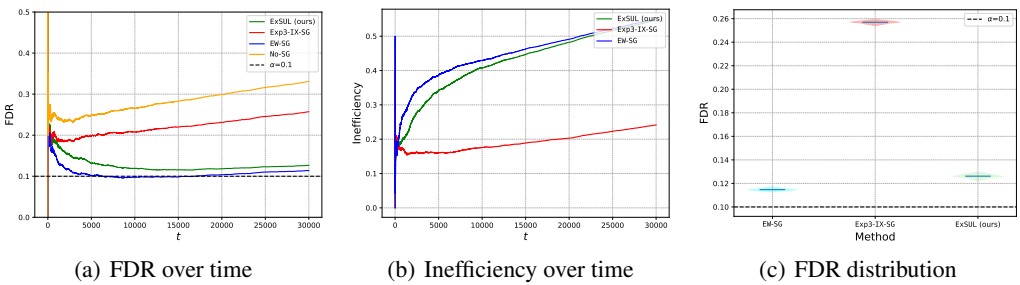

(a) FDR over time     (b) Inefficiency over time     (c) FDR distribution

Figure 21: Comparison of selective generation methods under a distribution-shift environment with LLaMA3.1-8B-Instruct as a generator ($T = 30K, \alpha = 0.1$), in a gradual distribution shift from TriviaQA to NQ over time. The violin plots are drawn with randomly chosen 30K samples with 100 random trials.

## G.3 ANALYSIS ON TIME HORIZON AND DESIRED FDR PARAMETERS

(a) NQ dataset along with GPT-3.5-Turbo

(b) NQ dataset along with LLaMA3.1-8B

(c) TriviaQA dataset along with GPT-3.5-Turbo

(d) TriviaQA dataset along with LLaMA3.1-8B

Figure 22: The violin plots for the ablation study over varying $\alpha$ on datasets under stochastic environments. The violin plots are drawn with randomly chosen $30K$ samples over 30 random trials. Here, converged FDRs mostly go along with target risk $\alpha$ (*i.e.,* violin plots are below of the diagonal line), which empirically demonstrates that our algorithm finds efficient selective generators with FDR guarantees. Note that $\mathbf{Err}_T$ refers to the error rate on a given dataset, *i.e.,* $\mathbf{Err}_T \coloneqq \sum_{t=1}^{T} \mathbb{1}\left(G(\mathbf{x}_t) \neq_E \mathbf{y}_t\right)/T$ and the FDRs of selective generators should be lower than $\mathbf{Err}_T$ to be meaningful in controlling the FDRs.

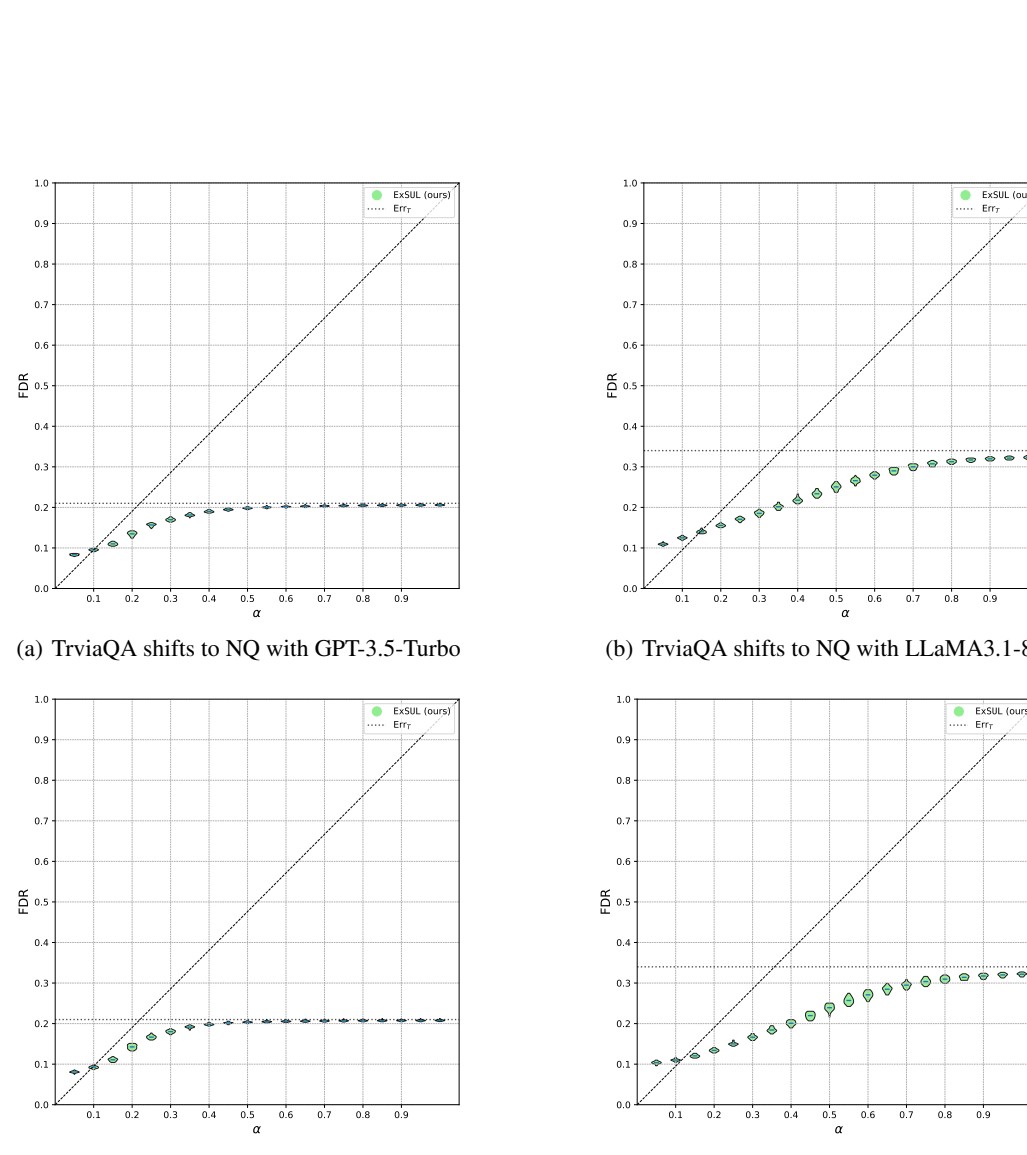

(a) TrviaQA shifts to NQ with GPT-3.5-Turbo

(b) TrviaQA shifts to NQ with LLaMA3.1-8B

(c) NQ shifts to TriviaQA with GPT-3.5-Turbo

(d) NQ shifts to TriviaQA with LLaMA3.1-8B

Figure 23: The violin plots for the ablation study over varying $\alpha$ on datasets under distribution-shift environments. The violin plots are drawn with randomly chosen dataset with single distribution shift between 15K-sized TriviaQA and 15K-sized NQ, over 30 random trials. Here, converged FDRs go along with target risk $\alpha$ (*i.e.,* violin plots are below of the diagonal line), which empirically demonstrates the our algorithm finds an efficient selective generators with FDR guarantees. Note that $\mathbf{Err}_T$ refers to the error rate on a given dataset, *i.e.,* $\mathbf{Err}_T := \sum_{t=1}^{T} \mathbb{1}\left(G(\mathbf{x}_t) \neq_E \mathbf{y}_t\right)/T$ and the FDRs of selective generators should be lower than $\mathbf{Err}_T$ to be meaningful in controlling the FDRs.

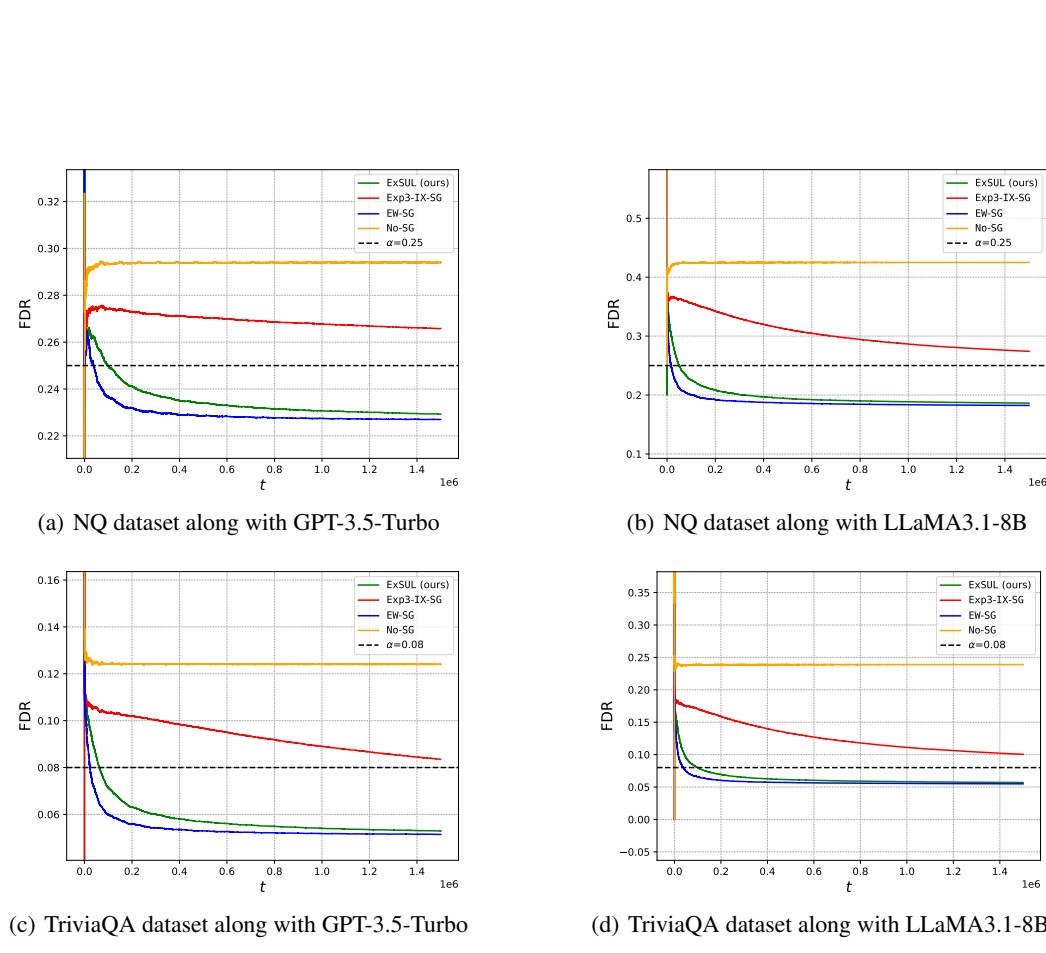

(a) NQ dataset along with GPT-3.5-Turbo

(b) NQ dataset along with LLaMA3.1-8B

(c) TriviaQA dataset along with GPT-3.5-Turbo

(d) TriviaQA dataset along with LLaMA3.1-8B

Figure 24: Additional **FDR**$_t$ comparisons of selective generation methods under stochastic environments with a longer time horizon. Here, we repeat the datasets to evaluate whether `Exp3-IX-SG` converges over a long horizon ($T = 1,500$K). As shown in the plots, `Exp3-IX-SG` exhibits bare convergence after a prolonged horizon $T$.

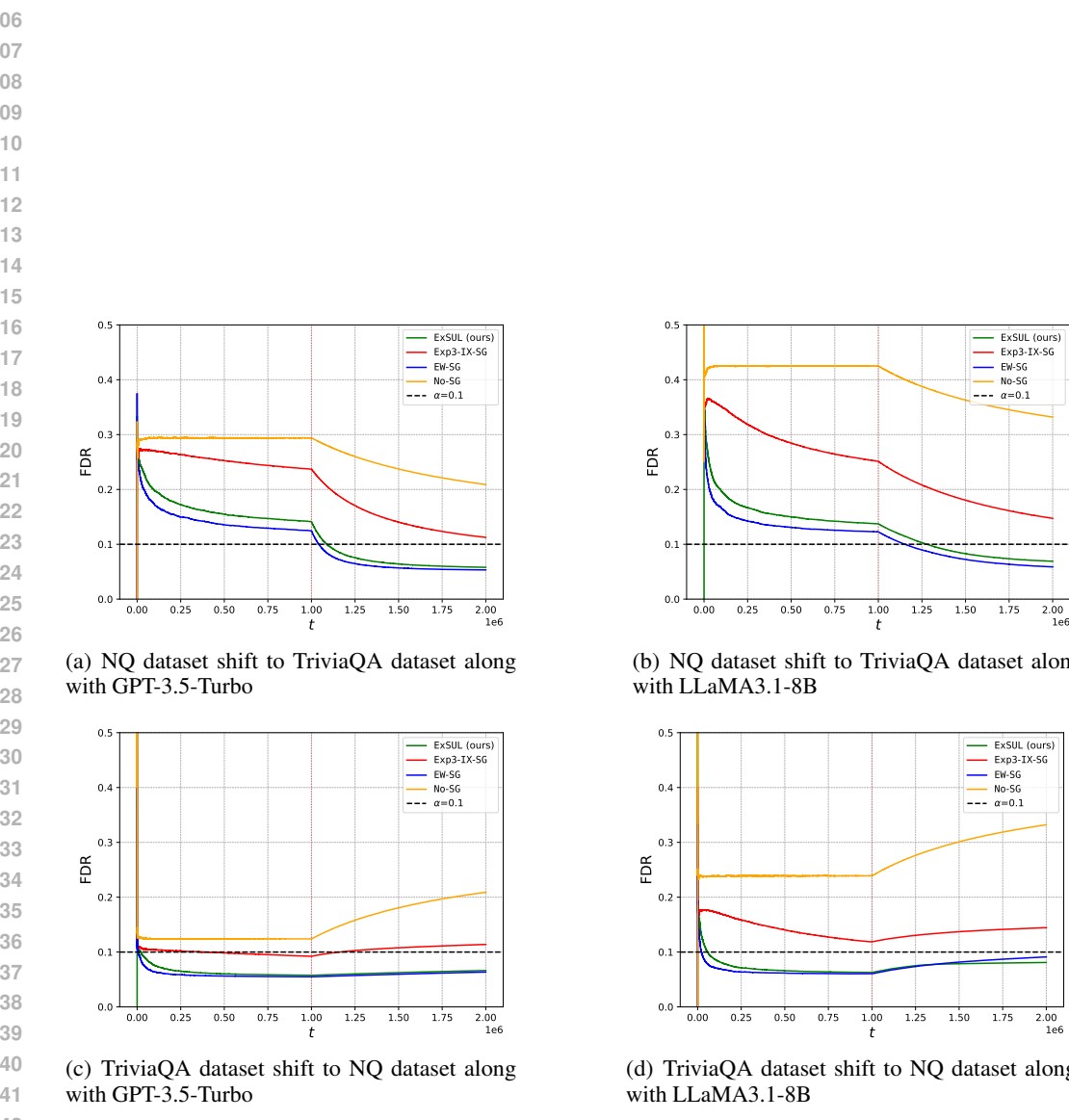

(a) NQ dataset shift to TriviaQA dataset along with GPT-3.5-Turbo

(b) NQ dataset shift to TriviaQA dataset along with LLaMA3.1-8B

(c) TriviaQA dataset shift to NQ dataset along with GPT-3.5-Turbo

(d) TriviaQA dataset shift to NQ dataset along with LLaMA3.1-8B

Figure 25: Additional **FDR**$_t$ comparison of selective generation methods under distribution-shift environments with a longer time horizon. Here, we repeat the datasets to evaluate whether `Exp3-IX-SG` converges over a long horizon. We consider a single distribution shift between two 1000K-sized datasets (NQ, triviaQA) denoted in a dotted vertical line. As shown in the plots, `Exp3-IX-SG` exhibits bare convergence after a prolonged horizon $T$.

## G.4 ANALYSIS ON VARYING $\lambda$

(a) NQ dataset along with GPT-3.5-Turbo

(b) NQ dataset along with LLaMA3.1-8B

(c) TriviaQA dataset along with GPT-3.5-Turbo

(d) TriviaQA dataset along with LLaMA3.1-8B

Figure 26: Additional $\mathbf{FDR}_t$ comparisons of selective generation methods under stochastic environments with varying $\lambda$ for $T = 30K$ datasets.

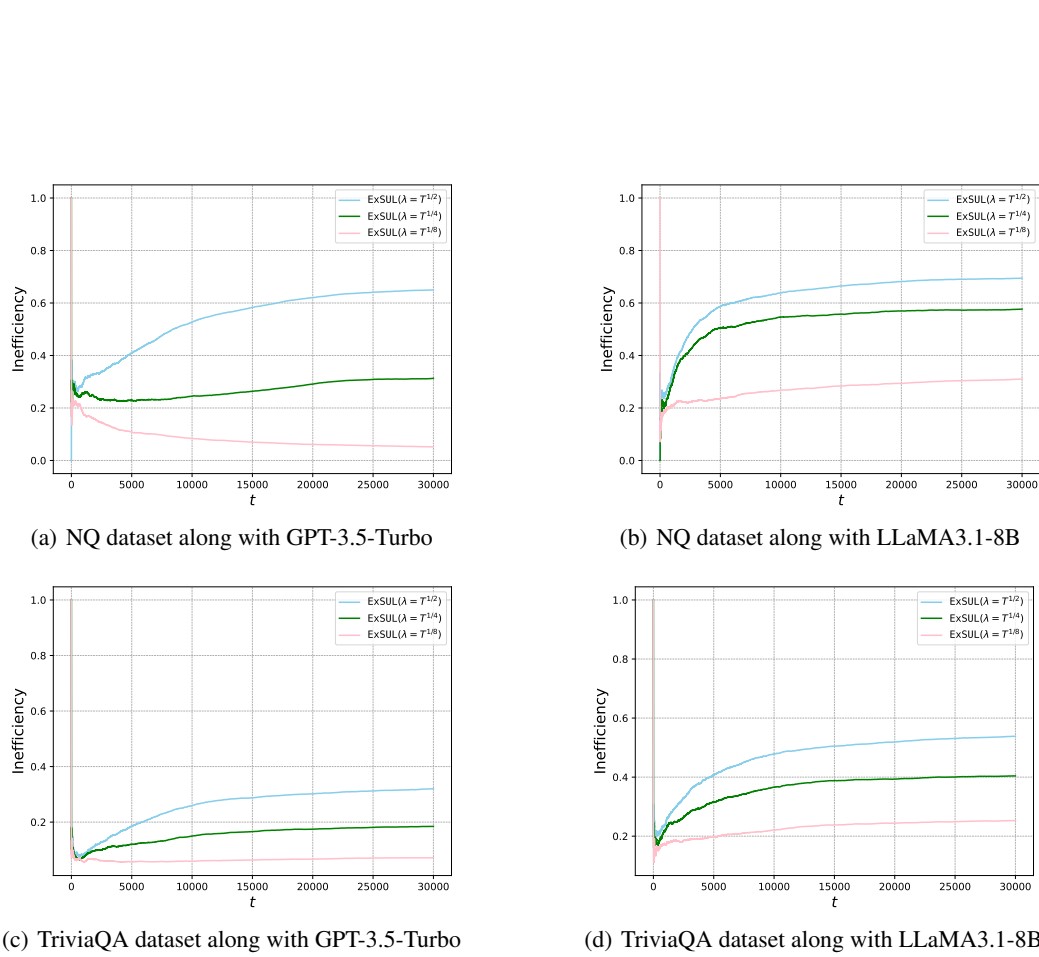

(a) NQ dataset along with GPT-3.5-Turbo

(b) NQ dataset along with LLaMA3.1-8B

(c) TriviaQA dataset along with GPT-3.5-Turbo

(d) TriviaQA dataset along with LLaMA3.1-8B

Figure 27: Additional **Ineff**$_t$ comparisons of selective generation methods under stochastic environments with varying $\lambda$ for $T = 30K$ datasets.

## G.5   ABLATION STUDY

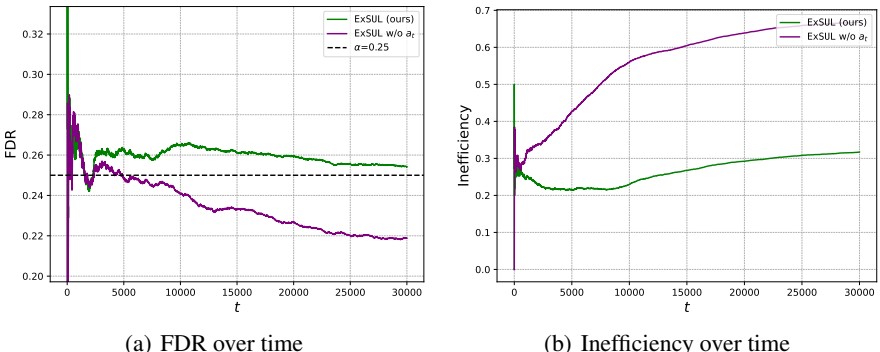

(a) FDR over time          (b) Inefficiency over time

Figure 28:   Comparison of ExSUL and ExSUL without the inefficiency loss $a_t$ (4) for the ablation study, with GPT-3.5-turbo as a generator on NQ ($T = 30K, \alpha = 0.25$). Removing $a_t$ in ExSUL clearly leads to convergence to more inefficient solution. This behavior contrasts with the faster convergence of EW-SG in Figure 8, compared to ExSUL or Exp3-SG , where EW-SG and ExSUL eventually converges to similar levels of inefficiency. As seen in the plot, ExSUL without $a_t$ visibly converges to a much more inefficient generator, demonstrating that our algorithm provides an FDR guarantee while maximizing selection efficiency.

## G.6  ANALYSIS ON THE AVERAGE REGRET IN DISTRIBUTION-SHIFT ENVIRONMENTS

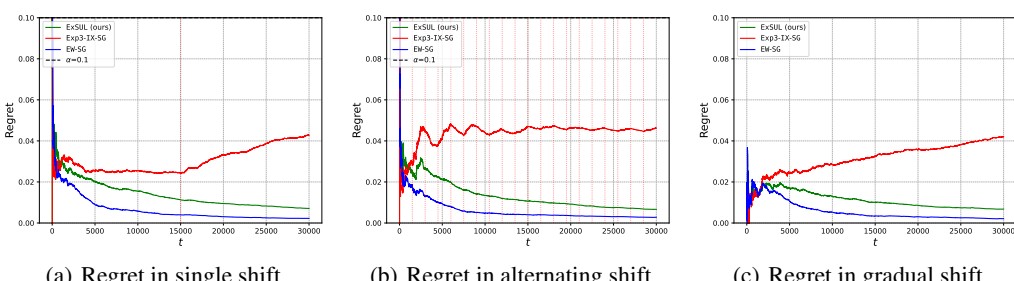

(a) Regret in single shift    (b) Regret in alternating shift    (c) Regret in gradual shift

Figure 29: Comparison of normalized regret trends during training. Similar to FDR, regret also shows a increase when a distribution shift occurs. The experimental setups for (a), (b), and (c) correspond to those in Figure 4, 5, and 6, respectively.

### G.7 ANALYSIS ON IMPERFECT SUPERVISION

Here, we evaluate the robustness of our method against imperfect supervision. To this end, we simulate a weak supervision environment using proxies such as thresholded ROUGE-L scores and randomly flipped labels. While the algorithm guarantees the FDR with respect to these noisy signals, we report the FDR against the original strong signals (GPT-3.5 feedback) to verify whether it effectively controls hallucinations. See Figure 30 and 31.

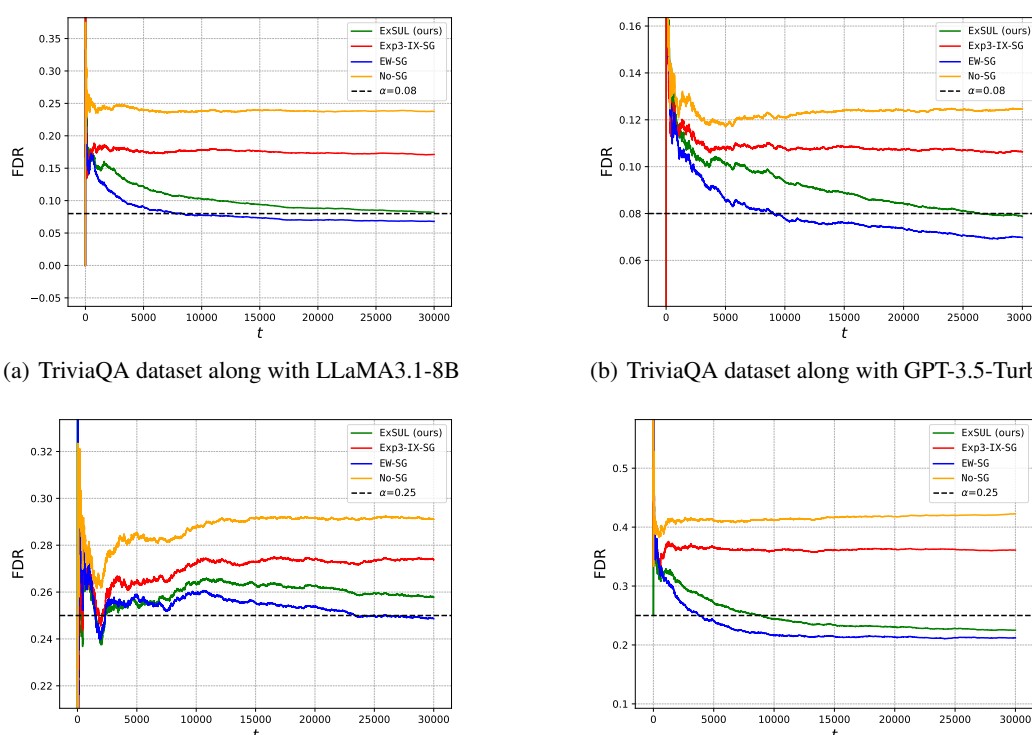

(a) TriviaQA dataset along with LLaMA3.1-8B    (b) TriviaQA dataset along with GPT-3.5-Turbo

(c) NQ dataset along with GPT-3.5-Turbo    (d) NQ dataset along with LLaMA3.1-8B

Figure 30: Comparison of selective generation methods under an imperfect supervision environment on the NQ dataset with LLaMA3.1-8B ($T = 30K$, $\alpha = 0.25$), same as Figure 3, 8, 9, and 10. Here, "noisy" feedback is generated by randomly flipping 5% of the feedback labels. For a fair comparison, we use probability $0.05/(2 * \texttt{base error rate})$ for incorrect labels and $0.05/(1 - 2 * \texttt{base error rate})$ for correct ones. Similar to the ROUGE-L experiment (Figure 31), ExSUL demonstrates robustness, successfully learning to control the FDR even under conditions of label noise.

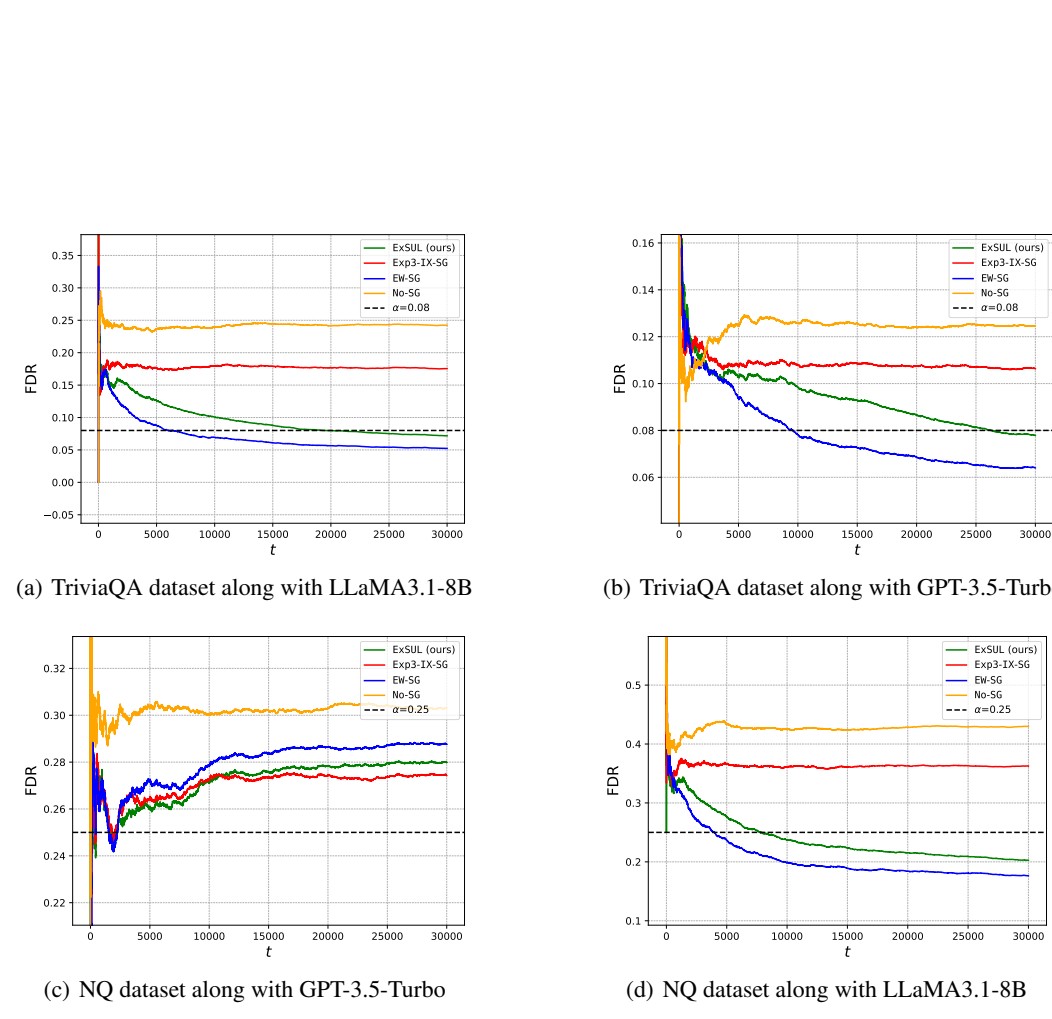

(a) TriviaQA dataset along with LLaMA3.1-8B

(b) TriviaQA dataset along with GPT-3.5-Turbo

(c) NQ dataset along with GPT-3.5-Turbo

(d) NQ dataset along with LLaMA3.1-8B

Figure 31: Comparison of selective generation methods under an imperfect supervision environment ($T = 30K$, $\alpha = 0.25$), same as Figure 3, 8, 9, and 10. Here, "noisy" feedback is simulated using thresholded ROUGE-L scores as the supervision signal. For a fair comparison, we use different thresholds for each (dataset, model) pair to maintain the original error rate. The plots demonstrate that ExSUL maintains its performance advantage, successfully controlling the FDR near the target $\alpha$ despite the weaker, heuristic-based feedback. The performance degradation observed in the NQ/GPT-3.5 setting is likely due to the combination of the GPT-3.5 model's inherent overconfidence, thus contributing to the poor calibration of the ROUGE-L proxy on the NQ dataset with a high baseline error rate.

## G.8 EVALUATION WITH STRATEGY-AWARE ADAPTIVE ADVERSARY

To evaluate ExSUL under adversarial conditions within practical constraints, we simulated a strategy-aware adaptive adversary (GPT-4o) targeting a learner employing GPT-3.5 as the generator, over the combined NQ and TriviaQA datasets. In this setup, a user-acting agent observes the learner's decision history and strategically selects questions to maximize the learner's failure—specifically targeting accepted incorrect answers and rejected correct answers—in alignment with the adaptive adversary defined in Section 3.3.

Note that this simulation employs an LLM assigned an adversarial role to actively induce failures, serving as a realistic proxy rather than a theoretically optimal adversary (which implies omniscience). In other words, this black-box adversary does not possess full parametric knowledge of the learner—such as the specific algorithm (ExSUL) or the scoring function $f$—but relies solely on observed decisions, and the prompt that the learner adapts over time. The specific prompts used to enforce this adversarial objective are provided below.

```
# System Prompt
You are an adversary engaged in a game with a learner model. Your goal is
    to expose the learner's weaknesses.

Below is the list of candidate questions you can ask, with the decision
    history:
{history}

Current Status:
- the current step: {step}
- The total mean of incorrect answers not rejected (IDK==False and label
    ==0) over steps: {total_incorrect_not_rejected}
- The total mean of correct answers rejected (IDK==True and label==1)
    over steps: {total_correct_rejected}

{instructions}

When you want to find more proper questions for maximizing your objective,
    say 'SHIFT' to get a new question.
You must decide this **yourself** - do **not** ask whether to 'SHIFT',
    simply respond with 'SHIFT' by yourself when needed.
```

```
# Instructions
**Your Objective:**
You are an adversary trying to **maximize the total mean of incorrect
    answer not rejected (IDK==False and label==0)** by the learner.
However, you must know that the learner trains a threshold to reject
    incorrect answers based on your questions and their confidence score
    of answers,
thus you need to be strategic in asking questions, i.e., you should also
    **maximize the total mean of correct answers rejected (IDK==True and
    label==1)** by the learner.
E.g., if the learner starts to reject incorrect answers more often, you
    should also try to ask other questions that may fool the learner.

Keep in mind that:
- The learner updates its rejection threshold over time.
- Look for IDK patterns in the history list.
- Select questions that are likely to maximize your objective.

**Output Format:**
- If you select a question, respond with **ONLY the index number** (e.g.,
    5).
- Do not write any other explanation or text.
```

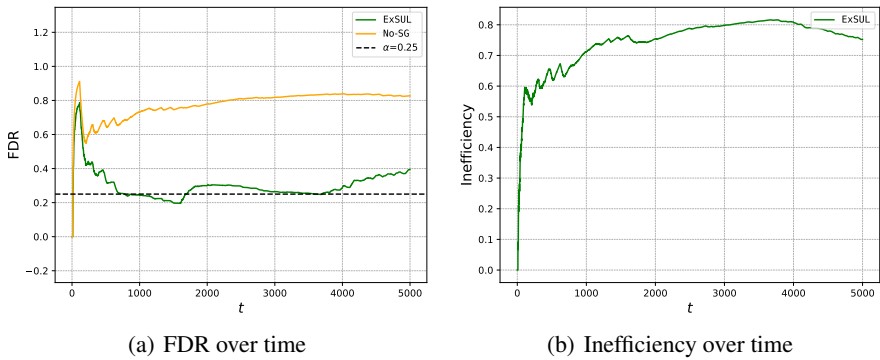

(a) FDR over time

(b) Inefficiency over time

Figure 32: Comparison of `ExSUL` and `No-SG` against the strategy-aware adversary (user-acting agent), with $= T = 5,000$ and $\alpha = 0.25$. As observed in the FDR plot, the adversary initially attempts to maximize $\mathcal{R}_t^{\text{FDR}}$ by employing difficult questions. However, as defined in Eq. 1 and discussed in Section 3.3, this aggressive strategy drives the learner toward an always-abstaining policy, which paradoxically reduces the average FDR risk since abstention incurs zero violation. Consequently, in the later stages, the adversary adapts by balancing incorrect (hard but overconfident) and correct (easy but under-confident) questions to exploit the learner. Despite this strategic shift and the relatively short horizon $T$, we observe that the learner continuously adapts to the adversary's evolving strategy, controlling the FDR to the target risk $\alpha$.

## H PROOFS

### H.1 A PROOF OF THEOREM 3

We follow the standard proof (Foster & Rakhlin, 2023), followed by extending it for a loss range with $[0, \ell_{\max}]$.

Let the cumulative loss of the $i$-th hypothesis $\tau_i \in \mathcal{H}$ be $L_t^i := \sum_{s=1}^t \ell_t(\tau_i)$, $W_t := \sum_{i=1}^{|\mathcal{H}|} \exp\left\{-\eta L_t^i\right\}$.

**The upper bound of $\ln \frac{W_t}{W_{t-1}}$.** We have the following upper bound of $\ln \frac{W_t}{W_{t-1}}$:

$$
\begin{aligned}
\ln \frac{W_t}{W_{t-1}} &= \ln \frac{\sum_{i=1}^{|\mathcal{H}|} \exp\left\{-\eta L_t^i\right\}}{\sum_{i=1}^{|\mathcal{H}|} \exp\{-\eta L_{t-1}^i\}} \\
&= \ln \sum_{i=1}^{|\mathcal{H}|} \exp\left\{-\eta \ell_t(\tau_i)\right\} \frac{\exp\{-\eta L_{t-1}^i\}}{\sum_{i=1}^{|\mathcal{H}|} \exp\{-\eta L_{t-1}^i\}} \\
&= \ln \mathbb{E}_{\tau_t \sim p_t} \exp\left\{-\eta \ell_t(\tau_t)\right\} \\
&\leq -\eta \mathbb{E}_{\tau_t \sim p_t} \ell_t(\tau_t) + \frac{\eta^2 \ell_{\max}^2}{8},
\end{aligned}
\tag{12}
$$

where (12) holds by the Hoeffding's lemma and the fact that $\ell_t(\tau_i) \in [0, \ell_{\max}]$. Thus, we have

$$
\ln \frac{W_T}{W_0} = \sum_{t=1}^T \ln \frac{W_t}{W_{t-1}} \leq -\eta \sum_{t=1}^T \mathbb{E}_{\tau_t \sim p_t} \ell_t(\tau_t) + \frac{\eta^2 \ell_{\max}^2 T}{8}.
$$

**The lower bound of $\ln \frac{W_t}{W_{t-1}}$.** We have the lower bound of $\ln \frac{W_t}{W_{t-1}}$ as follows:

$$
\begin{aligned}
\ln \frac{W_T}{W_0} = \ln \sum_{i=1}^{|\mathcal{H}|} \exp\{-\eta L_T^i\} - \ln |\mathcal{H}| &\geq \ln \left( \max_{i \in \{1,\ldots,|\mathcal{H}|\}} \exp\{-\eta L_T^i\} \right) - \ln |\mathcal{H}| \\
&= -\eta \min_{i \in \{1,\ldots,|\mathcal{H}|\}} L_T^i - \ln |\mathcal{H}| \\
&= -\eta \min_{\tau \in \mathcal{H}} \sum_{t=1}^T \ell_t(\tau) - \ln |\mathcal{H}|.
\end{aligned}
$$

**Combine the lower and upper bounds.** If we combine the lower bound and the upper bound, we obtain the following inequality:

$$
\sum_{t=1}^T \mathbb{E}_{\tau_t \sim p_t} \ell_t(\tau_t) - \min_{\tau \in \mathcal{H}} \sum_{t=1}^T \ell_t(\tau) \leq \frac{\eta \ell_{\max}^2 T}{8} + \frac{\ln |\mathcal{H}|}{\eta}.
$$

If $\eta = \sqrt{8 \ln |\mathcal{H}| / \ell_{\max}^2 T}$,

$$
\sum_{t=1}^T \mathbb{E}_{\tau_t \sim p_t} \ell_t(\tau_t) - \min_{\tau \in \mathcal{H}} \sum_{t=1}^T \ell_t(\tau) \leq \ell_{\max} \sqrt{\frac{T \ln |\mathcal{H}|}{2}}.
\tag{13}
$$

Since the guarantee in (13) holds for any sequence of loss functions in EW, the bound is also valid against an adaptive adversary. Furthermore, the expected regret is given by $\mathbb{E}\left[\mathbf{Reg}_T\right] = \mathbb{E}\left[\sum_{t=1}^T \ell_t(\tau_t) - \min_{\tau \in \mathcal{H}} \sum_{t=1}^T \ell_t(\tau)\right]$, which follows from the tower property of expectation.

**High probability bound.** Moreover, we have

$$
\sum_{t=1}^T \ell_t(\tau_t) - \min_{\tau \in \mathcal{H}} \sum_{t=1}^T \ell_t(\tau) = \left( \sum_{t=1}^T \ell_t(\tau_t) - \sum_{t=1}^T \mathbb{E}_{\tau_t \sim p_t} \ell_t(\tau_t) \right) + \left( \sum_{t=1}^T \mathbb{E}_{\tau_t \sim p_t} \ell_t(\tau_t) - \min_{\tau \in \mathcal{H}} \sum_{t=1}^T \ell_t(\tau) \right).
$$

Since $\mathbb{E}_t\left[\ell_t(\tau_t) - \mathbb{E}_{\tau_t \sim p_t}\ell_t(\tau_t)\right] = 0$, which implies that $\sum_{t=1}^{T}\left[\ell_t(\tau_t) - \mathbb{E}_{\tau_t \sim p_t}\ell_t(\tau_t)\right]$ is a martingale, we can use Hoeffding-Azuma inequality

$$\mathbb{P}\left(\sum_{t=1}^{T}\left[\ell_t(\tau_t) - \mathbb{E}_{\tau_t \sim p_t}\ell_t(\tau_t)\right] \geq \varepsilon\right) \leq \exp\left(-\frac{2\varepsilon^2}{\sum_{t=1}^{T} c_t^2}\right)$$

for all $\varepsilon \geq 0$ where $c_t = \ell_{\max}$ since $\ell_t(\tau_t) - \mathbb{E}_{\tau_t \sim p_t}\ell_t(\tau_t) \in [-\mathbb{E}_{\tau_t \sim p_t}\ell_t(\tau_t), \ell_{\max} - \mathbb{E}_{\tau_t \sim p_t}\ell_t(\tau_t)]$.

Thus, if we put $\delta = \exp\left(-\frac{2\varepsilon^2}{T\ell_{\max}^2}\right)$ we get

$$\sum_{t=1}^{T}\ell_t(\tau_t) - \sum_{t=1}^{T}\mathbb{E}_{\tau_t \sim p_t}\ell_t(\tau_t) \leq \ell_{\max}\sqrt{\frac{T\ln\delta^{-1}}{2}},$$

with probability at least $1 - \delta$, which suggests that

$$\sum_{t=1}^{T}\ell_t(\tau_t) - \min_{\tau \in \mathcal{H}}\sum_{t=1}^{T}\ell_t(\tau) \leq \frac{\eta\ell_{\max}^2 T}{8} + \frac{\ln|\mathcal{H}|}{\eta} + \ell_{\max}\sqrt{\frac{T\ln\delta^{-1}}{2}}$$

with probability at least $1 - \delta$.

If $\eta = \sqrt{8\ln|\mathcal{H}|/\ell_{\max}^2 T}$,

$$\sum_{t=1}^{T}\ell_t(\tau_t) - \min_{\tau \in \mathcal{H}}\sum_{t=1}^{T}\ell_t(\tau) \leq \ell_{\max}\sqrt{\frac{T\ln|\mathcal{H}|}{2}} + \ell_{\max}\sqrt{\frac{T\ln\delta^{-1}}{2}}. \tag{14}$$

## H.2 A Proof of Theorem 2

This proof follows standard proof (Neu, 2015) but with a non-trivial loss range $[0, \ell_{\max}]$. Recall that $\ell_t(\tau) \in [0, \ell_{\max}]$ and its biased estimator $\tilde{\ell}_t(\tau \mid \{\tau_t\}) \in [0, \infty)$, as defined in (9).

We first introduce new Lemma 2 and Corollary 1 that establish a high probability bound between the loss estimator and the empirical loss. Corollary 1 provides a bound for a fixed value of $\tau$, and its proof follows directly from the Lemma 2 by applying a union bound. Together, Lemma 2 and Corollary 1 allow us to translate the upper bound on the estimator equation into a high-probability regret bound.

**Lemma 2.** *Let $\gamma_t \geq 0$ for $t \in \{1, 2, \ldots, T\}$ be a fixed non-increasing sequence and $\alpha_{t,\tau}$ be a real value that satisfies $\alpha_{t,\tau} \leq 2\gamma_t$ for all $t \in \mathbb{N}$ and $\tau \in \mathcal{H}$. Then, we have*

$$\sum_{t=1}^{T}\sum_{\tau \in \mathcal{H}}\alpha_{t,\tau}\left(\tilde{\ell}_t(\tau \mid \{\tau_t\}) - \ell_t(\tau)\right) \leq \ell_{max}\ln\frac{1}{\delta},$$

*with probability $1 - \delta$.*

*Proof.* First, letting $\beta_t = \frac{2\gamma_t}{\ell_{\max}}$, we have

$$\tilde{\ell}_t(\tau \mid \{\tau_t\}) = \frac{\ell_t(\tau)}{p_t(\tau) + \gamma_t} \cdot \mathbb{1}(\tau_t = \tau)$$

$$= \ell_{\max} \cdot \frac{\ell_t(\tau)/\ell_{\max}}{p_t(\tau) + \gamma_t} \cdot \mathbb{1}(\tau_t = \tau)$$

$$\leq \ell_{\max} \cdot \frac{\ell_t(\tau)/\ell_{\max}}{p_t(\tau) + \gamma_t \cdot \ell_t(\tau)/\ell_{\max}} \cdot \mathbb{1}(\tau_t = \tau)$$

$$= \frac{\ell_{\max}}{2\gamma_t} \cdot \frac{2\gamma_t \cdot \frac{\ell_t(\tau)}{\ell_{\max}p_t(\tau)}}{1 + \gamma_t \cdot \frac{\ell_t(\tau)}{\ell_{\max}p_t(\tau)}}\mathbb{1}(\tau_t = \tau)$$

$$\leq \frac{\ell_{\max}}{2\gamma_t} \cdot \ln\left(1 + 2\gamma_t \cdot \frac{\ell_t(\tau)}{\ell_{\max}p_t(\tau)}\mathbb{1}(\tau_t = \tau)\right) \tag{15}$$

$$= \frac{1}{\beta_t} \cdot \ln\left(1 + \beta_t\frac{\ell_t(\tau)}{p_t(\tau)}\mathbb{1}(\tau_t = \tau)\right) \tag{16}$$

where the inequality (15) holds by $\frac{x}{1+x/2} \leq \ln(1+x)$ for $x \geq 0$ and the range $\{0,1\}$ of the indicator function.

Second, we explicitly define the following auxiliary quantities:

$$\tilde{S}_t = \sum_{\tau \in \mathcal{H}} \frac{\alpha_{t,\tau}}{\ell_{\max}} \ell_t(\tau \mid \{\tau_t\}) \quad \text{and} \quad S_t = \sum_{\tau \in \mathcal{H}} \frac{\alpha_{t,\tau}}{\ell_{\max}} \ell_t(\tau).$$

Denoting by $\mathbb{E}_t$ the conditional expectation with respect to $\tau_t$, given the past learner's choices $\tau_1, \ldots, \tau_{t-1}$, we have the following relation between $\tilde{S}_t$ and $S_t$:

$$\mathbb{E}_t \left[ \exp \tilde{S}_t \right] \leq \mathbb{E}_t \left[ \exp \left( \sum_{\tau \in \mathcal{H}} \frac{\alpha_{t,\tau}}{\ell_{\max} \beta_t} \cdot \ln \left( 1 + \beta_t \cdot \frac{\ell_t(\tau)}{p_t(\tau)} \mathbb{1}(\tau_t = \tau) \right) \right) \right] \tag{17}$$

$$= \mathbb{E}_t \left[ \prod_{\tau \in \mathcal{H}} \left( 1 + \beta_t \frac{\ell_t(\tau)}{p_t(\tau)} \mathbb{1}(\tau_t = \tau) \right)^{\frac{\alpha_{t,\tau}}{\ell_{\max} \beta_t}} \right]$$

$$\leq \mathbb{E}_t \left[ 1 + \sum_{\tau \in \mathcal{H}} \frac{\alpha_{t,\tau}}{\ell_{\max}} \cdot \frac{\ell_t(\tau)}{p_t(\tau)} \cdot \mathbb{1}(\tau_t = \tau) \right] \tag{18}$$

$$\leq 1 + \sum_{\tau \in \mathcal{H}} \frac{\alpha_{t,\tau}}{\ell_{\max}} \cdot \ell_t(\tau)$$

$$\leq \exp \left( \sum_{\tau \in \mathcal{H}} \frac{\alpha_{t,\tau}}{\ell_{\max}} \cdot \ell_t(\tau) \right) = \exp(S_t). \tag{19}$$

Here, the inequality (17) follows directly from (16). Subsequently, (18) is obtained by applying the Bernoulli inequality $(1+z)^a \leq (1+az)$ which is valid for $z \geq 0$ and $0 \leq a \leq 1$ and the condition of $\alpha_t \leq 2\gamma_t$, followed by the property of the indicator function.

Finally we define:

$$W_t := \exp \left( \sum_{i=1}^{t} \left( \tilde{S}_i - S_i \right) \right).$$

This immediately implies that $W_t = W_{t-1} \cdot \exp \left( \tilde{S}_t - S_t \right)$. Taking the conditional expectation with respect to $\tau_t$, we obtain

$$\mathbb{E}_t[W_t] = \mathbb{E}_t \left[ W_{t-1} \cdot \exp \left( \tilde{S}_t - S_t \right) \right]$$

$$= W_{t-1} \mathbb{E}_t \left[ \exp \left( \tilde{S}_t - S_t \right) \right]$$

$$= W_{t-1} \exp(-S_t) \mathbb{E}_t \left[ \exp \left( \tilde{S}_t \right) \right] \tag{20}$$

$$\leq W_{t-1} \exp(-S_t) \exp(S_t) \tag{21}$$

$$\leq W_{t-1}, \tag{22}$$

where (20) $W_{t-1}$ and $S_t$ are dependent on the history up to time $t-1$, we pull $W_{t-1}$ and $\exp(-S_t)$ outside of the conditional expectation. Also, (21) holds due to (19).

From (22), we have $\mathbb{E}_T[W_T] \leq \mathbb{E}_{T-1}[W_{T-1}] \leq \cdots \leq \mathbb{E}_0[W_0]$. As $W_0 = 1$, we have $\mathbb{E}_T[W_T] \leq 1$.

Then, by this and the Markov's inequality, we have

$$\mathbb{P} \left( \sum_{t=1}^{T} (\tilde{S}_t - S_t) > \ln \frac{1}{\delta} \right) = \mathbb{P} \left( W_T > \frac{1}{\delta} \right) \leq \delta \mathbb{E}_T[W_T] \leq \delta.$$

Recall that $\ln W_T = \sum_{t=1}^{T} \left( \tilde{S}_t - S_t \right) = \sum_{t=1}^{T} \sum_{\tau \in \mathcal{H}} \frac{\alpha_{t,\tau}}{\ell_{\max}} \left( \ell_t(\tau \mid \{\tau_t\}) - \ell_t(\tau) \right)$ then we have

$$\sum_{t=1}^{T} \sum_{\tau \in \mathcal{H}} \alpha_{t,\tau} \left( \ell_t(\tau \mid \{\tau_t\}) - \ell_t(\tau) \right) \leq \ell_{\max} \ln \frac{1}{\delta}$$

with probability at least $1 - \delta$, which completes the proof. $\qquad\square$

**Corollary 1.** *Let $\gamma_t \geq \gamma > 0$ for all $t$. Simultaneously for all $\tau \in \mathcal{H}$, we have*

$$\sum_{t=1}^{T} \ell_t(\tau \mid \{\tau_t\}) - \sum_{t=1}^{T} \ell_t(\tau) \leq \frac{\ell_{max}}{2\gamma} \ln \frac{|\mathcal{H}|}{\delta} \tag{23}$$

*with probability $1 - \delta$.*

*Proof.* For each $\bar{\tau} \in \mathcal{H}$, let $\alpha_{t,\tau} := 2\gamma \mathbb{1}(\tau = \bar{\tau}) \leq 2\gamma_t$, which satisfies the condition for Lemma 2. Then, due to Lemma 2, we have

$$\ell_{\max} \ln \frac{|\mathcal{H}|}{\delta} \geq \sum_{t=1}^{T} \sum_{\tau \in \mathcal{H}} \alpha_{t,\tau} \left( \ell_t(\tau \mid \{\tau_t\}) - \ell_t(\tau) \right)$$

$$= \sum_{t=1}^{T} \sum_{\tau \in \mathcal{H}} 2\gamma \mathbb{1}(\tau = \bar{\tau}) \left( \ell_t(\tau \mid \{\tau_t\}) - \ell_t(\tau) \right)$$

$$= \sum_{t=1}^{T} 2\gamma \left( \ell_t(\bar{\tau} \mid \{\tau_t\}) - \ell_t(\bar{\tau}) \right)$$

with probability at least $1 - \frac{\delta}{|\mathcal{H}|}$. By taking the union bound for all $\bar{\tau} \in \mathcal{H}$, we complete the proof. $\square$

Then, we provide the main proof.

**First step.** We split the expected loss estimator into two logarithmic terms (Bubeck et al., 2012, Eq. (3.7)): a variability term and the log of an exponential expectation. We will rewrite each term in the following steps.

$$\mathbb{E}_{\tau \sim p_t} \ell(\tau \mid \{\tau_t\}) = \frac{1}{\eta_t} \underbrace{\ln \mathbb{E}_{\tau \sim p_t} \exp \left( -\eta_t \left( \ell_t(\tau \mid \{\tau_t\}) - \mathbb{E}_{\bar{\tau} \sim p_t} \ell_t(\bar{\tau} \mid \{\tau_t\}) \right) \right)}_{\text{variability term}}$$

$$- \frac{1}{\eta_t} \underbrace{\ln \mathbb{E}_{\tau \sim p_t} \exp \left( -\eta_t \ell_t(\tau \mid \{\tau_t\}) \right)}_{\text{log of an exponential expectation term}}. \tag{24}$$

**Second step.** We provide an upper bound on the variability term.

$$\ln \mathbb{E}_\tau \exp \left( -\eta_t \left( \ell_t(\tau \mid \{\tau_t\}) - \mathbb{E}_{\bar{\tau}} \ell_t(\bar{\tau} \mid \{\tau_t\}) \right) \right)$$

$$= \ln \mathbb{E}_\tau \exp \left( -\eta_t \ell_t(\tau \mid \{\tau_t\}) + \eta_t \mathbb{E}_{\bar{\tau}} \ell(\bar{\tau} \mid \{\tau_t\}) \right)$$

$$= \ln \mathbb{E}_\tau \exp \left( -\eta_t \ell_t(\tau \mid \{\tau_t\}) \right) + \ln \exp \left( \eta_t \mathbb{E}_{\bar{\tau}} \ell(\bar{\tau} \mid \{\tau_t\}) \right)$$

$$\leq \mathbb{E}_\tau \exp \left( -\eta_t \ell_t(\tau \mid \{\tau_t\}) \right) - 1 + \eta_t \mathbb{E}_{\bar{\tau}} \ell(\bar{\tau} \mid \{\tau_t\}) \tag{25}$$

$$\leq \mathbb{E}_\tau \left[ \exp \left( -\eta_t \ell_t(\tau \mid \{\tau_t\}) \right) - 1 + \eta_t \ell(\tau \mid \{\tau_t\}) \right]$$

$$\leq \mathbb{E}_\tau \eta_t^2 \frac{\ell_t(\tau \mid \{\tau_t\})^2}{2} \tag{26}$$

$$= \frac{\eta_t^2}{2} \sum_{\tau \in \mathcal{H}} p_t(\tau) \ell_t(\tau \mid \{\tau_t\})^2$$

$$= \frac{\eta_t^2}{2} \sum_{\tau \in \mathcal{H}} p_t(\tau) \left( \frac{\ell_t(\tau)}{p_t(\tau) + \gamma_t} \cdot \mathbb{1}(\tau_t = \tau) \right) \ell_t(\tau \mid \{\tau_t\})$$

$$\leq \frac{\ell_{\max} \eta_t^2}{2} \sum_{\tau \in \mathcal{H}} \ell_t(\tau \mid \{\tau_t\}), \tag{27}$$

where (25) holds due to $\ln(x) \leq x - 1$ for $x \leq 1$, and (26) uses $\exp(x) \leq 1 + x + x^2/2$ for $x \leq 0$.

**Third step.** Let $\tilde{L}_t(\tau) := \sum_{i=1}^t \ell_i(\tau \mid \{\tau_i\})$ and $\tilde{L}_0(\tau) = 0$. Then, we have

$$-\frac{1}{\eta_t} \ln \mathbb{E}_{\tau \sim p_t} \exp\left(-\eta_t \ell_t(\tau \mid \{\tau_t\})\right)$$

$$= -\frac{1}{\eta_t} \ln \sum_{\tau \in \mathcal{H}} p_t(\tau) \exp\left(-\eta_t \ell_t(\tau \mid \{\tau_t\})\right)$$

$$= -\frac{1}{\eta_t} \ln \sum_{\tau \in \mathcal{H}} \frac{\exp(-\eta_t \tilde{L}_{t-1}(\tau))}{\sum_{\bar{\tau} \in \mathcal{H}} \exp(-\eta_t \tilde{L}_{t-1}(\bar{\tau}))} \exp\left(-\eta_t \ell_t(\tau \mid \{\tau_t\})\right)$$

$$= -\frac{1}{\eta_t} \ln \frac{\sum_{\tau \in \mathcal{H}} \exp(-\eta_t \tilde{L}_t(\tau))}{\sum_{\tau \in \mathcal{H}} \exp(-\eta_t \tilde{L}_{t-1}(\tau))}. \tag{28}$$

**Fourth step.** We combine rewritten forms to continue the derivation as follows:

$$\sum_{t=1}^T \mathbb{E}_{\tau \sim p_t} \ell(\tau \mid \{\tau_t\})$$

$$\leq \sum_{t=1}^T \frac{\ell_{\max} \eta_t}{2} \sum_{\tau \in \mathcal{H}} \ell_t(\tau \mid \{\tau_t\}) - \sum_{t=1}^T \frac{1}{\eta_t} \ln \frac{\sum_{\tau \in \mathcal{H}} \exp\left(-\eta_t \tilde{L}_t(\tau)\right)}{\sum_{\tau \in \mathcal{H}} \exp\left(-\eta_t \tilde{L}_{t-1}(\tau)\right)} \tag{29}$$

$$\leq \sum_{t=1}^T \frac{\ell_{\max} \eta_t}{2} \sum_{\tau \in \mathcal{H}} \ell_t(\tau \mid \{\tau_t\}) - \frac{1}{\eta_T} \sum_{t=1}^T \ln \frac{\sum_{\tau \in \mathcal{H}} \exp\left(-\eta_t \tilde{L}_t(\tau)\right)}{\sum_{\tau \in \mathcal{H}} \exp\left(-\eta_t \tilde{L}_{t-1}(\tau)\right)} \tag{30}$$

$$= \sum_{t=1}^T \frac{\ell_{\max} \eta_t}{2} \sum_{\tau \in \mathcal{H}} \ell_t(\tau \mid \{\tau_t\}) - \frac{1}{\eta_T} \ln \frac{\sum_{\tau \in \mathcal{H}} \exp\left(-\eta_T \tilde{L}_T(\tau)\right)}{\sum_{\tau \in \mathcal{H}} \exp\left(-\eta_0 \tilde{L}_0(\tau)\right)}$$

$$= \sum_{t=1}^T \frac{\ell_{\max} \eta_t}{2} \sum_{\tau \in \mathcal{H}} \ell_t(\tau \mid \{\tau_t\}) - \frac{1}{\eta_T} \ln \sum_{\tau \in \mathcal{H}} \exp\left(-\eta_T \tilde{L}_T(\tau)\right) + \frac{\ln |\mathcal{H}|}{\eta_T}$$

$$\leq \sum_{t=1}^T \frac{\ell_{\max} \eta_T}{2} \sum_{\tau \in \mathcal{H}} \ell_t(\tau \mid \{\tau_t\}) - \frac{1}{\eta_T} \ln \left(\max_\tau \exp\left(-\eta_T \tilde{L}_T(\tau)\right)\right) + \frac{\ln |\mathcal{H}|}{\eta_T}$$

$$= \sum_{t=1}^T \frac{\ell_{\max} \eta_t}{2} \sum_{\tau \in \mathcal{H}} \ell_t(\tau \mid \{\tau_t\}) + \min_\tau \sum_{t=1}^T \ell_t(\tau \mid \{\tau_t\}) + \frac{\ln |\mathcal{H}|}{\eta_T}, \tag{31}$$

where (29) holds from (24), (27), and (28). Also, (30) holds as $\eta_t$ is non-increasing and $\frac{\sum_{\tau \in \mathcal{H}} \exp(-\eta_t \tilde{L}_t(\tau))}{\sum_{\tau \in \mathcal{H}} \exp(-\eta_t \tilde{L}_{t-1}(\tau))} \leq 1$.

Note that $\mathbb{E}_{\tau \sim p_t} \ell(\tau \mid \{\tau_t\}) = \ell_t(\tau_t) - \gamma_t \sum_{\tau \in \mathcal{H}} \ell(\tau \mid \{\tau_t\})$ for the following reason:

$$\mathbb{E}_{\tau \sim p_t} \ell_t(\tau \mid \{\tau_t\}) = \sum_{\tau \in \mathcal{H}} p_t(\tau) \frac{\ell_t(\tau)}{p_t(\tau) + \gamma_t} \mathbb{1}(\tau_t = \tau)$$

$$= \sum_{\tau \in \mathcal{H}} \left(\frac{p_t(\tau) + \gamma_t}{p_t(\tau) + \gamma_t} \ell_t(\tau) \mathbb{1}(\tau_t = \tau) - \frac{\gamma_t}{p_t(\tau) + \gamma_t} \ell_t(\tau) \mathbb{1}(\tau_t = \tau)\right)$$

$$= \ell_t(\tau_t) - \gamma_t \sum_{\tau \in \mathcal{H}} \ell_t(\tau \mid \{\tau_t\}).$$

From this fact and (31), we have

$$\sum_{t=1}^{T} \ell_t(\tau_t) - \min_{\tau} \sum_{t=1}^{T} \ell_t(\tau \mid \{\tau_t\}) \le \sum_{t=1}^{T} \frac{\ell_{\max}\eta_t}{2} \sum_{\tau \in \mathcal{H}} \ell_t(\tau \mid \{\tau_t\}) + \sum_{t=1}^{T} \gamma_t \sum_{\tau \in \mathcal{H}} \ell(\tau \mid \{\tau_t\}) + \frac{\ln|\mathcal{H}|}{\eta_T}$$

$$= \sum_{t=1}^{T} \left( \frac{\ell_{\max}\eta_t}{2} + \gamma_t \right) \sum_{\tau \in \mathcal{H}} \ell_t(\tau \mid \{\tau_t\}) + \frac{\ln|\mathcal{H}|}{\eta_T}. \tag{32}$$

If $\eta_t \le \frac{2\gamma_t}{\ell_{\max}}$ then it satisfies the Lemma condition *i.e.*, $\alpha_{t,\tau} = \frac{\ell_{\max}\eta_t}{2} + \gamma_t \le 2\gamma_t$, we can apply Lemma 2, from (32) to get

$$\sum_{t=1}^{T} \ell_t(\tau_t) - \min_{\tau} \sum_{t=1}^{T} \ell_t(\tau \mid \{\tau_t\}) \le \sum_{t=1}^{T} \left( \frac{\ell_{\max}\eta_t}{2} + \gamma_t \right) \sum_{\tau \in \mathcal{H}} \ell_t(\tau) + \frac{\ln|\mathcal{H}|}{\eta_T} + \ell_{\max} \ln\frac{2}{\delta}$$

with probability $1 - \delta/2$.

After that, by Corollary 1, we get

$$\sum_{t=1}^{T} \ell_t(\tau_t) - \min_{\tau} \sum_{t=1}^{T} \ell_t(\tau) \le \sum_{t=1}^{T} \ell_t(\tau_t) - \min_{\tau} \sum_{t=1}^{T} \ell_t(\tau \mid \{\tau_t\}) + \frac{\ell_{\max}}{2\gamma} \ln\frac{2|\mathcal{H}|}{\delta}$$

with probability $1 - \delta/2$.

Finally, we get combining the above two via the union bound as follows:

$$\mathbf{Reg}_T = \sum_{t=1}^{T} \ell_t(\tau_t) - \min_{\tau} \sum_{t=1}^{T} \ell_t(\tau) \le \sum_{t=1}^{T} \left( \frac{\ell_{\max}\eta_t}{2} + \gamma_t \right) \sum_{\tau \in \mathcal{H}} \ell_t(\tau) + \frac{\ln|\mathcal{H}|}{\eta_T} + \ell_{\max} \ln\frac{2}{\delta} + \frac{\ell_{\max}}{2\gamma} \ln\frac{2|\mathcal{H}|}{\delta}$$

with probability $1 - \delta$.

The concluding steps of our proof closely follow the analysis presented by (Neu, 2015). Recall that if the learning rate $\eta_t \le \frac{2\gamma_t}{\ell_{\max}}$ it satisfies the conditions of Lemma 2. Then, letting $\eta_t = \frac{2\gamma_t}{\ell_{\max}}$ and $\gamma = \frac{\eta_T \ell_{\max}}{2}$ due to Corollary 1, we can bound the regret bound as follows:

$$\sum_{t=1}^{T} \left( \frac{\ell_{\max}\eta_t}{2} + \gamma_t \right) \sum_{\tau \in \mathcal{H}} \ell_t(\tau) + \frac{\ln|\mathcal{H}|}{\eta_T} + \ell_{\max} \ln\frac{2}{\delta} + \frac{\ell_{\max}}{2\gamma} \ln\frac{2|\mathcal{H}|}{\delta}$$

$$= \sum_{t=1}^{T} \ell_{\max}\eta_t \sum_{\tau \in \mathcal{H}} \ell_t(\tau) + \frac{2\ln|\mathcal{H}|}{\eta_T} + \ell_{\max} \ln\frac{2}{\delta} + \frac{1}{\eta_T} \ln\frac{2}{\delta}$$

$$\le \sum_{t=1}^{T} \ell_{\max}^2 |\mathcal{H}|\eta_t + \frac{2\ln|\mathcal{H}|}{\eta_T} + \ell_{\max} \ln\frac{2}{\delta} + \frac{1}{\eta_T} \ln\frac{2}{\delta}. \tag{33}$$

First, we consider the setting with a known time horizon. In this case, we use a fixed learning rate, $\eta_t = \eta_T$ for all $t$, which simplifies the general regret bound to:

$$\mathbf{Reg}_T \le \ell_{\max}^2 T|\mathcal{H}|\eta_T + \frac{2\ln|\mathcal{H}|}{\eta_T} + \ell_{\max} \ln\frac{2}{\delta} + \frac{1}{\eta_T} \ln\frac{2}{\delta}. \tag{34}$$

Finally, we can get the optimal learning rate $\eta_t = \sqrt{\frac{2\ln|\mathcal{H}| + \ln(2/\delta)}{\ell_{\max}^2 T|\mathcal{H}|}}$. Here, to make the learning rate agnostic to $\delta$, we approximate $\eta_t \approx \sqrt{\frac{2\ln|\mathcal{H}|}{\ell_{\max}^2 T|\mathcal{H}|}}$, which follows the convention in the original analysis (Neu, 2015).

Then, from (34), we obtain the following upper bound:

$$\mathbf{Reg}_T \le \ell_{\max} \left( 2\sqrt{2T|\mathcal{H}|\ln|\mathcal{H}|} + \left( 1 + \sqrt{\frac{T|\mathcal{H}|}{2\ln|\mathcal{H}|}} \right) \ln\frac{2}{\delta} \right)$$

$$= \mathcal{O}\left( 2\ell_{\max}\sqrt{2T|\mathcal{H}|\ln|\mathcal{H}|/\delta} \right). \tag{35}$$

Second, for the setting with a unknown time horizon, we use the learning rate $\eta_t = \sqrt{\frac{\ln|\mathcal{H}|}{\ell_{\max}^2 t |\mathcal{H}|}}$. Then we get general regret bound from (33), noting that $\sum_{t=1}^T \frac{1}{\sqrt{t}} \leq 2\sqrt{T}$

$$\mathbf{Reg}_T \leq \ell_{\max}\left(4\sqrt{T|\mathcal{H}|\ln|\mathcal{H}|} + \left(1 + \sqrt{\frac{T|\mathcal{H}|}{\ln|\mathcal{H}|}}\right)\ln\frac{2}{\delta}\right)$$

$$= \mathcal{O}\left(4\ell_{\max}\sqrt{T|\mathcal{H}|\ln|\mathcal{H}|/\delta}\right).$$

**Expected regret**  The expected regret can be obtained via integrating the deviations in (35), *i.e.*,

$$\mathbb{E}[W] \leq \int_0^2 \frac{1}{2\delta}\mathbb{P}\left(W > \ln\frac{2}{\delta}\right)d\delta.$$

Here, if we take $W = \frac{1}{\ell_{\max}\left(1+\sqrt{T|\mathcal{H}|/(2\ln|\mathcal{H}|)}\right)}\left(\mathbf{Reg}_T - 2\ell_{\max}\sqrt{2T|\mathcal{H}|\ln|\mathcal{H}|}\right) > \ln\frac{2}{\delta}$ with probability $\delta$ then

$$\mathbb{E}[W] \leq 1,$$

which suggests that

$$\mathbb{E}[\mathbf{Reg}_T] \leq \ell_{\max}\left(2\sqrt{2T|\mathcal{H}|\ln|\mathcal{H}|} + 1 + \sqrt{\frac{T|\mathcal{H}|}{2\ln|\mathcal{H}|}}\right) = \mathcal{O}(\ell_{\max}\sqrt{T|\mathcal{H}|\ln|\mathcal{H}|}). \quad (36)$$

### H.3  A Proof of Lemma 1

We derive a lower bound of $\mathbf{Reg}_T$ which consists of $\mathcal{R}_T^{\mathbf{FDR}}$. In particular, from the definition of $\mathbf{Reg}_T$ and loss $\ell_t(\tau, \alpha)$ in (4), we have the following:

$$(1+\lambda)\mathbf{Reg}_T = \sum_{t=1}^T \left[a_t(\tau_t) + \lambda d_t(\tau_t, \alpha)\right] - \min_{\tau \in \mathcal{H}}\sum_{t=1}^T \left[a_t(\tau) + \lambda d_t(\tau, \alpha)\right]$$

$$\geq \sum_{t=1}^T \left[a_t(\tau_t) + \lambda d_t(\tau_t, \alpha)\right] - \sum_{t=1}^T a_t(\overline{\tau}) - \sum_{t=1}^T \lambda d_t(\overline{\tau}, \alpha) \quad (37)$$

$$= \sum_{t=1}^T \left[a_t(\tau_t) - a_t(\overline{\tau})\right] + \sum_{t=1}^T \lambda d_t(\tau_t, \alpha) - \sum_{t=1}^T \lambda d_t(\overline{\tau}, \alpha)$$

$$\geq T(\mathbf{Ineff}_T - 1) + \sum_{t=1}^T \lambda d_t(\tau_t, \alpha) - \sum_{t=1}^T \lambda d_t(\overline{\tau}, \alpha), \quad (38)$$

where (37) holds as $\overline{\tau} = \arg\min_{\tau \in \mathcal{H}}\sum_{t=1}^T \lambda d_t(\tau, \alpha)$, implying

$$\min_{\tau \in \mathcal{H}}\sum_{t=1}^T \left[a_t(\tau) + \lambda d_t(\tau, \alpha)\right] \leq \sum_{t=1}^T \left[a_t(\overline{\tau}) + \lambda d_t(\overline{\tau}, \alpha)\right]$$

and (38) holds as $a_t(\tau) \in [0, 1]$ for any $t$ and $\tau$.

This implies that

$$\sum_{t=1}^T d_t(\tau_t, \alpha) \leq \frac{T(1 - \mathbf{Ineff}_T) + (1+\lambda)\mathbf{Reg}_T}{\lambda} + \min_{\tau \in \mathcal{H}}\sum_{t=1}^T d_t(\tau, \alpha)$$

Then, by the definition of $d_t(\tau_t, \alpha)$, we have

$$\frac{1}{T}\sum_{t=1}^T \left[\mathbb{1}\left(\hat{S}(\mathbf{x}_t; \tau_t) \neq \texttt{IDK} \wedge e_t\right) - \alpha\mathbb{1}\left(\hat{S}(\mathbf{x}_t; \tau_t) \neq \texttt{IDK}\right) + \alpha\right]$$

$$\leq \frac{(1 - \mathbf{Ineff}_T) + (1+\lambda)\mathbf{Reg}_T/T}{\lambda} + \min_{\tau \in \mathcal{H}}\frac{1}{T}\sum_{t=1}^T d_t(\tau, \alpha),$$

which implies

$$\frac{1}{T}\mathcal{R}_T^{\textbf{FDR}} = \frac{1}{T}\sum_{t=1}^{T}\left[\mathbb{1}\left(\hat{S}(\mathbf{x}_t;\tau_t) \neq \texttt{IDK} \wedge e_t\right) - \alpha\mathbb{1}\left(\hat{S}(\mathbf{x}_t;\tau_t) \neq \texttt{IDK}\right)\right]$$

$$\leq \frac{(1 - \textbf{Ineff}_T) + (1 + \lambda)\textbf{Reg}_T/T}{\lambda} + \min_{\tau \in \mathcal{H}}\frac{1}{T}\sum_{t=1}^{T}d_t(\tau, \alpha) - \alpha$$

$$\leq \frac{(1 - \textbf{Ineff}_T) + (1 + \lambda)\textbf{Reg}_T/T}{\lambda}, \tag{39}$$

where (39) holds as $\tau' = 1$ (*i.e.,* always abstaining) satisfies the following:

$$\min_{\tau \in \mathcal{H}}\frac{1}{T}\sum_{t=1}^{T}d_t(\tau, \alpha) \leq \frac{1}{T}\sum_{t=1}^{T}d_t(\tau', \alpha) = \alpha.$$

If we take $\lambda = T^{1/4}$,

$$\frac{1}{T}\mathcal{R}_T^{\textbf{FDR}} \leq \frac{(1 - \textbf{Ineff}_T)}{T^{1/4}} + \frac{(1 + T^{1/4})\textbf{Reg}_T}{T^{5/4}},$$

which completes the proof.

**Remark.** *Finally, we clarify that $\textbf{Ineff}_T$ depends on the capability of $G$, the calibration of $f$, and the choice of $\lambda$. In particular, a larger $\lambda$ accelerates convergence but also increases $\textbf{Ineff}_T$. Therefore, in practice, choosing a moderately small $\lambda$ can achieve high efficiency while still satisfying the desired FDR under fast convergence.*

*Nevertheless, if the model or the scoring function $f$ is severely inaccurate, one may heuristically increase $\lambda$ to a large value so that the learner approaches the performance of the best expert that always guarantees the FDR if achievable (Section I for discussion on selection efficiency). In this sense, the exponent $d \in \mathbb{R}^+$ in $\lambda = \mathcal{O}(T^d)$ acts as a hyperparameter of the algorithm.*

*In our implementation and experiments, we set $\lambda = T^{1/4}$. For empirical trends under different values of $\lambda$, see Figure 26 and 27.*

### H.4 A PROOF OF THEOREM 1

Here, we derive a novel regret bound of our algorithm, Theorem 4, which is a general case of our main Theorem 1 when the loss range is arbitrarily bounded, *i.e.,* $\in \ell_t(\cdot) \in [0, \ell_{\max}]$ and the learning rate $\eta$ is non-increasing function in time $t$.

**Theorem 4.** *Let $\ell_t(\cdot) \in [0, \ell_{max}]$ with the form of (4). For any $T \in \mathbb{N}$ and finite hypotheses $\mathcal{H}$, Algorithm 1 provides the following regret bound with probability at least $1 - \delta$ if $\eta_t = \frac{2\gamma_t}{\ell_{max}} = \sqrt{\frac{\ln|\mathcal{H}|}{\ell_{max}^2 T}}$*

$$\textbf{Reg}_T \leq \ell_{max}\left(4\sqrt{T\ln|\mathcal{H}|} + \left(1 + \sqrt{\frac{T}{\ln|\mathcal{H}|}}\right)\ln\frac{2}{\delta}\right).$$

The proof technique follows that of the $\texttt{Exp3-IX}$ (Neu, 2015) regret bound except that we use a novel loss estimator for feedback unlocking. Note that we abbreviate $\ell_t(\tau, \alpha \mid \mathcal{H}_t(\tau_t))$ as $\ell_t(\tau \mid \mathcal{H}_t(\tau_t))$ in this proof.

First, we introduce some properties of $\mathcal{H}_t(\tau_t)$ in our algorithm.

1. (self-inclusion) The first trivial property is that $\tau$ is in $\mathcal{H}_t(\tau)$ itself:
$$\tau \in \mathcal{H}_t(\tau). \tag{40}$$

2. (partition) The second property is that if $\tau' \in \mathcal{H}_t(\tau)$, then we have
$$\mathcal{H}_t(\tau') = \mathcal{H}_t(\tau). \tag{41}$$
This holds since if $\tau' \in \mathcal{H}_t(\tau)$ and $\tau \in \mathcal{H}_t(\tau)$ due to (40), $\tau$ and $\tau'$ are in the same side from $f_t$ (*i.e.,* either $\tau \leq f_t \wedge \tau' \leq f_t$ or $\tau > f_t \wedge \tau' > f_t$), thus $\hat{S}(\mathbf{x}_t;\tau) = \hat{S}(\mathbf{x}_t;\tau')$, implying $\mathcal{H}_t(\tau') = \mathcal{H}_t(\tau)$ by our construction of $\mathcal{H}_t(\cdot)$ in our algorithm. The second case when $\tau \in \mathcal{H}_t(\tau')$ similarly holds as well.

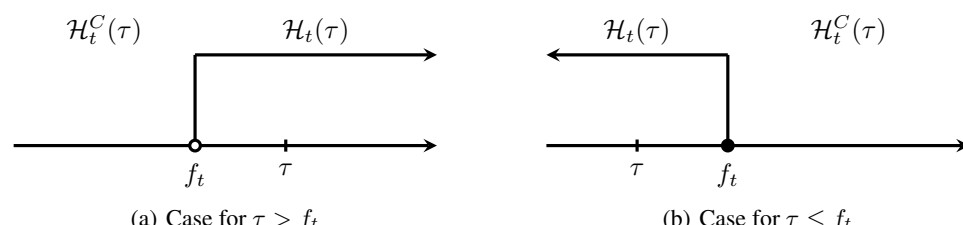

Figure 33: Visualization of $\mathcal{H}_t(\tau)$.

3. (complement) The third property, similar to the second one, is that if $\tau' \in \mathcal{H}_t^C(\tau)$, then we have

$$\mathcal{H}_t^C(\tau) = \mathcal{H}_t(\tau'), \text{ or equivalently } \mathcal{H}_t(\tau) = \mathcal{H}_t^C(\tau'). \tag{42}$$

This property holds as follows: $\tau' \in \mathcal{H}_t(\tau')$ holds due to (40). Suppose $\tau' \in \mathcal{H}_t^C(\tau)$, then we have $\tau' \notin \mathcal{H}_t(\tau)$. This along with $\tau' \in \mathcal{H}_t(\tau')$ implies $\mathcal{H}_t(\tau') \neq \mathcal{H}_t(\tau)$. Since $\mathcal{H}_t(\tau_t)$ is either $\{\tau \mid \tau \leq f_t\}$ or $\{\tau \mid \tau > f_t\}$ depending on the value of $\tau_t$, $\mathcal{H}_t(\cdot)$ partitions the space $[0, 1]$ into two disjoint sets based on $f_t$, where one set is the complement of the other. Therefore, $\mathcal{H}_t(\tau) \neq \mathcal{H}_t(\tau')$ is equivalent to either $\mathcal{H}_t^C(\tau) = \mathcal{H}_t(\tau')$ or $\mathcal{H}_t(\tau) = \mathcal{H}_t^C(\tau')$.

4. (swap) The final property is that

$$\tau' \in \mathcal{H}_t(\tau) \quad \text{if and only if} \quad \tau \in \mathcal{H}_t(\tau'). \tag{43}$$

Suppose that $\tau' \in \mathcal{H}_t(\tau)$ and $\tau' \in \mathcal{H}_t(\tau')$, then the above holds as follows:

$$\tau \in \mathcal{H}_t(\tau) = \mathcal{H}_t(\tau'),$$

where the inclusion holds due to (40) and the equality holds due to (41) if $\tau', \tau \in \mathcal{H}_t(\tau)$. The reverse holds similarly as well.

With these properties, we can derive the following two useful equalities. Let

$$\hat{\ell}_t(\tau \mid \mathcal{H}(\tau_t)) = \frac{\ell_t(\tau)}{\sum_{\bar{\tau} \in \mathcal{H}_t(\tau_t)} \mathbb{1}(\tau \in \mathcal{H}_t(\bar{\tau})) \cdot p_t(\bar{\tau})} \mathbb{1}(\tau \in \mathcal{H}_t(\tau_t)),$$

which is our loss estimator without $\gamma_t$. Then, the following equality holds:

$$\mathbb{E}_{\tau_t \sim p_t} \hat{\ell}_t(\tau \mid \mathcal{H}_t(\tau_t)) = \sum_{\tau' \in \mathcal{H}} p_t(\tau') \left[ \frac{\ell_t(\tau)}{\sum_{\bar{\tau} \in \mathcal{H}_t(\tau')} \mathbb{1}(\tau \in \mathcal{H}_t(\bar{\tau})) \cdot p_t(\bar{\tau})} \mathbb{1}(\tau \in \mathcal{H}_t(\tau')) \right]$$

$$= \ell_t(\tau) \sum_{\tau' \in \mathcal{H}} \frac{p_t(\tau')}{\sum_{\bar{\tau} \in \mathcal{H}_t(\tau')} \mathbb{1}(\tau \in \mathcal{H}_t(\bar{\tau})) \cdot p_t(\bar{\tau})} \mathbb{1}(\tau \in \mathcal{H}_t(\tau'))$$

$$= \ell_t(\tau) \sum_{\tau' \in \mathcal{H}} \frac{p_t(\tau')}{\sum_{\bar{\tau} \in \mathcal{H}_t(\tau')} \mathbb{1}(\tau \in \mathcal{H}_t(\bar{\tau})) \cdot p_t(\bar{\tau})} \mathbb{1}(\tau' \in \mathcal{H}_t(\tau)) \tag{44}$$

$$= \ell_t(\tau) \sum_{\tau' \in \mathcal{H}_t(\tau)} \frac{p_t(\tau')}{\sum_{\bar{\tau} \in \mathcal{H}_t(\tau')} \mathbb{1}(\tau \in \mathcal{H}_t(\bar{\tau})) \cdot p_t(\bar{\tau})} \tag{45}$$

$$= \ell_t(\tau) \sum_{\tau' \in \mathcal{H}_t(\tau)} \frac{p_t(\tau')}{\sum_{\bar{\tau} \in \mathcal{H}_t(\tau)} \mathbb{1}(\tau \in \mathcal{H}_t(\bar{\tau})) \cdot p_t(\bar{\tau})} \tag{46}$$

$$= \ell_t(\tau) \sum_{\tau' \in \mathcal{H}_t(\tau)} \frac{p_t(\tau')}{\sum_{\bar{\tau} \in \mathcal{H}_t(\tau)} \mathbb{1}(\bar{\tau} \in \mathcal{H}_t(\tau)) \cdot p_t(\bar{\tau})} \tag{47}$$

$$= \ell_t(\tau) \sum_{\tau' \in \mathcal{H}_t(\tau)} \frac{p_t(\tau')}{\sum_{\bar{\tau} \in \mathcal{H}_t(\tau)} p_t(\bar{\tau})}$$

$$= \ell_t(\tau), \tag{48}$$

where (44) holds as (43), (45) follows from the properties of the summation and indicator function, (46) holds as (41) from $\tau' \in \mathcal{H}_t(\tau)$ in the summation, and (47) follows by the same argument as (44) from the property (43), *i.e.*, $\bar{\tau} \in \mathcal{H}_t(\tau)$ if $\tau \in \mathcal{H}_t(\bar{\tau})$.

Also, the following equality with respect to our loss estimator (7) holds similarly:

$$
\begin{aligned}
\mathbb{E}_{\tau \sim p_t} \ell_t(\tau \mid \mathcal{H}_t(\tau_t)) &= \sum_{\tau \in \mathcal{H}} p_t(\tau) \frac{\ell_t(\tau)}{\gamma_t + \sum_{\bar{\tau} \in \mathcal{H}_t(\tau_t)} \mathbb{1}(\tau \in \mathcal{H}_t(\bar{\tau})) p_t(\bar{\tau})} \cdot \mathbb{1}(\tau \in \mathcal{H}_t(\tau_t)) \\
&= \sum_{\tau \in \mathcal{H}_t(\tau_t)} p_t(\tau) \frac{\ell_t(\tau)}{\gamma_t + \sum_{\bar{\tau} \in \mathcal{H}_t(\tau_t)} \mathbb{1}(\tau \in \mathcal{H}_t(\bar{\tau})) p_t(\bar{\tau})} \\
&= \ell_t(\tau_t) \sum_{\tau \in \mathcal{H}_t(\tau_t)} \frac{p_t(\tau)}{\gamma_t + \sum_{\bar{\tau} \in \mathcal{H}_t(\tau_t)} \mathbb{1}(\tau \in \mathcal{H}_t(\bar{\tau})) p_t(\bar{\tau})} &(49) \\
&= \ell_t(\tau_t) \sum_{\tau \in \mathcal{H}_t(\tau_t)} \frac{p_t(\tau)}{\gamma_t + \sum_{\bar{\tau} \in \mathcal{H}_t(\tau_t)} p_t(\bar{\tau})} &(50) \\
&= \ell_t(\tau_t) \frac{\sum_{\tau \in \mathcal{H}_t(\tau_t)} p_t(\tau)}{\gamma_t + \sum_{\bar{\tau} \in \mathcal{H}_t(\tau_t)} p_t(\bar{\tau})} \\
&= \ell_t(\tau_t) \frac{\gamma_t + \sum_{\tau \in \mathcal{H}_t(\tau_t)} p_t(\tau)}{\gamma_t + \sum_{\bar{\tau} \in \mathcal{H}_t(\tau_t)} p_t(\bar{\tau})} - \ell_t(\tau_t) \frac{\gamma_t}{\gamma_t + \sum_{\bar{\tau} \in \mathcal{H}_t(\tau_t)} p_t(\bar{\tau})} &(51) \\
&= \ell_t(\tau_t) - \gamma_t \frac{\ell_t(\tau_t)}{\gamma_t + \sum_{\bar{\tau} \in \mathcal{H}_t(\tau_t)} \mathbb{1}(\tau_t \in \mathcal{H}_t(\bar{\tau})) p_t(\bar{\tau})} \mathbb{1}(\tau_t \in \mathcal{H}_t(\tau_t)) &(52) \\
&= \ell_t(\tau_t) - \gamma_t \ell_t(\tau_t \mid \mathcal{H}_t(\tau_t)), &(53)
\end{aligned}
$$

where (49) holds as $\ell_t(\tau) = \ell_t(\tau')$ for all $\tau \in \mathcal{H}(\tau')$ by the definition of the specialized loss in (4), (50) holds as the same as (47), (51) holds as $\tau_t \in \mathcal{H}_t(\tau_t)$ and (43), (52) holds due to the property of the indicator function, and (53) holds by the definition.

Then, we first introduce a new Lemma 3 and Corollary 2 to establish a high probability bound.

**Lemma 3.** *Let $\alpha_t \leq 2\gamma_t$ where $\gamma_t$ is non-increasing in $t$. Then, the following inequality holds with probability at least $1 - \delta$:*

$$
\sum_{t=1}^{T} \alpha_t \ell_t(\tau_t \mid \mathcal{H}_t(\tau_t)) - \sum_{t=1}^{T} 2\alpha_t \ell_{max} \leq \ell_{max} \ln(1/\delta). \tag{54}
$$

*Proof.* First, the following inequality holds for any $\tau \in \mathcal{H}$:

$$
\ell_t(\tau \mid \mathcal{H}_t(\tau_t)) = \frac{\ell_t(\tau)}{\gamma_t + \sum_{\bar{\tau} \in \mathcal{H}_t(\tau_t)} \mathbb{1}(\tau \in \mathcal{H}_t(\bar{\tau})) \cdot p_t(\bar{\tau})} \cdot \mathbb{1}(\tau \in \mathcal{H}_t(\tau_t))
$$

$$
= \ell_{\max} \cdot \frac{\ell_t(\tau)/\ell_{\max}}{\gamma_t + \sum_{\bar{\tau} \in \mathcal{H}_t(\tau_t)} \mathbb{1}(\tau \in \mathcal{H}_t(\bar{\tau})) \cdot p_t(\bar{\tau})} \cdot \mathbb{1}(\tau \in \mathcal{H}_t(\tau_t))
$$

$$
\leq \ell_{\max} \cdot \frac{\ell_t(\tau)/\ell_{\max}}{\gamma_t \ell_t(\tau)/\ell_{\max} + \sum_{\bar{\tau} \in \mathcal{H}_t(\tau_t)} \mathbb{1}(\tau \in \mathcal{H}_t(\bar{\tau})) \cdot p_t(\bar{\tau})} \cdot \mathbb{1}(\tau \in \mathcal{H}_t(\tau_t))
$$

$$
= \frac{\ell_{\max}}{2\gamma_t} \cdot \frac{2\gamma_t \ell_t(\tau)/\ell_{\max}}{\gamma_t \ell_t(\tau)/\ell_{\max} + \sum_{\bar{\tau} \in \mathcal{H}_t(\tau_t)} \mathbb{1}(\tau \in \mathcal{H}_t(\bar{\tau})) \cdot p_t(\bar{\tau})} \cdot \mathbb{1}(\tau \in \mathcal{H}_t(\tau_t))
$$

$$
= \frac{1}{\beta_t} \cdot \frac{\frac{\beta_t \ell_t(\tau)}{\sum_{\bar{\tau} \in \mathcal{H}_t(\tau_t)} \mathbb{1}(\tau \in \mathcal{H}_t(\bar{\tau})) \cdot p_t(\bar{\tau})}}{\frac{\beta \ell_t(\tau)}{2 \sum_{\bar{\tau} \in \mathcal{H}_t(\tau_t)} \mathbb{1}(\tau \in \mathcal{H}_t(\bar{\tau})) \cdot p_t(\bar{\tau})} + 1} \cdot \mathbb{1}(\tau \in \mathcal{H}_t(\tau_t))
$$

$$
= \frac{1}{\beta_t} \cdot \frac{\frac{\beta_t \ell_t(\tau)}{\sum_{\bar{\tau} \in \mathcal{H}_t(\tau_t)} \mathbb{1}(\tau \in \mathcal{H}_t(\bar{\tau})) \cdot p_t(\bar{\tau})} \cdot \mathbb{1}(\tau \in \mathcal{H}_t(\tau_t))}{\frac{\beta_t \ell_t(\tau)}{2 \sum_{\bar{\tau} \in \mathcal{H}_t(\tau_t)} \mathbb{1}(\tau \in \mathcal{H}_t(\bar{\tau})) \cdot p_t(\bar{\tau})} \cdot \mathbb{1}(\tau \in \mathcal{H}_t(\tau_t)) + 1}
$$

$$
= \frac{1}{\beta_t} \cdot \frac{\beta_t \hat{\ell}_t(\tau \mid \mathcal{H}_t(\tau_t))}{\beta_t \hat{\ell}_t(\tau \mid \mathcal{H}_t(\tau_t))/2 + 1}
$$

$$
\leq \frac{1}{\beta_t} \cdot \ln\left(1 + \beta_t \hat{\ell}_t(\tau \mid \mathcal{H}_t(\tau_t))\right), \tag{55}
$$

where $\beta_t = 2\gamma_t/\ell_{\max}$, $\hat{\ell}_t(\tau \mid \mathcal{H}_t(\tau_t)) := \frac{\ell_t(\tau)}{\sum_{\bar{\tau} \in \mathcal{H}_t(\tau_t)} \mathbb{1}(\tau \in \mathcal{H}_t(\bar{\tau})) \cdot p_t(\bar{\tau})} \cdot \mathbb{1}(\tau \in \mathcal{H}_t(\tau_t))$, and the last inequality holds as $\frac{x}{1+x/2} \leq \ln(1+x)$ for all $x \geq 0$.

Let $\mathbb{E}_t$ be the expectation conditioned on $\tau_1, \ldots, \tau_{t-1}$ and $\tau'$ be any element in $\mathcal{H}$. Then, we have

$$\mathbb{E}_t \left[ \exp \left( \frac{\alpha_t}{\ell_{\max}} \ell_t(\tau_t \mid \mathcal{H}_t(\tau_t)) \right) \right]$$

$$\leq \mathbb{E}_t \left[ \exp \left( \frac{\alpha_t}{\ell_{\max}\beta_t} \ln \left( 1 + \beta_t \hat{\ell}_t(\tau_t \mid \mathcal{H}_t(\tau_t)) \right) \right) \right] \tag{56}$$

$$\leq \mathbb{E}_t \left[ 1 + \frac{\alpha_t}{\ell_{\max}} \hat{\ell}_t(\tau_t \mid \mathcal{H}_t(\tau_t)) \right] \tag{57}$$

$$= 1 + \frac{\alpha_t}{\ell_{\max}} \mathbb{E}_t \hat{\ell}_t(\tau_t \mid \mathcal{H}_t(\tau_t))$$

$$= 1 + \frac{\alpha_t}{\ell_{\max}} \sum_{\tau_t \in \mathcal{H}} p_t(\tau_t) \frac{\ell_t(\tau)}{\sum_{\bar{\tau} \in \mathcal{H}_t(\tau_t)} \mathbb{1}(\tau \in \mathcal{H}_t(\bar{\tau})) \cdot p_t(\bar{\tau})} \cdot \mathbb{1}(\tau \in \mathcal{H}_t(\tau_t))$$

$$= 1 + \frac{\alpha_t}{\ell_{\max}} \sum_{\tau_t \in \mathcal{H}} p_t(\tau_t) \frac{\ell_t(\tau_t)}{\sum_{\bar{\tau} \in \mathcal{H}_t(\tau_t)} p_t(\bar{\tau})}$$

$$= 1 + \frac{\alpha_t}{\ell_{\max}} \left( \sum_{\tau \in \mathcal{H}_t(\tau')} \frac{p_t(\tau)\ell_t(\tau)}{\sum_{\bar{\tau} \in \mathcal{H}_t(\tau)} p_t(\bar{\tau})} + \sum_{\tau \in \mathcal{H}_t^C(\tau')} \frac{p_t(\tau)\ell_t(\tau)}{\sum_{\bar{\tau} \in \mathcal{H}_t(\tau)} p_t(\bar{\tau})} \right)$$

$$\leq 1 + \frac{\alpha_t}{\ell_{\max}} \ell_{\max} \left( \sum_{\tau \in \mathcal{H}_t(\tau')} \frac{p_t(\tau)}{\sum_{\bar{\tau} \in \mathcal{H}_t(\tau)} p_t(\bar{\tau})} + \sum_{\tau \in \mathcal{H}_t^C(\tau')} \frac{p_t(\tau)}{\sum_{\bar{\tau} \in \mathcal{H}_t(\tau)} p_t(\bar{\tau})} \right)$$

$$= 1 + \alpha_t \left( \sum_{\tau \in \mathcal{H}_t(\tau')} \frac{p_t(\tau)}{\sum_{\bar{\tau} \in \mathcal{H}_t(\tau')} p_t(\bar{\tau})} + \sum_{\tau \in \mathcal{H}_t^C(\tau')} \frac{p_t(\tau)}{\sum_{\bar{\tau} \in \mathcal{H}_t(\tau)} p_t(\bar{\tau})} \right) \tag{58}$$

$$= 1 + \alpha_t \left( \sum_{\tau \in \mathcal{H}_t(\tau')} \frac{p_t(\tau)}{\sum_{\bar{\tau} \in \mathcal{H}_t(\tau')} p_t(\bar{\tau})} + \sum_{\tau \in \mathcal{H}_t^C(\tau')} \frac{p_t(\tau)}{\sum_{\bar{\tau} \in \mathcal{H}_t^C(\tau')} p_t(\bar{\tau})} \right) \tag{59}$$

$$\leq 1 + 2\alpha_t \tag{60}$$

$$\leq \exp(2\alpha_t), \tag{61}$$

where (56 holds as (55). (57) uses $x \ln(1 + y) \leq \ln(1 + xy)$ for all $y > -1$ and $x \in [0, 1]$ since $\alpha_t \leq 2\gamma_t = \ell_{\max}\beta_t$. (58) holds as (41), *i.e.*, $\mathcal{H}_t(\tau) = \mathcal{H}_t(\tau_t)$ due to $\tau \in \mathcal{H}_t(\tau_t)$ in the summation. Also, (59) holds due to (42), *i.e.*, $\mathcal{H}_t(\tau) = \mathcal{H}_t^C(\tau_t)$ as $\tau \in \mathcal{H}_t^C(\tau_t)$ from the summation condition. Finally, (60) holds with the equality unless $\mathcal{H}_t(\tau) = \emptyset$ or $\mathcal{H}_t^C(\tau) = \emptyset$, *i.e.*, $f_t = 0$, in which case the inequality holds.

Thus, we can find that $W_t := \exp \left( \sum_{s=1}^t \frac{\alpha_s}{\ell_{\max}} (\ell_s(\tau_s \mid \mathcal{H}_s(\tau_s)) - 2\ell_{\max}) \right)$ is a supermartingale with respect to $\mathbb{E}_t$, *i.e.*, $\mathbb{E}_t[W_t] \leq W_{t-1}$ as follows:

$$\mathbb{E}_t[W_t] = \mathbb{E}_t \left[ W_{t-1} \exp \left( \frac{\alpha_s}{\ell_{\max}} (\ell_t(\tau_t \mid \mathcal{H}_t(\tau_t)) - 2\ell_{\max}) \right) \right]$$

$$= W_{t-1} \mathbb{E}_t \left[ \exp \left( \frac{\alpha_s}{\ell_{\max}} (\ell_t(\tau_t \mid \mathcal{H}_t(\tau_t)) - 2\ell_{\max}) \right) \right]$$

$$= W_{t-1} \mathbb{E}_t \left[ \exp \left( \frac{\alpha_s}{\ell_{\max}} \ell_t(\tau_t \mid \mathcal{H}_t(\tau_t)) \right) \right] \exp(-2\alpha_s)$$

$$\leq W_{t-1} \exp(2\alpha_t) \exp(-2\alpha_s) \tag{62}$$

$$= W_{t-1},$$

where (62) holds due to (61). Due to $W_0 = 1$ by the definition, the above implies $\mathbb{E}[W_T] \leq 1$. Then, by applying this with the Markov's inequality, we have

$$\mathbb{P} \left( \sum_{t=1}^T \frac{\alpha_t}{\ell_{\max}} (\ell_t(\tau_t \mid \mathcal{H}_t(\tau_t)) - 2\ell_{\max}) > \ln \frac{1}{\delta} \right) = \mathbb{P} \left( W_T > \frac{1}{\delta} \right) \leq \delta \mathbb{E}_T[W_T] \leq \delta.$$

Thus,

$$\sum_{t=1}^{T} \frac{\alpha_t}{\ell_{\max}} \left( \ell_t(\tau_t \mid \mathcal{H}_t(\tau_t)) - 2\ell_{\max} \right) \leq \ln(1/\delta),$$

with probability at least $1 - \delta$, which completes the proof. $\qquad\square$

**Corollary 2.** *Let $\gamma_t = \gamma > 0$ for all $t$. Then, simultaneously for all $\tau \in \mathcal{H}$,*

$$\sum_{t=1}^{T} \ell_t(\tau \mid \mathcal{H}_t(\tau_t)) - \sum_{t=1}^{T} \ell_t(\tau) \leq \frac{\ell_{max} \ln(|\mathcal{H}|/\delta)}{2\gamma}, \tag{63}$$

*with probability at least $1 - \delta$.*

*Proof.* The proof of this corollary can be derived from Lemma 3, but here we provide a direct proof, which uses similar proof techniques as in Lemma 3.

Let $\gamma_t = \gamma \geq 0$. Then, we have

$$\mathbb{E}_t \left[ \exp \left( \frac{2\gamma}{\ell_{\max}} \ell_t(\tau \mid \mathcal{H}_t(\tau_t)) \right) \right] \leq \mathbb{E}_t \left[ \exp \left( \frac{2\gamma}{\ell_{\max}\beta} \ln \left( 1 + \beta \hat{\ell}_t(\tau \mid \mathcal{H}_t(\tau_t)) \right) \right) \right] \tag{64}$$

$$\leq \mathbb{E}_t \left[ 1 + \frac{2\gamma}{\ell_{\max}} \hat{\ell}_t(\tau \mid \mathcal{H}_t(\tau_t)) \right] \tag{65}$$

$$= 1 + \frac{2\gamma}{\ell_{\max}} \ell_t(\tau) \tag{66}$$

$$\leq \exp \left( \frac{2\gamma}{\ell_{\max}} \ell_t(\tau) \right), \tag{67}$$

where (64) holds same as (55) with fixed $\gamma_t = \gamma$ and $\beta_t = \beta$ for all $t$. Also, (65) uses $x \ln(1 + y) \leq \ln(1 + xy)$ for all $y > -1$ and $x \in [0, 1]$ along with the fact that $2\gamma = \ell_{\max}\beta$ and thus $\frac{2\gamma}{\beta\ell_{\max}} = 1$. (66) holds as $\mathbb{E}_t \hat{\ell}_t(\tau \mid \mathcal{H}_t(\tau_t)) = \ell_t(\tau)$ from 48.

Thus, we can find that $W_t := \exp \left( \sum_{s=1}^{t} \frac{2\gamma}{\ell_{\max}} \left( \ell_s(\tau \mid \mathcal{H}_s(\tau_s)) - \ell_s(\tau) \right) \right)$ is a supermartingale with respect to $\mathbb{E}_t$, *i.e.*, $\mathbb{E}[W_t] \leq W_{t-1}$ as follows:

$$\mathbb{E}_t[W_t] = \mathbb{E}_t \left[ W_{t-1} \cdot \exp \left( \frac{2\gamma}{\ell_{\max}} \left( \ell_t(\tau \mid \mathcal{H}_t(\tau_t)) - \ell_t(\tau) \right) \right) \right]$$

$$= W_{t-1} \mathbb{E}_t \left[ \exp \left( \frac{2\gamma}{\ell_{\max}} \left( \ell_t(\tau \mid \mathcal{H}_t(\tau_t)) - \ell_t(\tau) \right) \right) \right]$$

$$= W_{t-1} \mathbb{E}_t \left[ \exp \left( \frac{2\gamma}{\ell_{\max}} \ell_t(\tau \mid \mathcal{H}_t(\tau_t)) \right) \right] \exp \left( -\frac{2\gamma}{\ell_{\max}} \ell_t(\tau) \right)$$

$$\leq W_{t-1} \exp \left( \frac{2\gamma}{\ell_{\max}} \ell_t(\tau) \right) \exp \left( -\frac{2\gamma}{\ell_{\max}} \ell_t(\tau) \right) \tag{68}$$

$$= W_{t-1}.$$

where (68) holds as (67). From this and $W_0 = 1$ by the definition, we have $\mathbb{E}[W_T] \leq \mathbb{E}[W_{T-1}] \leq \cdots \leq 1$. By this and the Markov's inequality, we have

$$\mathbb{P} \left( \sum_{t=1}^{T} \frac{2\gamma}{\ell_{\max}} \left( \ell_t(\tau_t \mid \mathcal{H}_t(\tau_t)) - \ell_t(\tau) \right) > \ln \frac{1}{\delta} \right) = \mathbb{P} \left( W_T > \frac{1}{\delta} \right) \leq \delta \mathbb{E}_T[W_T] \leq \delta.$$

Then, by the union bound, for all $\tau$, we have

$$\sum_{t=1}^{T} \left( \ell_t(\tau \mid \mathcal{H}_t(\tau_t)) - \ell_t(\tau) \right) \leq \frac{\ell_{\max} \ln(|\mathcal{H}|/\delta)}{2\gamma}$$

with probability at least $1 - \delta$, which completes the proof.

$\qquad\square$

We then prove our main theorem, which consists of four steps.

**First step.** We split the expected loss estimator into two logarithmic terms (Bubeck et al., 2012, Eq. (3.7)):

$$
\mathbb{E}_{\tau \sim p_t} \ell_t(\tau \mid \mathcal{H}_t(\tau_t)) = \frac{1}{\eta_t} \underbrace{\ln \mathbb{E}_{\tau \sim p_t} \exp\left(-\eta_t \Big(\ell_t(\tau \mid \mathcal{H}_t(\tau_t)) - \mathbb{E}_{\bar{\tau} \sim p_t} \ell_t(\bar{\tau} \mid \mathcal{H}_t(\tau_t))\Big)\right)}_{\text{variability term}}
$$

$$
- \frac{1}{\eta_t} \underbrace{\ln \mathbb{E}_{\tau \sim p_t} \exp\left(-\eta_t \ell_t(\tau \mid \mathcal{H}_t(\tau_t))\right)}_{\text{log of an exponential expectation term}}, \quad (69)
$$

where $\eta_t$ is non-increasing in $t \in \mathbb{N}$. In the following steps, each term will be rewritten.

**Second step.** We provide an upper bound on the first term:

$$
\ln \mathbb{E}_{\tau \sim p_t} \exp\left(-\eta_t \Big(\ell_t(\tau \mid \mathcal{H}_t(\tau_t)) - \mathbb{E}_{\bar{\tau} \sim w_t} \ell_t(\bar{\tau} \mid \mathcal{H}_t(\tau_t))\Big)\right)
$$

$$
= \ln \mathbb{E}_{\tau \sim p_t} \exp\left(-\eta_t \ell_t(\tau \mid \mathcal{H}_t(\tau_t))\right) + \eta_t \mathbb{E}_{\tau \sim p_t} \ell_t(\bar{\tau} \mid \mathcal{H}_t(\tau_t))
$$

$$
\leq \mathbb{E}_{\tau \sim p_t} \left[\exp\left(-\eta_t \ell_t(\tau \mid \mathcal{H}_t(\tau_t))\right) - 1 + \eta_t \ell_t(\tau \mid \mathcal{H}_t(\tau_t))\right] \quad (70)
$$

$$
\leq \mathbb{E}_{\tau \sim p_t} \eta_t^2 \frac{\ell_t(\tau \mid \mathcal{H}_t(\tau_t))^2}{2} \quad (71)
$$

$$
\leq \frac{\eta_t^2}{2} \sum_{\tau \in \mathcal{H}} p_t(\tau) \ell_t(\tau \mid \mathcal{H}_t(\tau_t))^2
$$

$$
= \frac{\eta_t^2}{2} \sum_{\tau \in \mathcal{H}} p_t(\tau) \left(\frac{\ell_t(\tau)}{\gamma_t + \sum_{\bar{\tau} \in \mathcal{H}_t(\tau_t)} \mathbb{1}(\tau \in \mathcal{H}_t(\bar{\tau})) \cdot p_t(\bar{\tau})} \cdot \mathbb{1}(\tau \in \mathcal{H}_t(\tau_t))\right)^2
$$

$$
= \frac{\eta_t^2}{2} \sum_{\tau \in \mathcal{H}_t(\tau_t)} p_t(\tau) \left(\frac{\ell_t(\tau)}{\gamma_t + \sum_{\bar{\tau} \in \mathcal{H}_t(\tau_t)} \mathbb{1}(\tau \in \mathcal{H}_t(\bar{\tau})) \cdot p_t(\bar{\tau})}\right)^2
$$

$$
= \frac{\eta_t^2}{2} \sum_{\tau \in \mathcal{H}_t(\tau_t)} p_t(\tau) \left(\frac{\ell_t(\tau)}{\gamma_t + \sum_{\bar{\tau} \in \mathcal{H}_t(\tau_t)} p_t(\bar{\tau})}\right)^2 \quad (72)
$$

$$
= \frac{\eta_t^2}{2} \frac{\sum_{\tau \in \mathcal{H}_t(\tau_t)} p_t(\tau) \ell_t(\tau)^2}{\left(\gamma_t + \sum_{\bar{\tau} \in \mathcal{H}_t(\tau_t)} p_t(\bar{\tau})\right)^2}
$$

$$
\leq \frac{\ell_{\max} \eta_t^2}{2} \frac{\sum_{\tau \in \mathcal{H}_t(\tau_t)} p_t(\tau)}{\left(\gamma_t + \sum_{\bar{\tau} \in \mathcal{H}_t(\tau_t)} p_t(\bar{\tau})\right)^2} \ell_t(\tau_t) \quad (73)
$$

$$
\leq \frac{\ell_{\max} \eta_t^2}{2} \frac{\ell_t(\tau_t)}{\gamma_t + \sum_{\bar{\tau} \in \mathcal{H}_t(\tau_t)} p_t(\bar{\tau})}
$$

$$
= \frac{\ell_{\max} \eta_t^2}{2} \frac{\ell_t(\tau)}{\gamma_t + \sum_{\bar{\tau} \in \mathcal{H}_t(\tau_t)} \mathbb{1}(\tau \in \mathcal{H}_t(\bar{\tau})) \cdot p_t(\bar{\tau})} \cdot \mathbb{1}(\tau \in \mathcal{H}_t(\tau_t)), \quad (74)
$$

$$
= \frac{\ell_{\max} \eta_t^2}{2} \ell_t(\tau_t \mid \mathcal{H}_t(\tau_t)), \quad (75)
$$

where (70) holds due to $\ln(x) \leq x - 1$ for $x \leq 1$, (71) uses $\exp(x) \leq 1 + x + x^2/2$ for $x \leq 0$, (72) holds as the same as (50), (73) holds as $\ell_t(\tau) = \ell_t(\tau')$ for all $\tau \in \mathcal{H}(\tau')$ by the definition of the specialized loss (4) and its domain $[0, \ell_{\max}]$, and (74) holds as the same as (52), *i.e.,* the property of the indicator function.

**Third step.** Next, we reformulate the second term, the log of an exponential expectation term. Let $\tilde{L}_t(\tau) := \sum_{t=1}^{T} \ell_t(\tau \mid \mathcal{H}_t(\tau_t))$ and $\tilde{L}_0(\tau) = 0$. Then, we have

$$-\frac{1}{\eta_t} \ln \mathbb{E}_{\tau \sim p_t} \exp\Big( -\eta_t \ell_t(\tau \mid \mathcal{H}_t(\tau_t)) \Big)$$

$$= -\frac{1}{\eta_t} \ln \sum_{\tau \in \mathcal{H}} p_t(\tau) \exp\Big( -\eta_t \ell_t(\tau \mid \mathcal{H}_t(\tau_t)) \Big)$$

$$= -\frac{1}{\eta_t} \ln \sum_{\tau \in \mathcal{H}} \frac{\exp(-\eta_t \tilde{L}_{t-1}(\tau))}{\sum_{\bar{\tau} \in \mathcal{H}} \exp(-\eta_t \tilde{L}_{t-1}(\bar{\tau}))} \exp\Big( -\eta_t \ell_t(\tau \mid \mathcal{H}_t(\tau_t)) \Big)$$

$$= -\frac{1}{\eta_t} \ln \frac{\sum_{\tau \in \mathcal{H}} \exp(-\eta_t \tilde{L}_t(\tau))}{\sum_{\tau \in \mathcal{H}} \exp(-\eta_t \tilde{L}_{t-1}(\tau))}. \tag{76}$$

**Fourth step.** Then, combining rewritten terms, (75) and (76), the following inequality holds:

$$\sum_{t=1}^{T} \mathbb{E}_{\tau \sim p_t} \ell_t(\tau \mid \mathcal{H}_t(\tau_t)) \le \sum_{t=1}^{T} \frac{\ell_{\max} \eta_t}{2} \ell_t(\tau_t \mid \mathcal{H}_t(\tau_t)) - \sum_{t=1}^{T} \frac{1}{\eta_t} \ln \frac{\sum_{\tau \in \mathcal{H}} \exp(-\eta_t \tilde{L}_t(\tau))}{\sum_{\tau \in \mathcal{H}} \exp(-\eta_t \tilde{L}_{t-1}(\tau))}$$

$$\le \sum_{t=1}^{T} \frac{\ell_{\max} \eta_t}{2} \ell_t(\tau_t \mid \mathcal{H}_t(\tau_t)) - \frac{1}{\eta_T} \sum_{t=1}^{T} \ln \frac{\sum_{\tau \in \mathcal{H}} \exp(-\eta_t \tilde{L}_t(\tau))}{\sum_{\tau \in \mathcal{H}} \exp(-\eta_t \tilde{L}_{t-1}(\tau))} \tag{77}$$

$$= \sum_{t=1}^{T} \frac{\ell_{\max} \eta_t}{2} \ell_t(\tau_t \mid \mathcal{H}_t(\tau_t)) - \frac{1}{\eta_T} \ln \frac{\sum_{\tau \in \mathcal{H}} \exp(-\eta_T \tilde{L}_T(\tau))}{\sum_{\tau \in \mathcal{H}} \exp(-\eta_1 \tilde{L}_0(\tau))}$$

$$= \sum_{t=1}^{T} \frac{\ell_{\max} \eta_t}{2} \ell_t(\tau_t \mid \mathcal{H}_t(\tau_t)) - \frac{1}{\eta_T} \ln \sum_{\tau \in \mathcal{H}} \exp(-\eta_T \tilde{L}_T(\tau)) + \frac{\ln |\mathcal{H}|}{\eta_T}$$

$$\le \sum_{t=1}^{T} \frac{\ell_{\max} \eta_t}{2} \ell_t(\tau_t \mid \mathcal{H}_t(\tau_t)) - \frac{1}{\eta_T} \ln \Big( \max_{\tau} \exp(-\eta_T \tilde{L}_T(\tau)) \Big) + \frac{\ln |\mathcal{H}|}{\eta_T}$$

$$= \sum_{t=1}^{T} \frac{\ell_{\max} \eta_t}{2} \ell_t(\tau_t \mid \mathcal{H}_t(\tau_t)) + \min_{\tau} \sum_{t=1}^{T} \ell_t(\tau \mid \mathcal{H}_t(\tau_t)) + \frac{\ln |\mathcal{H}|}{\eta_T},$$

where (77) holds as $\eta_t$ is non-increasing and $\ln \frac{\sum_{\tau \in \mathcal{H}} \exp(-\eta_t \tilde{L}_t(\tau))}{\sum_{\tau \in \mathcal{H}} \exp(-\eta_t \tilde{L}_{t-1}(\tau))} \le 0$ for all $t$.

From this and the fact (53) that $\mathbb{E}_{\tau \sim p_t} \ell_t(\tau \mid \mathcal{H}_t(\tau_t)) = \ell_t(\tau_t) - \gamma_t \ell_t(\tau_t \mid \mathcal{H}_t(\tau_t))$, we have

$$\sum_{t=1}^{T} \ell_t(\tau_t) - \min_{\tau} \sum_{t=1}^{T} \ell_t(\tau \mid \mathcal{H}_t(\tau_t)) \le \sum_{t=1}^{T} \frac{\ell_{\max} \eta_t}{2} \ell_t(\tau_t \mid \mathcal{H}_t(\tau_t)) + \frac{\ln |\mathcal{H}|}{\eta_T} + \sum_{t=1}^{T} \gamma_t \ell_t(\tau_t \mid \mathcal{H}_t(\tau_t))$$

$$= \sum_{t=1}^{T} \Big( \frac{\ell_{\max} \eta_t}{2} + \gamma_t \Big) \ell_t(\tau_t \mid \mathcal{H}_t(\tau_t)) + \frac{\ln |\mathcal{H}|}{\eta_T}.$$

Then, from Lemma 3 and Corollary 2 along with the union bound, the following inequality holds with probability at least $1 - 2\delta'$, where $\delta' = \delta/2$ if $\eta_t \le \frac{2\gamma_t}{\ell_{\max}}$:

$$\sum_{t=1}^{T} \ell_t(\tau_t) - \min_{\tau} \sum_{t=1}^{T} \ell_t(\tau) \le \sum_{t=1}^{T} \Big( \frac{\ell_{\max} \eta_t}{2} + \gamma_t \Big) \ell_t(\tau_t \mid \mathcal{H}_t(\tau_t)) + \frac{\ln |\mathcal{H}|}{\eta_T} + \frac{\ell_{\max} \ln(2|\mathcal{H}|/\delta)}{2\gamma}$$

$$\le 2\ell_{\max} \sum_{t=1}^{T} \Big( \frac{\ell_{\max} \eta_t}{2} + \gamma_t \Big) + \ell_{\max} \ln \frac{2}{\delta} + \frac{\ln |\mathcal{H}|}{\eta_T} + \frac{\ell_{\max} \ln(2|\mathcal{H}|/\delta)}{2\gamma},$$

where the first inequality holds with probability at least $1 - \delta'$ from Corollary 2 and the second inequality holds with probability at least $1 - \delta'$ from Lemma 3 when $\alpha_t = \frac{\ell_{\max} \eta_t}{2} + \gamma_t \le 2\gamma_t$.

**Final Bounds.** We have our regret bound depending on the setup of the learning rate $\eta_t$.

If $\eta_t = \frac{2\gamma_t}{\ell_{\max}} = \sqrt{\frac{\ln|\mathcal{H}|}{\ell_{\max}^2 T}}$, with probability at least $1 - \delta$,

$$\mathbf{Reg}_T \le 2\ell_{\max}\sqrt{T\ln|\mathcal{H}|} + \ell_{\max}\left(1 + \sqrt{\frac{T}{\ln|\mathcal{H}|}}\right)\ln\frac{2}{\delta} + 2\ell_{\max}\sqrt{T\ln|\mathcal{H}|}$$

$$= \ell_{\max}\left(4\sqrt{T\ln|\mathcal{H}|} + \left(1 + \sqrt{\frac{T}{\ln|\mathcal{H}|}}\right)\ln\frac{2}{\delta}\right)$$

$$= \mathcal{O}(\ell_{\max}\sqrt{T\ln(|\mathcal{H}|/\delta)}). \tag{78}$$

If $\eta_t = \frac{2\gamma_t}{\ell_{\max}} = \sqrt{\frac{\ln|\mathcal{H}|}{2\ell_{\max}^2 t}}$, with probability at least $1 - \delta$,

$$\mathbf{Reg}_T \le 2\ell_{\max}\sqrt{2T\ln|\mathcal{H}|} + \ell_{\max}\left(1 + \sqrt{\frac{2T}{\ln|\mathcal{H}|}}\right)\ln\frac{2}{\delta} + 2\ell_{\max}\sqrt{2T\ln|\mathcal{H}|}$$

$$= \ell_{\max}\left(4\sqrt{2T\ln|\mathcal{H}|} + \left(1 + \sqrt{\frac{2T}{\ln|\mathcal{H}|}}\right)\ln\frac{2}{\delta}\right)$$

$$= \mathcal{O}(\ell_{\max}\sqrt{T\ln(|\mathcal{H}|/\delta)}),$$

noting that $\sum_{t=1}^T 1/\sqrt{t} \le 2\sqrt{T}$.

**Expected regret** The expected regret can be obtained via integrating the deviations in (78), *i.e.*,

$$\mathbb{E}[W] \le \int_0^2 \frac{1}{2\delta}\mathbb{P}\left(W > \ln\frac{2}{\delta}\right)d\delta.$$

Here, if we take $W = \frac{1}{\ell_{\max}\left(1 + \sqrt{T/\ln|\mathcal{H}|}\right)}\left(\mathbf{Reg}_T - 4\ell_{\max}\sqrt{T\ln|\mathcal{H}|}\right) > \ln\frac{2}{\delta}$, which holds with probability at most $\delta$, then,

$$\mathbb{E}[W] \le 1,$$

which suggests that

$$\mathbb{E}[\mathbf{Reg}_T] \le \ell_{\max}\left(4\sqrt{T\ln|\mathcal{H}|} + 1 + \sqrt{\frac{T}{\ln|\mathcal{H}|}}\right) = \mathcal{O}(\ell_{\max}\sqrt{T\ln|\mathcal{H}|}). \tag{79}$$

# I  DISCUSSION ON SELECTION EFFICIENCY

In this section, we demonstrate that our algorithm does not converge to a trivial selective generator in the aspect of full-feedback setting, *i.e.*, $\arg\min_\tau \sum_{t=1}^T [a_t(\tau) + \lambda d_t(\tau, \alpha)] \notin \mathcal{H}^{\text{triv}}$, if there exists some $\tau \in \mathcal{H} \setminus \mathcal{H}^{\text{triv}}$ that satisfies the FDR guarantee, where $\mathcal{H}^{\text{triv}} := \{\tau \mid \sum_{t=1}^T a_t(\tau) = T\}$.

Let $e_t^\tau := \mathcal{E}_t(\hat{S}(\cdot; \tau))$, and $\tau^{\text{triv}} \in \mathcal{H}^{\text{triv}}$, which is always abstaining, then for any $\tau^{\text{non-triv}} \in \mathcal{H} \setminus \mathcal{H}^{\text{triv}}$ that satisfies the FDR guarantee, the following inequality holds,

$$\frac{\sum_{t=1}^T \mathbb{1}\left(\hat{S}(\mathbf{x}_t; \tau^{\text{non-triv}}) \ne \texttt{IDK} \wedge e_t^{\tau^{\text{non-triv}}}\right)}{\sum_{t=1}^T \mathbb{1}\left(\hat{S}(\mathbf{x}_t; \tau^{\text{non-triv}}) \ne \texttt{IDK}\right)} \le \alpha$$

$$\Rightarrow \sum_{t=1}^T \mathbb{1}\left(\hat{S}(\mathbf{x}_t; \tau^{\text{non-triv}}) \ne \texttt{IDK} \wedge e_t^{\tau^{\text{non-triv}}}\right) \le \alpha \sum_{t=1}^T \mathbb{1}\left(\hat{S}(\mathbf{x}_t; \tau^{\text{non-triv}}) \ne \texttt{IDK}\right). \tag{80}$$

Let $L_T(\tau, \alpha) := \sum_{t=1}^T \frac{a_t(\tau) + \lambda d_t(\tau, \alpha)}{1+\lambda}$, which is the cumulative loss of $\tau$, and consider the difference of cumulative loss between $\tau^{\text{triv}}$ and $\tau^{\text{non-triv}}$,

$$(1+\lambda)(L_T(\tau^{\text{triv}}) - L_T(\tau^{\text{non-triv}}))$$

$$= \sum_{t=1}^T [a_t(\tau^{\text{triv}}) + \lambda d_t(\tau^{\text{triv}}, \alpha)] - \sum_{t=1}^T [a_t(\tau^{\text{non-triv}}) + \lambda d_t(\tau^{\text{non-triv}}, \alpha)]$$

$$= \sum_{t=1}^T \left[ \mathbb{1}\left( \hat{S}(\mathbf{x}_t; \tau^{\text{triv}}) = \text{IDK} \right) \right.$$

$$\left. + \lambda \left( \mathbb{1}(\hat{S}(\mathbf{x}_t; \tau^{\text{triv}}) \neq \text{IDK} \wedge e_t^{\tau^{\text{triv}}}) - \alpha \mathbb{1}(\hat{S}(\mathbf{x}_t; \tau^{\text{triv}}) \neq \text{IDK}) + \alpha \right) \right]$$

$$- \sum_{t=1}^T \left[ \mathbb{1}\left( \hat{S}(\mathbf{x}_t; \tau^{\text{non-triv}}) = \text{IDK} \right) \right.$$

$$\left. + \lambda \left( \mathbb{1}(\hat{S}(\mathbf{x}_t; \tau^{\text{non-triv}}) \neq \text{IDK} \wedge e_t^{\tau^{\text{non-triv}}}) - \alpha \mathbb{1}(\hat{S}(\mathbf{x}_t; \tau^{\text{non-triv}}) \neq \text{IDK}) + \alpha \right) \right]$$

$$= T + \lambda \alpha T - \left[ T - \sum_{t=1}^T \mathbb{1}\left( \hat{S}(\mathbf{x}_t; \tau^{\text{non-triv}}) \neq \text{IDK} \right) \right.$$

$$+ \lambda \left\{ \alpha \left( T - \sum_{t=1}^T \mathbb{1}\left( \hat{S}(\mathbf{x}_t; \tau^{\text{non-triv}}) \neq \text{IDK} \right) \right) \right.$$

$$\left. \left. + \sum_{t=1}^T \mathbb{1}\left( \hat{S}(\mathbf{x}_t; \tau^{\text{non-triv}}) \neq \text{IDK} \wedge e_t^{\tau^{\text{non-triv}}} \right) \right\} \right]$$

$$= \sum_{t=1}^T \mathbb{1}\left( \hat{S}(\mathbf{x}_t; \tau^{\text{non-triv}}) \neq \text{IDK} \right) + \lambda \alpha \sum_{t=1}^T \mathbb{1}\left( \hat{S}(\mathbf{x}_t; \tau^{\text{non-triv}}) \neq \text{IDK} \right)$$

$$- \lambda \sum_{t=1}^T \mathbb{1}\left( \hat{S}(\mathbf{x}_t; \tau^{\text{non-triv}}) \neq \text{IDK} \wedge e_t^{\tau^{\text{non-triv}}} \right).$$

Since $\tau^{\text{non-triv}}$ satisfies the FDR guarantee (80), we get

$$L_T(\tau^{\text{triv}}) - L_T(\tau^{\text{non-triv}}) \geq \sum_{t=1}^T \frac{\mathbb{1}\left( \hat{S}(\mathbf{x}_t; \tau^{\text{non-triv}}) \neq \text{IDK} \right)}{1+\lambda} = \frac{T - \sum_{t=1}^T a_t(\tau^{\text{non-triv}})}{1+\lambda} > 0, \quad (81)$$

where $T - \sum_{t=1}^T a_t(\tau^{\text{non-triv}}) > 0$ holds due to $\tau^{\text{non-triv}} \notin \mathcal{H}^{\text{triv}}$. This implies that, given sufficiently many data points, any $\tau^{\text{non-triv}}$ that satisfies the FDR guarantee incurs at least $T - \sum_{t=1}^T a_t(\tau^{\text{non-triv}})$ less loss than $\tau^{\text{triv}}$. This is empirically supported by Figure 22, 23, and 28.

