# OpenReview forum: "Online Selective Generation with Adversarial Bandit Feedback"
_ICLR.cc/2026/Conference — Submitted to ICLR 2026_

### Official Review · Reviewer_SKdk · 2025-10-28

**Soundness:** 3
**Presentation:** 3
**Contribution:** 2
**Rating:** 4
**Confidence:** 4

**Summary:**

This paper introduces a novel online learning framework called ExSUL for selective language generation. The goal is to enable models to judiciously decide when to generate answers and when to abstain, especially in environments with noisy, incomplete, or adversarial feedback. The method leverages partial feedback—such as binary signals indicating correctness—to learn effective abstention policies without requiring full supervision. By incorporating techniques from adversarial bandit algorithms and providing regret guarantees, ExSUL balances the trade-off between reducing hallucinations and maintaining answer coverage. The experimental results demonstrate that the approach outperforms existing methods in controlling hallucinations while sustaining high answer quality across multiple datasets and settings.

**Strengths:**

1. The paper presents a novel framework that effectively utilizes partial, binary feedback signals to learn abstention policies in language generation tasks.
2. By adapting adversarial bandit algorithms, the proposed ExSUL method provides formal regret bounds, ensuring that the model's accumulated reward converges close to that of the best fixed policy in hindsight.
3. Experiments demonstrate that ExSUL achieves superior control over hallucination rates, maintaining high answer accuracy and low FDR across various datasets and under distribution shifts.

**Weaknesses:**

1. This paper investigates online selective generation, yet lacks corresponding online experiments or genuine user feedback, which to some extent undermines the reliability of the results. For instance, this paper employs GPT as a generative model and similarly utilises the GPT series to simulate user feedback. Homogeneous models may exhibit similar hallucinations and also harbour homogeneous biases, which diverge significantly from real-world user feedback. While the paper provides promising results in simulated and benchmark environments, it offers limited discussion on how the proposed abstention and learning strategies scale to large-scale, noise real-world applications.
2. The method relies on the presence of binary or weak feedback signals, which may vary in quality and consistency across different deployment scenarios. The paper does not thoroughly analyze how noisy or sparse feedback could impact the learning process, leaving some uncertainty about robustness to imperfect supervision.
3. The evaluations focus on their defined metrics (e.g., FDR, regret) without incorporating human judgment or user-centric metrics. Including such evaluation would better demonstrate the practical benefits and acceptance of the abstention policy in real-world decision-making contexts. Moreover, this paper aims to enhance the output quality of LLMs from the perspective of selective generation. However, it appears to place undue emphasis on FDR. An LLM that is overly reluctant to provide answers may exhibit a low FDR, yet it would hardly be considered a good model.

**Questions:**

1. To what extent does GPT-generated simulated user feedback deviate from genuine noisy and random user feedback?
2. Does ExSUL maintain its performance advantage under conditions of noisy and sparse user feedback?
3. In real-world model deployment, how should one balance false positive rate (FDR) with over-rejection?

---

> ### Author Response · Authors · 2025-11-21
>
> We thank the reviewer for insightful feedback. The following includes our clarifications and new experimental results. Feel free to let us know if there are remaining concerns.
>
> ### Q: To what extent does GPT-generated simulated user feedback deviate from genuine noisy and random user feedback?
> > 1. This paper investigates online selective generation, …
>
> We thank the reviewer for this valid point. We clarify that while we use a GPT model as the "user" (i.e., "adversary" in our formulation), its task is fundamentally different and simpler than the generative model's.
>
> The feedback mechanism in our experiments involves **comparing a generated sequence against a ground truth answer** (full context in the interactive environment). This is analogous to a simple Natural Language Inference (NLI) or textual-entailment task, not open-ended generation. This simpler, discriminative task is significantly less susceptible to the same modes of hallucination or bias that affect the generative model.
>
> Therefore, **we consider this setup as a valid proxy for strong supervision** to validate our algorithm's core convergence, rather than a simulation of noisy real-world user feedback (which we address in the following).
>
> ### Q: Does ExSUL maintain its performance advantage under conditions of noisy and sparse user feedback?
> > 2. The method relies on the presence of binary or weak feedback signals, which may vary in quality and consistency across different deployment scenarios. …
>
> This is an excellent point. We thank the reviewer for raising it. We acknowledge that our main experiments assumed a strong supervision signal, whereas real-world feedback can be noisy or sparse.
>
> To evaluate our method's robustness under imperfect supervision, we conducted **a new experiment using a proxy for weak feedback**. Specifically, we used “5% randomly flipped labels” and “ROUGE-L scores (thresholded)”  to simulate a 'noisy' feedback signal instead of the strong supervision signal.
>
> We added our new results in Figure 30-31 in the Appendix, which correspond to the stochastic environments in Figure 3 and 8-10. We clarify that *While the algorithm guarantees FDR control with respect to these noisy signals, the reported FDR is evaluated against the original strong signals (GPT 3.5 feedback) to assess empirical hallucination control.*
>
> In Figure 30 and 31, while ExSUL exhibits slightly increased FDR compared to the original settings (Figures 3, 8-10) as expected, we observe that **it successfully controls the FDR near $\alpha$ across most settings, maintaining its performance advantage over the baselines.**
>
> ### Q: In real-world model deployment, how should one balance false positive rate (FDR) with over-rejection?
>
> We thank the reviewer for this practical question. Our framework is designed precisely to manage the balance between the False Discovery Rate (FDR) and efficiency (over-rejection) using two distinct control points, focused on user access:
> - The primary mechanism for managing this trade-off is the user-defined desired risk level, $\alpha$. Users directly set $\alpha$ to establish the fundamental boundary between FDR and efficiency.
>   - A core advantage of our method is its operational flexibility: **if a user experiences unsatisfactory selection efficiency (i.e., over-rejection), they can simply and instantaneously adjust $\alpha$ in real-time.** This is rooted in our assumption of an adaptive adversary, which theoretically accommodates continuous distribution shifts. This allows the system to be re-optimized instantly with the new $\alpha$ value without significant penalty.
>  This capability allows the user to balance over-rejection issues while maintaining control over the FDR boundary.
>
> In addition, during the initial deployment stage, the deployer sets the hyperparameter $\lambda$. As noted in Appendix H.3, **$\lambda$ enables the deployer to set a fundamental trade-off between FDR convergence speed and selection inefficiency (over-rejection)**, for which we recommended the value $T^{1/4}$ in the remark.

---

> ### Author Response · Authors · 2025-11-21
>
> ### Q: Is there any evaluation over the actual user-centric metrics?
> > 3. The evaluations focus on their defined metrics (e.g., FDR, regret) without incorporating human judgment or user-centric metrics. …
>
> We thank the reviewer for this constructive suggestion regarding practical utility.
>
> We first want to highlight that our framework already optimizes **a critical user-centric metric: Selection Efficiency (the ratio of non-abstaining answers)**. Our objective is not just to minimize errors (FDR) but to maximize the number of answered queries (maximize Efficiency) within that safety constraint. A model that abstains too often is indeed not user-friendly, which is why we explicitly treat Efficiency (or Inefficiency) as a core objective in our loss function (4) and analysis.
>
> To further address the reviewer's concern about the quality of the responses provided to users, **we conducted new experiments to measure "Helpfulness"**, a widely used alignment metric that determines whether a response is concise, non-toxic, and efficiently well-aligned with the user's intent. We utilized the “gpt2-large-helpful-reward model”, which is fine-tuned on the HH-RLHF dataset, to score the generated answers.
>
> The table below reports the average helpfulness scores evaluated on non-abstaining responses.
>
> | Dataset-Model \ Method | No-SG | Exp3-IX-SG | ExSUL (Ours) | EW-SG (Full-feedback) |
> | ---: | :---: | :---: | :---: | :---: |
> | **NQ-LLaMA3.1-8B** | **1.2670** | 1.3661 | **1.5681** | 1.5915 |
> | **TriviaQA-LLaMA3.1-8B** | **1.4957** | 1.5907 | **1.6703** | 1.6786 |
> | **NQ-GPT-3.5-turbo** | **1.0814** | 1.1135 | **1.1420** | 1.1571 |
> | **TriviaQA-GPT-3.5-turbo** | **1.1916** | 1.2033 | **1.2265** | 1.2328 |
>
> As shown in the table, ExSUL consistently achieves a higher helpfulness score compared to the No-SG baseline. We interpret this result as evidence of effective selective generation: ExSUL identifies and abstains from uncertain or incorrect responses—which would typically receive low helpfulness scores. By filtering out these low-quality tails, the model does choose to answer, the response is not only correct but also more helpful to the user.
>
> We sincerely thank the reviewer again for their constructive and insightful feedback. We hope that our clarifications and the new experiments (on noisy feedback and user-centric helpfulness metrics) have successfully addressed the concerns raised. We are happy to elaborate further on any additional points.

---

### Official Review · Reviewer_nsoa · 2025-10-31

**Soundness:** 3
**Presentation:** 3
**Contribution:** 3
**Rating:** 4
**Confidence:** 1

**Summary:**

This paper introduces a novel framework for online selective generation—a setting where a large language model (LLM) can abstain from answering uncertain queries to reduce hallucination. The authors focus on a realistic adversarial and partial feedback scenario, where feedback is limited (e.g., thumbs-up/down on generated answers) instead of full supervision.
To tackle this, the paper reformulates selective generation as an adversarial multi-armed bandit problem and proposes an online learning algorithm that controls the False Discovery Rate (FDR)—the proportion of hallucinated outputs among generated ones—while maintaining high selection efficiency (i.e., answering rate). The key technical innovations are:
A Regret-to-FDR conversion lemma that connects the regret bounds of any bandit algorithm to FDR control guarantees.
A feedback unlocking mechanism that exploits structural properties of selective generation to reuse partial feedback efficiently.
Extensive experiments under diverse environments demonstrate that the proposed algorithm achieves effective FDR control and maintains competitive efficiency compared to baselines, showing robustness in adversarial and distribution-shifted conditions

**Strengths:**

- The paper addresses a realistic and underexplored setting - online generation with partial and adversarial feedback and provide theoretical link between bandit regret and false discovery rate, offering a principled approach to uncertainty control in generative models.
- This seems aligns well with human-in-the-loop and reinforcement learning from human feedback (RLHF) settings.

**Weaknesses:**

- Some algorithmic components (especially feedback unlocking) may require additional clarification for reproducibility.

**Questions:**

1. Can the proposed method be integrated with large-scale language model training pipelines?
2. How is the feedback unlocking mechanism implemented in practice—does it assume stationarity in the abstention threshold?
I am not very familiar with the learning theory field, so I leave a low confidence for my review.

---

> ### Author Response · Authors · 2025-11-17
>
> We thank the reviewer for constructive questions. We are happy to provide these clarifications.
>
>
> ### Q: Can the proposed method be integrated with large-scale language model training pipelines?
> Yes. A key strength and primary design goal of our method is its seamless integration. Our framework (ExSUL) is fully model-agnostic, makes minimal and practical assumptions on data distributions (i.e., Lines 149-155), and treats the large-scale language model (Generator $G$) as a complete black box.
>
> This means ExSUL requires no access to or modification of the LLM’s internal workings. **It is deployed as a lightweight, online wrapper on top of any model**. For example, it can be particularly well-integrated into RLHF and Reinforcement learning with verifiable rewards (RLVR), where we can fine-tune a pretrained model with rollouts via RL and learn our selective generator simultaneously.
>
>
> ### Q: How is the feedback unlocking mechanism implemented in practice—does it assume stationarity in the abstention threshold?
> First of all, we will release our code that provides implementation details on the feedback unlocking mechanism.
>
> Briefly speaking, the unlocking works as follows: For example, if the selective generator chooses to answer (i.e., $\tau_t \le f_t$), we know that all other arms (thresholds) $\tau$ that are also less than or equal to $f_t$ would have also chosen to answer. Therefore, they all share the same feedback (loss) as the selected arm ($\tau_t$). Similarly, if the generator abstains (i.e., $\tau_t > f_t$), we can infer the loss (feedback) for all other arms $\tau > f_t$.
> Figure 29 provides some insights of these two "unlocked" cases (which have the same losses) at each step $t$.
>
> For the stationarity concern, the "Feedback Unlocking" mechanism **does not rely on stationarity**. Instead, it exploits the **time-varying monotonic structure of the selection function** (which is parameterized by a time-varying threshold $\tau_t$ and also a possibly time-varying scoring function $f_t$) at a single point in time $t$. At each step $t$, the selective generator receives a confidence score $f_t(\mathbf{x}_t, G(\mathbf{x}_t))$​ for a generated answer $G(\mathbf{x}_t)$ and then update the threshold $\tau_t$.
>
>
> We thank the reviewer again for their time and thoughtful questions. We hope these clarifications have fully addressed your concerns and helped increase confidence in our work. We are happy to elaborate further.

---

### Official Review · Reviewer_jGcW · 2025-11-01

**Soundness:** 3
**Presentation:** 3
**Contribution:** 3
**Rating:** 8
**Confidence:** 2

**Summary:**

This work proposes an online selective-generation method that leverages “feedback unlocking” to learn a threshold from partial feedback, controlling FDR with competitive selection efficiency.

**Strengths:**

- The paper is clearly written and easy to follow.
- The paper relaxes i.i.d. to non-stochastic inputs with partial feedback. This matches real-world use.
- Results across multiple distribution-shift settings (single, alternating, gradual) substantiate the claims.

**Weaknesses:**

- The theory permits an adaptive adversary, but the experiments cover only distribution shifts and a templated interactive setup.
- The main text says the interactive environment uses two GPT-3.5-turbo models, whereas the appendix specifies GPT-3.5-turbo + two GPT-4o.

**Questions:**

- The theory assumes an adaptive adversary, but the experiments don’t seem to include one. How does the method perform against a strategy-aware adaptive adversary?
- How was the threshold grid size $|H|$ selected?

---

> ### Author Response · Authors · 2025-11-21
>
> Thanks for your support on our paper.
>
> ### Q: The theory assumes an adaptive adversary, but the experiments don’t seem to include one. How does the method perform against a strategy-aware adaptive adversary?
> > The theory permits an adaptive adversary, but the experiments cover only distribution shifts and a templated interactive setup.
>
> We thank the reviewer for this insightful question. We agree that testing against a strategy-aware adversary strengthens the validation of our theoretical claims
> - **Revisiting the Existing Setup**: First, we emphasize that our original interactive setup (Figure 7) is indeed **inherently adaptive**, as the generated questions depend dynamically on the interaction history. While not explicitly *adversarial* (i.e., optimized solely to break the learner), it effectively simulates a dynamic, non-stationary environment.
> - **Theoretical Context**: Furthermore, we clarify that the adaptive-adversary assumption serves as a powerful theoretical framework to ensure the robustness against **any non-stochastic data sequence**. Consequently, Lemma 1 guarantees that the FDR bound holds even in worst-case adversarial scenarios. While implementing such a theoretically optimal adversary is practically intractable, this theoretical guarantee is empirically supported by our distribution-shift experiments (Figures 4-7).
> - **New Experiment with Strategy-Aware Adversary**: To directly address the reviewer's concern, we have conducted a new realistic simulation designed to adaptively exploit the learner.
>   - *Setup*: We utilized a "user-acting agent" (GPT-4o) that observes the learner’s decision history and strategically selects questions from the combined NQ and TriviaQA datasets. The agent is explicitly prompted to maximize the learner's failures (i.e., maximizing accepted incorrect answers and rejected correct answers). Please refer to Appendix G.8 for full experimental details and prompts.
>   - *Results*: As shown in the new Figure 32 (simulated over $T=5,000$), ExSUL successfully controls the FDR even under this active adversarial pressure. **This observation empirically supports our theoretical robustness (Lemma 1), demonstrating that ExSUL continuously adapts to control the FDR near the target risk $\alpha$ despite the adversary's strategic attempts to induce failures.**
>
> Notably, we observe that the FDR exhibits dynamic fluctuations around the target $\alpha$, rather than strictly converging below it as in the standard interactive setup. This behavior is expected given the relatively short $T$, as the adversary actively induces high inefficiency, which theoretically slows down convergence (as analyzed in Section 3.3) . It demonstrates that **while the adversary creates a challenging non-stationary environment, ExSUL continuously adapts over time against the adaptive adversary**.
>
> In the final version, we will include results from a setup identical to the original interactive environment (i.e., a fully agentic framework with a longer time horizon $T=12,500$) to ensure a comprehensive comparison.
> We sincerely thank the reviewer for this constructive feedback, as it allowed us to provide a more rigorous validation of our method’s robustness.

---

> ### Author Response · Authors · 2025-11-21
>
> ### Q: How was the threshold grid size $|\mathcal{H}|$ selected?
>
> The selection of $|\mathcal{H}|$ involves a key theoretical trade-off, especially regarding the baseline we compare against.
> - **Impact on Bounds**: The size of the hypothesis space, $|\mathcal{H}|$, appears in the regret bounds for all methods (Theorems 1, 2, and 3). Since our FDR bound (Lemma 1) depends on the regret term, a smaller $|\mathcal{H}|$ is theoretically preferable for faster convergence at a large scale of $T$.
> - **Key Theoretical Difference between methods**: However, the dependency on $|\mathcal{H}|$ is not uniform.
>   - Our method, ExSUL (Theorem 1), and the full-feedback oracle, EW (Theorem 3), both have a $\sqrt{\ln|\mathcal{H}|}$ factor, while the baseline Exp3-IX (Theorem 2) has a much severe $\sqrt{|\mathcal{H}| \ln|\mathcal{H}|}$ factor.
>   - This is because it only observes the loss for the single chosen arm and must waste significant effort exploring the entire hypothesis space.
> - **The Choice of 1000**: Therefore, we selected $|\mathcal{H}|=1000$ as a practical comparison point. It is large enough to reasonably discretize the scoring function's space, but not so large that it makes the task for the Exp3-IX-SG baseline computationally impossible.
>   - For example, Figure 24 shows that the Exp3-IX baseline barely manages to converge to the desired FDR with $|\mathcal{H}|=1000, T=1500K$. If we had set $|\mathcal{H}|$ larger (e.g., 10,000), Exp3-IX completely fails to converge even for significantly larger $T$, making for a less informative comparison.
>
> We thank the reviewer again for this question, as it clarifies an important methodological choice.
> We will add this justification to the appendix.
>
>
> ### About Typo
> > The main text says the interactive environment uses two GPT-3.5-turbo models, whereas the appendix specifies GPT-3.5-turbo + two GPT-4o.
>
> Thank you for catching this. There was a typo in the main body, we have fixed “two GPT-3.5-turbo models” into “GPT-3.5-turbo and GPT-4o models”, as mentioned in the general response.
>
>
> Feel free to let us know if you have additional clarification and thanks again for your positive evaluation!

---

> > ### Comment · Reviewer_jGcW · 2025-11-26
> >
> > Thank you for the rebuttal. I will keep my positive score. However, since some parts are outside my core area, I feel it is appropriate to keep my low confidence rating.

---

> > > ### Author Response · Authors · 2025-11-26
> > >
> > > Thank you for your response. We appreciate your time reviewing our paper and are grateful for your decision to maintain the positive score.

---

### Official Review · Reviewer_ut5G · 2025-11-02

**Soundness:** 2
**Presentation:** 3
**Contribution:** 2
**Rating:** 2
**Confidence:** 3

**Summary:**

This work studies selective generation in Large Language Models, a mechanism that allows the model to selectively abstain from answering and prevent hallucination. The authors reduce this problem to a standard adversarial bandit problem and provide a theoretical analysis for the relationship between the bandit regret and the false discovery rate. Based on this reduction, the authors proposed a new algorithm, and experiments support the efficiency of the proposed method.

**Strengths:**

1. The paper focuses on preventing hallucination, directly contributing to a major area of concern for LLM safety and usability.

2. The proposed method's efficacy is experimentally validated by its success in lowering the false discovery rate.

**Weaknesses:**

1. The motivation for Selective Generation is not sufficiently strong, as the strategy's high cost in utility may outweigh its benefit in reducing hallucination. While selectively abstaining from answers helps reduce the False Discovery Rate, Figures 3-6 show an alarming inefficiency rate of 40% to 50%, meaning the LLM refuses to answer nearly half of the questions, which severely limits its practical impact. This approach risks reducing the problem to a trivial solution (e.g., only answering simple, high-certainty questions) that offers minimal real-world help. Instead of outright abstention, a more reasonable approach would be to output the low-confidence answer along with the low-confidence sign. This allows the user to still potentially gain some minor insight or utility while remaining informed and able to control for the risk of hallucination.

2. The reduction of the selective generation problem to an adversarial bandit framework is theoretically questionable due to a fundamental conflict in objectives. In standard adversarial bandits, regret is a comparative metric focused on minimizing the sub-optimality gap between the learner's performance and the performance of the unknown optimal policy ($\tau$). Conversely, the FDR risk in selective generation is a stand-alone metric that only measures the learner's performance without requiring a comparison to an optimal policy.

The authors address this gap by only comparing their algorithm against the trivial policy that always abstains from answering, as seen in the proof of Lemma 1 (Lines 1953-1957). This trivial policy yields a constant loss that simplifies the analysis. However, since the comparison policy is already known and fixed, the core difficulty of the bandit problem—dealing with the unknown optimal action—is bypassed. This decision to only ensure the learner does not perform significantly worse than the simplest, least effective baseline weakens the theoretical utility and necessity of employing complex bandit regret analysis.

3. The derived upper bound for the False Discovery Rate, presented in lines 253-257, is concerning. Even when neglecting the term related to inefficiency, the bound is $\alpha + 1/T^{1/4}$. This is worse than the optimal $\alpha$ achieved by the trivial policy that always abstains from answering. This finding supports the prior critique regarding the theoretical reduction: Theorem 1 essentially only guarantees that the FDR of the proposed method will not be significantly worse than that of the least-effective, baseline "always abstain" policy.

4. A significant concern regarding the experiments is the unexpected increasing trend observed in both the False Discovery Rate for Exp3-IX-SG and No-SG, and the inefficiency for ExSUL and Exp3-IX-SG after several thousand rounds (as shown in Figures 4 to 6). Intuitively, for a bandit algorithm, the cumulative regret increases, but the average regret per round should decrease; consequently, we would expect the FDR to decrease as the learner becomes more accurate. Similarly, a more accurate learner should result in decreasing inefficiency (fewer unnecessary abstentions). The observed increase directly contradicts the expected behavior for effective bandit learning.

**Questions:**

See weakness.

---

> ### Author Response · Authors · 2025-11-17
>
> We thank the reviewer for insightful feedback. The following includes our answers and feel free to let us know if there are remaining concerns.
>
> ### Q: How can selective generation be motivated if its high cost in utility outweighs the benefit of reducing hallucination?
> > 1\. The motivation for Selective Generation is not sufficiently strong, as the strategy's high cost in utility may outweigh its benefit in reducing hallucination. …
>
> We thank the reviewer for raising this important point about practical motivation. We agree that the trade-off between utility and safety is critical.
>
> **The following describes the motivations for selective generation despite its inherent utility cost.**
> - **Integration into Automated Frameworks (=no-human-in-the-loop)**: Selective generation is crucial for reliable automated systems (e.g., agentic AI). It provides an automated, threshold-based decision based on the desired risk level. In contrast, human-facing warnings cannot provide a clear, programmatic signal to the framework and introduce ambiguity in how to set the threshold for such warnings.
> - **High-Stakes Domains**: In critical domains (e.g., medical, finance, security), a hallucination—even one tagged with a warning—can cause irreversible, catastrophic harm. A simple warning is insufficient to guarantee safety.
> - **Harmfulness & Alignment**: Although this is not the current focus of our paper, when dealing with selective generation for harmfulness (not just hallucination), abstention is a non-negotiable safeguard.
>
> In addition, the high inefficiency (40-50%) inversely proves that this level of abstention is necessary for the base model and the scoring function, which highlights that the improvement of those components is needed.
> However, the task of **maximizing the achievable utility is orthogonal** to this paper.
> In other words, selective generation mainly controls the rate of hallucination with theoretical guarantees, while generator training (e.g., pretraining a model or fine-tuning a pretrained model) or calibrating a scoring function focuses on maximizing this achievable utility.
>
> In this sense, selective generation **complements** training as follows:
> - (when we *cannot* change a generator) a selective generator provides a desired level of hallucination by rejection with a theoretical guarantee given a fixed generator (e.g., when we cannot train the generator like commercial models) and
> - (when we *can* change a generator) the utility of a selective generator naturally increases as the underlying generator gets performant.
>
> ### Q: Why not output the low-confidence answer along with the low-confidence sign, instead of abstention?
> > Instead of outright abstention, a more reasonable approach would be to output the low-confidence answer along with the low-confidence sign. …
>
> As mentioned, our selective generation is mainly intended for the *no-human*-in-the-loop setup. In this sense, rejecting answers is more appropriate than providing confidence levels. However, this can be used with human-in-the-loop setup as well, where a deployer determines $\alpha$ and simply provides the underlying confidence level for interpretability.
>
> We will add an expanded discussion on the motivation for selective generation in the revised pdf.

---

> ### Author Response · Authors · 2025-11-17
>
> ### Q: How is the bandit framework's 'comparative' regret metric theoretically justified for optimizing the 'stand-alone' FDR risk? Specifically, the proof of Lemma 1 relies on a comparison against the trivial “always abstain” policy; does this not bypass the core difficulty of the bandit problem?
> > 2\. The reduction of the selective generation problem to an adversarial bandit framework is theoretically questionable…
>
> > The authors address this gap by only comparing their algorithm against the trivial policy that always abstains from answering, as seen in the proof of Lemma 1…
>
> **Our work does not bypass the core bandit problem, nor do we target the trivial (always abstain) baseline.**
> First, we solve the full, complex bandit problem by minimizing regret against an unknown, non-trivial expert. We then additionally prove (via Lemma 1) that this regret minimization provides a worst-case FDR guarantee.
>
> The theoretical utility of our framework is justified as follows:
> - **The Role of Lemma 1**: This lemma's purpose is not to set the trivial policy as the benchmark for our algorithm. Instead, Lemma 1 is a Regret-to-FDR *conversion* lemma. It proves that **with our specialized loss**, any algorithm that successfully minimizes regret is also guaranteed to control the FDR risk.
>   - The term in Lines 1953-1957 (Lines 2277-2281 in the revised version) is equivalent to the minimum FDR risk by the offline expert (i.e., $\min_{\tau} \mathcal{R}_{T}^{FDR}(\tau) / T$), which is always $\le$ 0 since the "trivial expert" achieves a FDR risk of 0, and other "good" experts that satisfy the FDR guarantee have a risk $\le 0$. Our proof simply upperbounds this non-positive expert term to isolate a clean, worst-case (where there are no good experts) FDR guarantee for the learner, which does not compare against the trivial expert.
> - **The Best Expert is Not Trivial**: Then, the reviewer's critique might imply that our "unknown best expert" $\tau^\*$ by our algorithm might be trivial anyway. We explicitly deal with this problem in the **Appendix I (Discussion on Selection Efficiency).** If any non-trivial expert that satisfies the FDR guarantee exists, it has a strictly lower cumulative loss than the "always abstaining" expert. **Therefore, $\tau^\*$ is non-trivial, and our algorithm converges toward this useful, non-trivial solution by a learner.**
>
> ### Q: How can the theoretical reduction be justified if its bound $\alpha + 1/T^{1/4}$ is worse than $\alpha$ achieved by the trivial “always abstain” policy?
> > 3\. The derived upper bound for the False Discovery Rate, presented in lines 253-257, is concerning. …
>
> We believe our previous response—our algorithm **does not target a trivial policy**—sufficiently addresses this critique.
> To add one more clarification, this upper bound is **analogous to standard regret analysis**; Bandit algorithms do not guarantee a regret $\le 0$, rather they provide a worst-case sublinear guarantee in $T$, ensuring that the average regret converges to 0. In the same vein, our FDR bound provides the sublinear guarantee that the average FDR risk converges to 0 (or the empirical FDR converges to $\alpha$ equivalently) in $T$.
>
> However, if any part of our explanation remains unclear, we would be happy to elaborate further.
>
>
> ### Q: Why do the experimental results (Figs 4-6) show increasing FDR and inefficiency over time, directly contradicting the theoretical expectation from effective bandit learning?
> > 4\. A significant concern regarding the experiments is the unexpected increasing trend…
>
> We thank the reviewer for this sharp observation.
> However, we clarify that **this trend is not a failure nor a contradiction of bandit learning**, but rather direct evidence of the distribution shift that Figures 4-6 are explicitly designed to test.
> - **Varying Data Difficulty**: The two datasets differ vastly in difficulty. For example, the baseline error rate for GPT-3.5-Turbo on the NQ is approximately 0.29, whereas on TriviaQA, it is only about 0.12.
> - **Adaptation, Not Failure**: When shifting from an easier to a harder environment (e.g., Figure 4, TriviaQA $\rightarrow$ NQ), the FDR naturally rises. In response, a successful learner must adapt by becoming more conservative (increasing inefficiency) to drive the FDR back toward the target $\alpha$.
>   - This adaptation in the long-term is visible in our existing experiments for large $T$. Figures 25 (c) and (d) show that the FDR eventually converges in shift environments.
>
> In addition, **the regret ($\text{Reg}_t$) exhibits this same adaptive behavior in those environments,** contrary to the reviewer’s intuition “the average regret per round should decrease”.
> - We have added plots of the average regret (Figure 29), showing that the average regret also exhibits unexpected increasing trends at the moment of the distribution shift. These trends in regret are especially distinct in the baseline Exp3-IX-SG, which has slower theoretical convergence.

---

> > ### Comment · Reviewer_ut5G · 2025-11-28
> >
> > Thanks for the response. I agree that the absence of human intervention has some practical motivation in the no-human-in-the-loop environment, and in this region, low-confidence answers have their limitations. Also, I agree that the experimental results align with the transition from simple cases to hard cases. A minor, but not necessary, change could also include a figure that transitions from hard to simple cases. However, I still have concerns about the theoretical analysis, as the trivial policy with $\text{FDR} = \alpha$ still satisfies the guarantee in Lines 253 to 257.
> >
> > Overall, I have some concern about the theoretical analysis, but I agree with the practical motivation and experimental results. Therefore, this seems like a borderline paper, marginally below the acceptance threshold. However, I would not mind if the paper is accepted.

---

> > > ### Author Response · Authors · 2025-11-30
> > >
> > > We sincerely thank the reviewer for the **positive re-evaluation** and for **acknowledging our practical motivation and our experimental results**!
> > >
> > > > However, I still have concerns about the theoretical analysis, as the trivial policy with $\alpha$ still satisfies the guarantee in Lines 253 to 257.
> > >
> > > **Finally, we would like to address and resolve your remaining concern regarding the theoretical analysis.**
> > >
> > > The trivial expert(policy) satisfying the guarantee is **not a theoretical weakness, but rather a critical, intentional feature by design** for ensuring robustness:
> > > - Consider a scenario where the environment is extremely adverse (e.g., the model answers incorrectly on almost all inputs). In such "near worst-case" scenarios, *if no non-trivial expert can satisfy the FDR guarantee, it is natural and necessary for the algorithm to converge to the trivial expert (always abstaining) to maintain the desired $\alpha$ by a user*.
> > > - Also, prior work in Selective Generation [1] (batch setup) handles similar cases—where the guarantee cannot be met on calibration data—by providing a guarantee at the upper bound of the final risk (*even if this exceeds the desired risk level*), signaling that the target risk is unattainable.
> > >
> > > In summary, as discussed in our previous rebuttal ("The Best Expert is Not Trivial"), our algorithm converges to a non-trivial expert whenever one exists.
> > > Thus, **The trivial expert is the best expert only when no other better expert exists, which is fully intended and a necessary condition for a robust guarantee.**
> > >
> > >
> > >
> > > > A minor, but not necessary, change could also include a figure that transitions from hard to simple cases.
> > >
> > > We are happy to confirm that we have already included this scenario in **Appendix G.2 (e.g., Figures 13-14 and 19-20 for NQ $\to$ TriviaQA)**.
> > >
> > > As expected, the plots show that the learner, initially trained conservatively (increasing inefficiency) on hard data, **successfully adapts to the easier data by becoming less conservative (reducing inefficiency) while maintaining the FDR convergence toward $\alpha$.**
> > >
> > > ---
> > >
> > > We truly appreciate your time and engagement throughout this discussion.
> > > **We believe this clarification fully resolves the concern by clarifying that the trivial policy serves as a necessary condition for a robust guarantee. We hope this contributes to a full understanding of our theoretical contribution.**
> > >
> > > ---
> > >
> > > [1] Lee et al., “Selective Generation for Controllable Language Models.”, Advances in Neural Information Processing Systems 37, 2024.

---

### Author Response · Authors · 2025-11-21

We have corrected several typos (e.g., related to the Lemma and experiment settings) and made minor notational corrections in the problem definition.

Furthermore, based on the reviewers’ comments, we have conducted **additional experiments**, which have been added to Appendix G. The following include the added experiments:
- Figure 29: Analysis on the average regret plot in distribution-shift environments
- Figures 30 and 31: Analysis on imperfect supervision
- Figure 32: Evaluation with strategy-aware adaptive adversary

These additions have been *marked in red* in the revised pdf for clarity.

---

### Author Response · Authors · 2025-12-03
**Summary of Updates and Clarifications**

Dear AC and Reviewers,

We sincerely thank  Reviewers for constructive feedback and AC for orchestrating the review process. Here, we provide **a brief summary on our contributions**, followed by a summary of **reviewers’ concerns and our responses to them**. In particular, we have executed comprehensive new experiments covering adversarial robustness (Figure 32), noisy feedback (Figures 30-31), and user-centric metrics (Table in [comment link](https://openreview.net/forum?id=MOW9zrKrNx&noteId=Ao94J7EZEJ)).
**We believe that these additional results, combined with our theoretical clarifications, empirically validate our claims and effectively address all raised concerns.**


## Contribution summary

**First Framework for Online Selective Generation with Adversarial Bandit Feedback**

To mitigate LLM hallucination, we leverage **selective generation**—a certified approach that controls the False Discovery Rate (FDR) at a desired level by abstaining from uncertain answers—as a practical solution given that fine-tuning models or heuristic methods can be costly and theoretically not guaranteed.
However, conventional selective generation relies on static calibration sets under a **strong stochastic (i.i.d.) assumption, making it brittle to distribution shifts**. Furthermore, it requires **full feedback (e.g., ground truth given a question) which is often impractical in real-world scenarios**.

We address these limitations by establishing a **realistic online learning framework with partial(bandit) feedback (e.g., thumbs-up/down)**. Crucially, we assume an **adaptive adversary** to ensure robustness against time-varying data distributions, model performance, or user feedback, thereby providing worst-case guarantees beyond standard stochastic assumptions.

Our key contributions are:
1. **Theoretical Novelty (Reduction to Adversarial Bandits)**: We establish a rigorous link between traditional bandit learning and online selective generation by introducing a novel **Regret-to-FDR conversion lemma (Lemma 1)**. This proves that any bandit algorithm with our specialized loss (4) is also theoretically guaranteed to control the FDR risk.
2. **Algorithmic Innovation (Feedback Unlocking)**: We propose ExSUL, an algorithm designed to overcome the **information scarcity of partial feedback**. By exploiting the monotonic structure of selective generation via Feedback Unlocking (7), which provides a theoretically tight regret bound (Theorem 1) and, in turn, leads to a practically efficient FDR guarantee (5), ExSUL achieves the same regret bound matching that of full-feedback algorithm EW-SG (oracle) and significantly outperforms standard bandit baselines (Exp3-IX-SG).
3. **Robust Empirical Validation**: We demonstrate the practical efficacy of the methods across diverse environments, including stochastic, distribution-shifted, and interactive setups (Figures 3-7). In addition, our new experiments confirm that *ExSUL remains robust even against strategy-aware adversaries and noisy feedback*, validating its applicability to reliable autonomous systems.

---

> ### Author Response · Authors · 2025-12-03
>
> ## Reviewers’ concerns and our responses
> ### Reviewer ut5G
> 1. **Justified Practical Motivation**
> ([comment link](https://openreview.net/forum?id=MOW9zrKrNx&noteId=TGNkxnI0wl)):
> Regarding the concern about the practical motivation of selective generation, we clarified that it is indispensable for **autonomous systems (i.e., no-human-in-the-loop)** and **High-Stakes Domains** (e.g., medical, finance, security). Unlike human-facing warnings, fully automated pipelines require clear, programmatic decisions to abstain to prevent **irreversible catastrophic harm**.
> 2. **Validated Adaptive Behavior**
> ([comment link](https://openreview.net/forum?id=MOW9zrKrNx&noteId=LRWziEkgXn)):
> We explained that the increasing trend in FDR/inefficiency (Figures 4-6) is not a failure but **a result of successful adaptation** to distribution shifts (Hard → Easy / Easy → Hard).
> 3. **Clarified the Role of the Trivial Policy**
> ([comment link 1](https://openreview.net/forum?id=MOW9zrKrNx&noteId=LRWziEkgXn),
> [2](https://openreview.net/forum?id=MOW9zrKrNx&noteId=4ZXFTUa11x)):
> We addressed the misunderstanding regarding Lemma 1 by highlighting that the trivial policy (always abstaining) satisfying the guarantee is a **standard design choice** as in prior selective generation literature, and it is not the target of the algorithm. Specifically, it serves as a **necessary condition** for worst-case robustness. We discussed that our method converges to a non-trivial expert whenever one exists in *Appendix I (Discussion on Selection Efficiency), which is empirically supported by Figures 22-23 and 28*.
>
> Consequently, **thanks to the reviewer's constructive engagement, points (1) and (2) were acknowledged during the discussion period, leading to a positive re-evaluation**.
> Regarding point (3), we further provided an in-depth clarification to solidify our theoretical justification.
>
> ### Reviewer jGcW
> 1. **New Experiment with Strategy-Aware Adversary**
> ([comment link](https://openreview.net/forum?id=MOW9zrKrNx&noteId=kS2HyQv8IX)):
> To address the concern about the gap between theory (adaptive adversary) and experiments, we conducted a new simulation using a GPT-4o agent explicitly prompted to exploit the learner. The results (new Figure 32) demonstrate that ExSUL successfully **controls FDR adapting under active adversarial pressure**, empirically validating our theoretical robustness.
> 2. **Clarified Hyperparameters**
> ([comment link](https://openreview.net/forum?id=MOW9zrKrNx&noteId=CIOLgyT6oC)):
> We provided the theoretical justification for the choice of grid size $\mathcal{H}$, balancing the trade-off between discretization granularity and convergence speed relative to baselines.
>
> ### Reviewer nsoa
> 1. **Model-Agnostic Design**
> ([comment link](https://openreview.net/forum?id=MOW9zrKrNx&noteId=z1FDJlOoVK)):
> We clarified that ExSUL is a model-agnostic, lightweight wrapper that requires no internal access to the LLM. It can be seamlessly integrated into large-scale training pipelines, including RLHF and RLVR.
> 2. **Feedback Unlocking Mechanism**
> ([comment link](https://openreview.net/forum?id=MOW9zrKrNx&noteId=z1FDJlOoVK)):
> We explained that the **"Feedback Unlocking" mechanism leverages the monotonic structure of the selection function** rather than stationarity, ensuring validity in dynamic environments.
>
> ### Reviewer SKdk
> 1. **New Experiment on Noisy Feedback**
> ([comment link](https://openreview.net/forum?id=MOW9zrKrNx&noteId=0BjQZiqySv)):
> We clarified that GPT feedback in the paper acts as a valid proxy for strong supervision.
> In addition, to verify the robustness under imperfect supervision, we conducted additional experiments using **noisy/sparse feedback proxies** (e.g., 5% flipped labels) and evaluating the FDR strong supervision. The results (new Figures 30-31) confirm that ExSUL maintains FDR control and performance advantages even with imperfect supervision.
> 2. **Balancing FDR and Inefficiency**
> ([comment link](https://openreview.net/forum?id=MOW9zrKrNx&noteId=0BjQZiqySv)):
> We provided a practical guideline for balancing FDR and inefficiency; users can dynamically manage this trade-off by adjusting the target risk level $\alpha$ and deployers may adjust the hyperparameter $\lambda$.
> 3. **New Evaluation on User-Centric Metrics**
> ([comment link](https://openreview.net/forum?id=MOW9zrKrNx&noteId=Ao94J7EZEJ)):
> We clarified that **“Efficiency”** is already a user-centric metric, and we additionally evaluated the **“Helpfulness”** of selectively generated answers using a reward model, demonstrating that ExSUL consistently achieves higher helpfulness scores than baselines.
>
>
> We are confident that these extensive updates and clarifications fully address the reviewers' concerns and validate our method.
> We thank you again for your time and consideration.

---

### Meta-Review · Area_Chair_iVyq · 2026-01-06

**Summary:**

This paper proposes ExSUL, an online selective generation framework that reduces selective generation with partial feedback to adversarial bandits, introducing a regret-to-FDR conversion lemma and a feedback-unlocking variant of Exp3-IX, with empirical evaluations under stochastic, shifted, and interactive settings.

Reviewers appreciated the problem motivation, technical ambition, and extensive experiments, noting potential relevance to safety-oriented LLM deployment. However, they consistently raised concerns about theoretical clarity and strength: the regret-to-FDR conversion yields weak and sometimes counterintuitive guarantees, relies heavily on comparisons to trivial baselines, and leaves ambiguities around practical FDR control. Empirically, gains over simpler alternatives are inconsistent, and some trends contradict theoretical expectations.

While the rebuttal clarified intent and added experiments, it did not fully resolve concerns about the core theoretical framing and evaluation validity. Collectively, these issues prevent acceptance at this time.

**Reviewer Concerns:**

Please see my summary.

**Reviewer Scores:**

It is difficult to say.  Overall, the authors provided some solid rebuttal, but it's a subjective judgement for the reviewer whether they would like to raise their score.

---

### Decision · Program_Chairs · 2026-01-26

Reject